# Quantifying local stability and noise levels from time series in the US Western Interconnection blackout on 10th August 1996

**Martin Heßler** [1,2] ✉ **& Oliver Kamps**[1]

Critical transitions necessitate anticipation to prevent adverse outcomes. While many studies focus on bifurcation-induced tipping, noise-induced tipping is also possible. We propose to use the open-source (non-Markovian) Bayesian Langevin estimation to quantify deterministic and stochastic dynamics simultaneously. By analysing bus voltage frequency time series from the Western Interconnection blackout on 10th August 1996, complemented by conceptual network models of its key events, we reveal the interplay of changing local restoring rates and noise levels. Furthermore, a comparison of these findings to the blackout's timeline supports our frequency Langevin model driven by correlated noise. A state change is indicated two minutes before the official triggering event, potentially by establishing a tree-to-line fault. This study highlights the importance of distinguishing destabilising factors for anticipating critical transitions and provides a tool for understanding such events across various disciplines.

Complex systems like Earth's global climate, ecosystems, the human brain, or infrastructure, such as power grids and communication systems, are composed of a large number of interacting parts operating on various scales in space and time.

Often, the observation of such a system gives only partial information about the involved processes and it is not simply possible to identify the reason leading to a certain behaviour. In this context, very prominent phenomena are sudden transitions, so-called tipping events, where the system undergoes a fast transition into a different state. There are numerous pathways to tipping events that can differ strongly in the key mechanisms. This fact significantly complicates the anticipation of such critical transitions, which is desired in a wide range of research fields and everyday life to mitigate or prevent damage and in order to control complex systems[1–10]. To achieve these goals, research is done to develop stability measures or leading indicators based on time series data that are applicable in the absence of detailed knowledge about the underlying laws of the system dynamics[11–16].

Common leading indicators rely on *critical slowing down* (CSD) or *flickering*, which are mentioned as general phenomena in connection with *bifurcation-induced tipping* (B-tipping)[17,18]. Briefly summarised, CSD denotes the phenomenon of decreasing local restoring rates−i.e. an increasing relaxation time−prior to a bifurcation which can lead to higher lag-1 autocorrelation (AR1) $\hat{\rho}_1$ and standard deviation (STD) $\hat{\sigma}$ of a time series (cf. Supplementary Information (SI) S1 and S2 for more details). Ongoing jumps between two branches in a bistable regime are called flickering and can cause an increasing skewness of the data distribution. However, since common leading indicators rely on CSD and flickering, they are inherently limited to the generic B-tipping scenario, which is only one of many pathways to destabilisation.

In addition to B-tipping, rapid variation of the control parameter without crossing a B-tipping threshold can also lead to *rate-dependent tipping* (R-tipping) when the phase space is altered on a time scale much faster than that needed by the system to relax onto the modified stable branch. Furthermore, the overall noise level must be sufficiently

[1]Center for Nonlinear Science, University of Münster, Münster, Germany. [2]Institute of Theoretical Physics, University of Münster, Münster, Germany. ✉e-mail: m_hess23@uni-muenster.de

low to avoid *noise-induced tipping* (N-tipping) to alternative stable states. Notably, N-tipping can occur for parameter configurations far from a critical B-tipping threshold. Such N-tipping is intrinsically hard to predict, but it is a crucial factor in driving tipping events across many natural and technical systems, e.g. power grids[19–21].

In the framework of graph theory and networks[22,23], power grids can be modelled by coupled nonlinear oscillators, such as provided by the Kuramoto model[24–30]. Their operating regimes are threatened by the risk of desynchronisation, which leads to power outages. Such outages can occasionally have severe consequences for industry and private consumers. Fortunately, unlike in many other sciences, e.g. ecology[31], highly resolved data of characteristic observables, such as bus voltage frequency $\omega$ time series, are readily available today.

Previous studies focused on destabilisation due to B-tipping[20,21]. However, power grids are continuously exposed to intrinsic variable noise, e.g. from renewable energy sources such as wind power or solar stations (cf. Supplementary Box S1a)[32,33]. Furthermore, the currently pressing trend towards an environmentally sustainable grid architecture is likely to significantly increase the number of noisy grid participants in the future[34,35]. Under these circumstances, N-tipping, or at least the influence of internal noise, can play an essential role in destabilising power grids[36].

It would be worthwhile to have the opportunity to monitor changes in both the local restoring rates and noise levels simultaneously at a specified time resolution to effectively control the system, gain a better understanding of the frequency dynamics before power outage events, and, in rare cases, potentially avoid power outages.

In principle, the *Bayesian Langevin estimation* (BLE)[37,38] is able to do so. Specifically, it assumes a frequency $\omega$ Langevin model[39] (cf. also Supplementary Box S1a for details regarding its connection to the high-dimensional power grid state $\underline{x}$)

$$\dot{\omega}(\omega(t), t) = h(\omega(t), t) + g(\omega(t), t) \cdot \Gamma(t), \qquad (1)$$

with the drift $h(\omega(t), t)$, the diffusion $g(\omega(t), t)$, and stochasticity $\Gamma(t)$. The BLE provides access to robust estimates of local restoring rates through the drift term, and to noise level estimates via the diffusion term for a given time scale resolution. Decreasing restoring rates directly relate to CSD prior to B-tipping. Monitoring of increasing noise levels can be beneficial when a system is suspected to exhibit multiple stable states. Here is a sketch of the method: We parameterise a one-dimensional stationary drift function $h(x)$ as a third-order polynomial with a constant diffusion $g(x) = \text{const.} \equiv \sigma$. The *probability density functions* (PDFs) of the parameters are obtained by *Markov Chain Monte Carlo* (MCMC) sampling, which enables a statistically sound definition of *credibility bands* (CBs). Throughout this article, we use 16% to 84% and 1% to 99% percentile CBs for the measures of interest, i.e. the drift slope $\zeta = \frac{dh(x)}{dx}\big|_{x=x^*}$, representing local restoring rates, and the noise level $\sigma$. The drift slope $\zeta < 0$ quantifies the local restoring rate of the given observable. It approaches zero when a B-tipping event occurs. The noise level $\sigma$ accounts for stochastic contributions in the observed data, which may trigger N-tipping in bistable systems. Furthermore, we take advantage from the expanded *non-Markovian BLE* (NBLE) procedure. This expansion is two-dimensional: In addition to the measured data, it assumes these observations to be driven by a hidden *Ornstein-Uhlenbeck* (OU)[39] process. Under these circumstances, the observed process is described by non-Markovian dynamics. The hidden process can represent fast-scale red noise, but also the slow dynamics of an unobserved driving process. We denote the drift slopes derived this way by $\zeta_{\text{NBLE}}$ and the noise level by $\Psi$. See Methods and Box 1 for more details on the mathematical background.

For power grid analysis, the (N)BLE approach is well justified by the macroscopic *aggregated swing equation* (ASE; Eq. S.4 in the SI)[36,40–42] for stable grid conditions and less coarse-grained model simulations (cf. SI S3). Under certain constraints, the mesoscopic frequency dynamics of an $N$-node power grid, described by the less coarse-grained *classical swing equation* (CSE; Eq. S.5 in the SI) can be summarised by the coarse-grained ASE as the center-of-inertia frequency with dynamics that follow an OU process[39], i.e. a Langevin-type equation. Nevertheless, the ASE is a model approximation, and real-world bus voltage frequency data can exhibit nonlinear drift behaviour[43]. This is also the case for the analysed time series in this article (cf. SI S4). Therefore, by using the (N)BLE, a heuristic nonlinear drift is assumed in the ASE.

The total balance between power generation and consumption is critical for maintaining the stable synchrony of bus voltage frequency, typically 60 Hz (e.g. USA, Japan) or 50 Hz (e.g. Europe, also Japan). Variations in the electric energy generation and consumption of grid components (some of which are shown as an example in Supplementary Box S1a) thus affect the frequency dynamics. This is why the bus voltage frequency $\omega$ is a macroscopic key observable for assessing grid stability in control rooms.

Power outages as well as control mechanisms live on time scales of seconds up to several minutes (cf. SI S5). For these reasons, we focus on the sub-second to minute resolution. Since our theoretical considerations (cf. also SI S6 for B- and N-tipping, and S7 for R-tipping) are largely supported by the NBLE analysis of frequency data from the North America Western Interconnection (NAWI) cascading failure on 10th August 1996, the timeline of this historic event is briefly summarised in Fig. 1a. A more detailed description is left for SI S8. Please note that all times in this article are given in the local Pacific Daylight Time (PDT).

Not least, we emphasise that the (N)BLE incorporates correlated fast-scale contributions, which are typically relevant in time series applications, in a straightforward way—a feature that standard leading indicators lack. Additionally, the (N)BLE outperforms existing indicators, such as AR1, STD, direct drift-diffusion estimation[38,44,45], and the linear approximation through OU estimation (OUE)[46]. The (N)BLE is significantly more stable and precise, as demonstrated for synthetic and real-world datasets (cf. SI S1, S2, and S9). Moreover, the (N)BLE is generally applicable across various research fields, including palaeoclimatology[10], climate science[47], theoretical ecology[38], population dynamics of living bacteria[1], power grids, and more (cf. SI S10).

In this work, we first demonstrate the (N)BLE's performance by applying it to four synthetic time series from different systems. We then investigate two bus voltage frequency time series which cover periods before and after the power outage of the NAWI on 10th August 1996, with the NBLE to gain insights into the complex interplay of the power grid's stability and varying noise influence. As suspected, aside from possible B-tipping, the noise level plays an important role for the emerging power outage. A previous study by Ehebrecht[48] supports our findings on the pre-outage time series by a related approach[44,45]. In contrast to this study, we additionally contextualise our findings regarding the pre-outage and post-outage time series in close comparison to the real timeline of events and quantify uncertainties in the estimates. This leads to substantially deeper insights into the impact of certain events during the outage cascade and their modelling fingerprints. Furthermore, it largely confirms the hypothetical considerations in Supplementary Boxes S1 and S2. Moreover, we provide evidence for several mistakes in the derived time scale of the second post-outage time series used in previous studies[20,48–50], which is most likely explainable due to sparse literature sources of low print quality (cf. SI S11–S13 for more details). The subsequent misinterpretation due to incorrect time scaling and direction erroneously implies that cascading failures might be typically preceded by smoothly changing warning metrics based on the theory of CSD and B-tipping. This might be surprising, considering that grid components can fail rather abruptly. Although our analysis supports basic theoretical considerations of refs. 20,48–50, it reveals a significantly more complex picture:

## BOX 1 (NON-MARKOVIAN)

# Bayesian Langevin estimation

**1 Initialise**: Load time series $x(t)$. Choose window size $N_w$ and window shift $N_{sh}$. Define drift $h_{(x)}(x) = h_{MC}(x)$, diffusion $g_{(x)}(x) = const. = \theta_4$, noise type $\Gamma(t)$ (BLE) or $y(t)$ (NBLE), percentiles $q_a$, $q_b$ (in percent) for credibility bands (cf. second illustration below, right part), along with prior types and ranges, all according to the "Estimation schemes" section. For simplicity, the following notation assumes the BLE with drift slope $\zeta$ and noise level $\sigma$; for the NBLE, replace $\zeta$ and $\sigma$ with $\zeta_{NBLE}$ and $\Psi$, respectively.

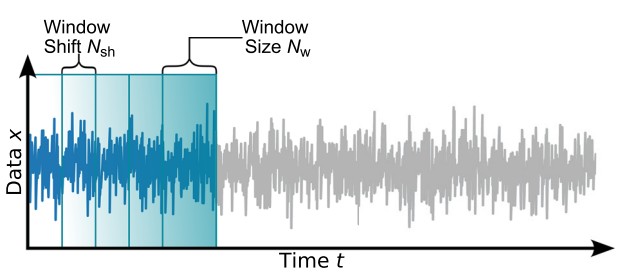

**2 Perform estimation per window**:

**2.1** Draw parameter sets $\hat{\underline{\theta}}$ from the joint posterior distribution $p(\underline{\theta}|\underline{x}, \mathcal{I})$ via Markov Chain Monte Carlo sampling based on the current window's data $\underline{x}$ and background information $\mathcal{I}$.

**2.2** Transform the sampled parameter sets $\hat{\underline{\theta}}$ into drift slopes $\hat{\zeta}$ and noise levels $\hat{\sigma}$.

**2.3** Approximate the marginalised posterior distributions $p(\zeta|\underline{x}, \mathcal{I})$ and $p(\sigma|\underline{x}, \mathcal{I})$ by summing the density kernels of the corresponding samples (cf. illustration below).

**2.4** Read out and store maximum posterior estimates and predefined percentiles for credible intervals (cf. illustration below).

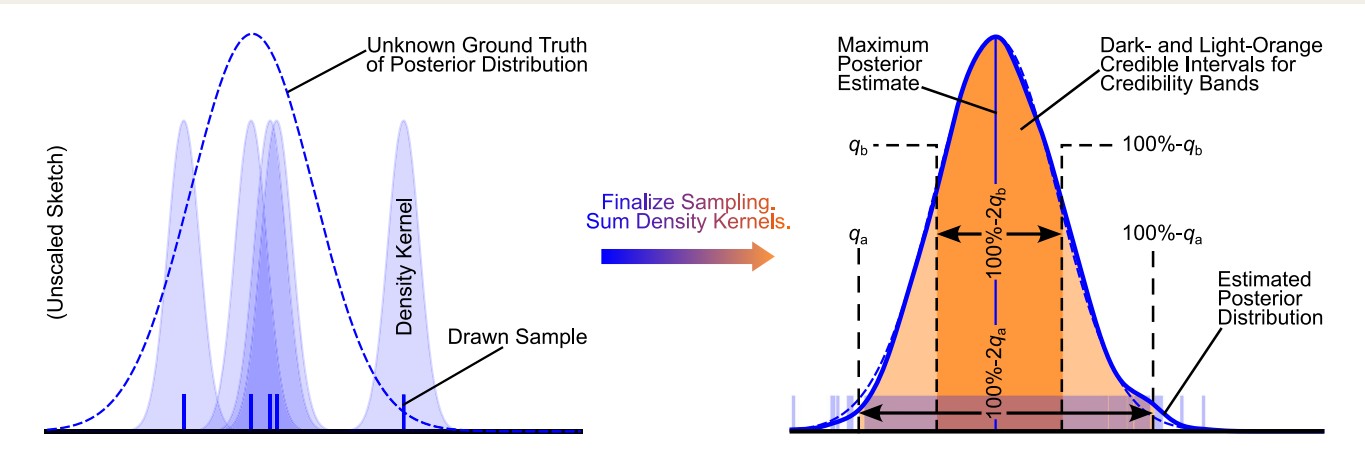

**3 Shift window and repeat:** Shift to the next rolling data window and repeat Step 2 until all windows are processed.

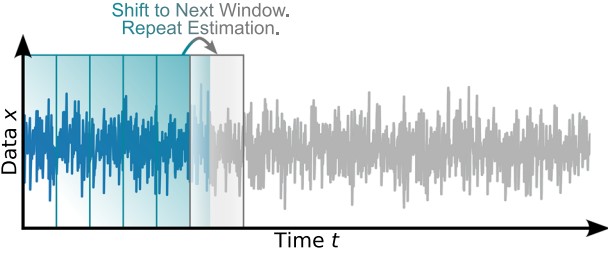

**4 Collect output:** Time evolution of drift slopes $\hat{\zeta}(t)$ and noise levels $\hat{\sigma}(t)$ including CBs.

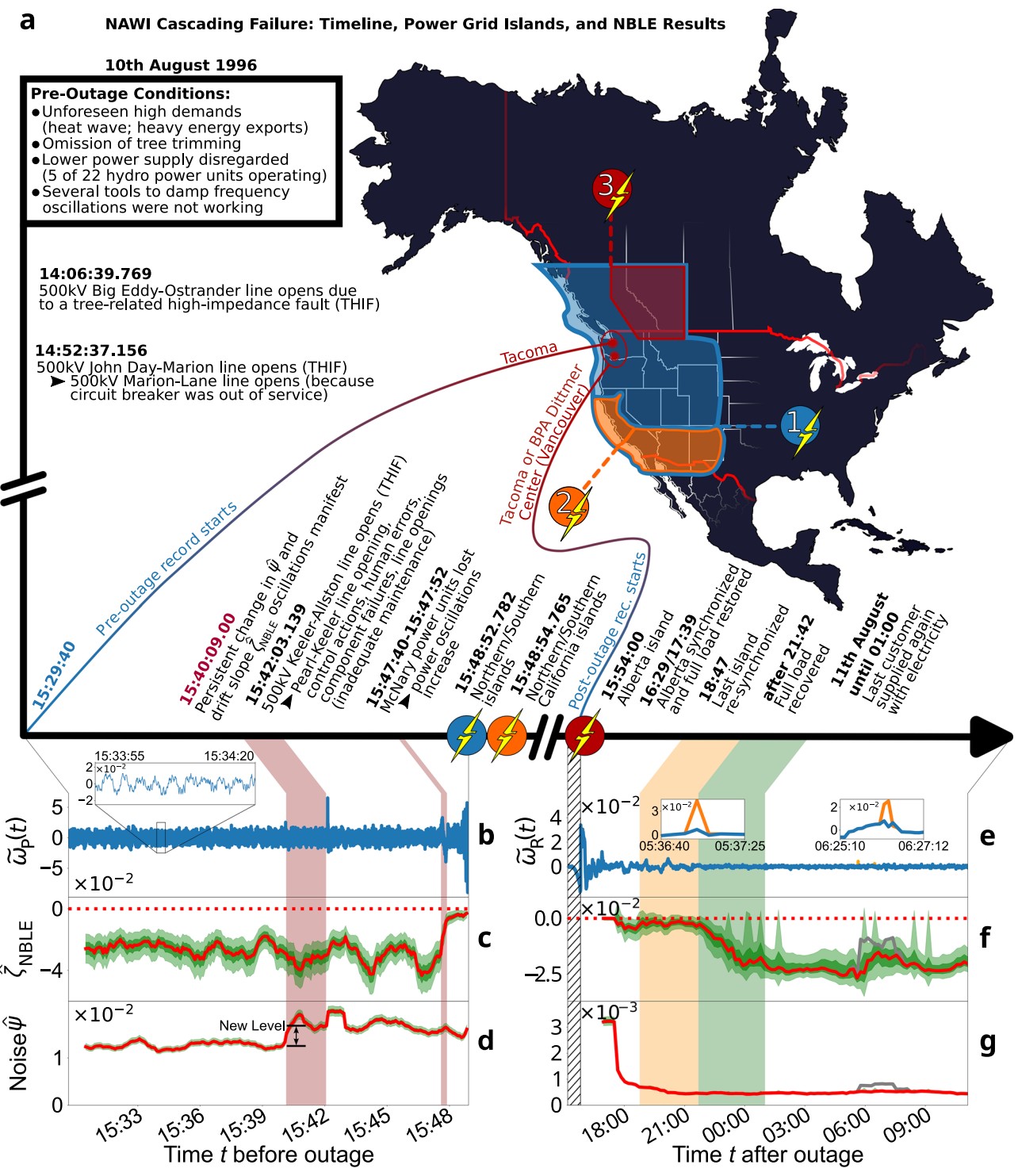

**a**  NAWI Cascading Failure: Timeline, Power Grid Islands, and NBLE Results

The dynamics change rather abruptly related to real-world grid modifications, which can contribute in a stochastic and/or deterministic manner. Our results capture these state changes and account for the different contributing scales from stochastic to deterministic influence. Finally, a brief discussion of the results is given and complemented by suggestions for future research.

## Results

In "Studies on synthetic data", the (N)BLE is applied to four synthetic test cases, before the method is used to analyse the NAWI power outage frequency $\omega$ in "North America Western Interconnection power outage on 10th August 1996".

## Studies on synthetic data

The following four datasets and two corresponding models are introduced to demonstrate that the method can track the noise level that eventually leads to N-tipping, while simultaneously providing local restoring rates playing a role for B-tipping. The first example dataset is simulated with $4 \cdot 10^4$ data samples of the pitchfork model equations:

$$h(x) = \nu \cdot x - x^3 \qquad g(t) = \sigma(t) = \begin{cases} 0.05 & \text{for } t = [0, 500], \\ 2.4 \cdot 10^{-4} \cdot t + 0.05 & \text{for } t = (500, 1750], \\ 0.35 & \text{for } t = (1750, 2000]. \end{cases}$$

$$(2)$$

**Fig. 1 | Pre- and post-outage NBLE results aligned with the NAWI blackout's key events.** The results are robust to variations in window size (cf. SI S16). Dark- and light-green shadings represent 16% to 84% and 1% to 99% percentile credibility bands, respectively. Shaded time intervals refer to the listed events. **a** Timeline (in PDT) and power grid islands[61]. BPA denotes Bonneville Power Administration. **b** Detrended pre-outage bus voltage frequency $\tilde{\omega}_P(t)$. The data oscillations at - 0.2 Hz to 0.4 Hz (cf. inset) are not captured by the NBLE drift (cf. SI S3, S8, and S15). **c** Following the 500 kV Keeler-Allston line trip, a turbulent period of control actions and destabilising events is reflected by pronounced fluctuations around drift slope $\tilde{\zeta}_{\text{NBLE}} \approx -3.3$. Estimates increase sharply with the loss of the McNary power units, exhibiting a smoother trend toward the islanding process, consistent with the concepts in Supplementary Box S1, as grid component failures affect the frequency drift rather than noise levels. **d** Noise levels $\dot{\Psi}$ increase sharply about 2 min before the 500 kV Keeler-Allston line trip, indicating greater impact of fast-scale phenomena, possibly due to the *tree-related high-impedance fault* (THIF) or a

sudden load imbalance (cf. SI S17). **e** Detrended bus voltage frequency $\tilde{\omega}_R(t)$ during restoration. The black-hatched interval's end coincides with the end of $\tilde{\omega}_P(t)$. **f** Initial drift slopes $\tilde{\zeta}_{\text{NBLE}} \gtrsim 0$ are unreliable, as the windows include pre-outage data. The barely stable grid state during the orange-shaded key restoration interval is reasonable, as most of the grid components are recovered only stepwise throughout this period, with some not restored until much later, on 16th August 1996. The system steadily approaches the physical pre-outage configuration, including lines, power units, and loads, reflected by drift slopes reaching a stable plateau during the green-shaded period. Nine windows exhibit spiking credible intervals due to outliers. **g** Initial noise levels $\dot{\Psi}$ are unreliable (cf. **f**). The steepening potential during restoration lowers the noise levels as the system stabilises in the operational fixed point (cf. Supplementary Box S1b). **f**, **g** Grey lines show outlier (orange in insets) effects, causing a bias of roughly one window length (cf. Methods; SI S18).

The control parameter is fixed at $v = 1$ and the simulations are performed via the Euler-Maruyama method[39]. The model's two stable states are confined to regions around the modes of the bimodal state PDF at $\pm 1$[51]. The realised simulation undergoes an N-tipping transition into a flickering state due to the increasing noise level. The example is referred to as the *N-tipping dataset* $x_g$. The index notation indicates the terms that are varied in each time series.

The second system, called the simultaneous *drift-diffusion-varying dataset* $x_{h,g}$, is governed by a fold model

$$h(x) = r + x - x^3 \qquad g(t) = \sigma(t) = \begin{cases} 9.5 \cdot 10^{-4} \cdot t + 0.05 & \text{for } t = [0, 1000], \\ -9.5 \cdot 10^{-4} \cdot (t - 1000) + 1 & \text{for } t = (1000, 2000). \end{cases} \tag{3}$$

This model features linearly increasing noise from 0.05 to 1 in the first half [0, 1000] of the simulated time interval, followed by decreasing noise in the second half (1000, 2000). At the same time, the control parameter $r$ increases linearly from 5 to 15 over the entire time interval, with $3 \cdot 10^4$ samples computed using the Euler-Maruyama method. The system could potentially undergo a fold bifurcation at $r_{\text{crit}} = 0$ and, consequently, tends to stabilise with increasing $r$ in the test case. Thus, the system could be envisioned as a power grid in which frequency diffusion is driven by noise of variable amplitude $g(t)$, potentially including consumer decisions or renewable energy sources, while the drift is increasingly stabilised by damping, control actions, or suitable infrastructure extensions (cf. Supplementary Box S1).

As stated in Supplementary Box S1, there is a relationship between the grid components, the high-dimensional power grid states $\underline{x}$, and the related frequency $\omega$ Langevin model. By applying the (N)BLE, we consider a macroscopic frequency model motivated by the ASE (cf. SI S3). This means that the noise level estimates $\hat{\sigma}$ and $\dot{\Psi}$ can reflect the absolute noise changes of less coarse-grained levels, such as those from the orange-tiled components in Supplementary Box S1, as well as relative changes of their impact on the macroscopic ASE scale. For example, modified damping strengths on mesoscopic scales result in increasing macroscopic ASE noise estimates, even though the overall mesoscopic noise level remains constant. Fast macroscopic signal contributions are less damped and become more relevant for explaining the data at the macroscopic descriptive level. Expectedly, the drift slope estimates increase as well, indicating lower damping; i.e. weaker control leads to reduced deterministic stability of the frequency state (cf. SI S3). However, if a stricter distinction between absolute and relative noise strengths than that provided by the current (N)BLE is desired, this can be achieved by introducing a suitable high-dimensional mesoscale model in the (N)BLE estimation procedure, potentially incorporating high-dimensional time series feed-in. Finally, we remark that the (N)BLE does not formally distinguish between signal noise and measurement noise (cf. SI S10 for an illustrative

example[1]). If necessary, extensions to explicitly incorporate measurement noise in these types of models are accessible[52–54].

Next, we simulate two models with correlated noise to demonstrate the NBLE as a correction to the basic BLE under non-Markovian conditions. The third example, called the *correlated B-tipping dataset* $x_h^{\text{corr}}$, illustrates how the NBLE's noise estimate $\dot{\Psi}$ can correct for the biased BLE noise estimates $\hat{\sigma}$ when the non-Markovian system with hidden slow-scale dynamics approaches a bifurcation in the fast observed process $x_h^{\text{corr}}$. The correlated B-tipping dataset $x_h^{\text{corr}}$ is simulated via the drift, coupling, and diffusion terms of $x$ and $y$ (cf. notation of Eq. (18)):

$$h_x(x) = r + x - x^3 \qquad g_x(x) = \text{const.} \equiv q = 0.5$$
$$h_y(y) = -c \cdot y = -0.75 \cdot y \qquad g_y(y) = \text{const.} \equiv \sigma_y = \sqrt{c}. \tag{4}$$

A B-tipping destabilisation is reached by a linear shift of the control parameter $r$ from 15 to $-5$ over the simulated time interval. The relation between the inverse correlation length $c$ and the noise level $\sigma_y$ is chosen to reduce the number of independent model parameters for demonstration purposes of the NBLE method. Nonetheless, the NBLE is relatively robust to imperfect parameterisation of the drift and diffusion parameters, as demonstrated by our last example, the *correlated drift-diffusion varying dataset* $x_{h,g}^{\text{corr}}$. It introduces a twofold misfit in the NBLE. The process $x$, following the pitchfork model Eq. (2), is multiplicatively coupled through $g_x(x)$ to a slower-scale OU process $y$. Furthermore, the NBLE's correspondence between the inverse correlation length and the diffusion, i.e. $\sigma_y = \sqrt{c}$, is undermined by a time-dependent diffusion $g_y(t)$:

$$h_x(x) = \alpha \cdot x - x^3 \quad g_x(x) = x \quad h_y(y) = -c \cdot y = -0.75 \cdot y \quad g_y(t) = \sigma_y(t) \tag{5}$$

$$\text{with} \quad \sigma_y(t) = \begin{cases} 1.45 \cdot 10^{-3} \cdot t + 0.05 & \text{for } t = [0, 1000] \\ -1.45 \cdot 10^{-3} \cdot (t - 1000) + 1.5 & \text{for } t = (1000, 2000). \end{cases} \tag{6}$$

The control parameter $v$ is increased linearly from 5 to 15. The additive noise coefficient $\sigma_y(t)$ increases linearly over the range [0.05, 1.5] in the first half of the simulation, before it decreases linearly to its starting value $\sigma_y(0) = 0.05$.

The simulations and the results of the analyses are presented in Fig. 2. The example datasets are analysed with the (N)BLE in windows of size $N_w = 2 \cdot 10^3$ points with a shift of 100 points per window. For the analyses in Fig. 2d, g, j, the data from each window are linearly detrended to account for the non-stationary trend in the mean.

In Fig. 2a, the N-tipping dataset $\underline{x}_g$ is presented. The red dotted vertical line indicates the time $t_{\text{N-tip}} \approx 1386.8$ at which the N-tipping transition into a flickering regime occurs. Since the control parameter

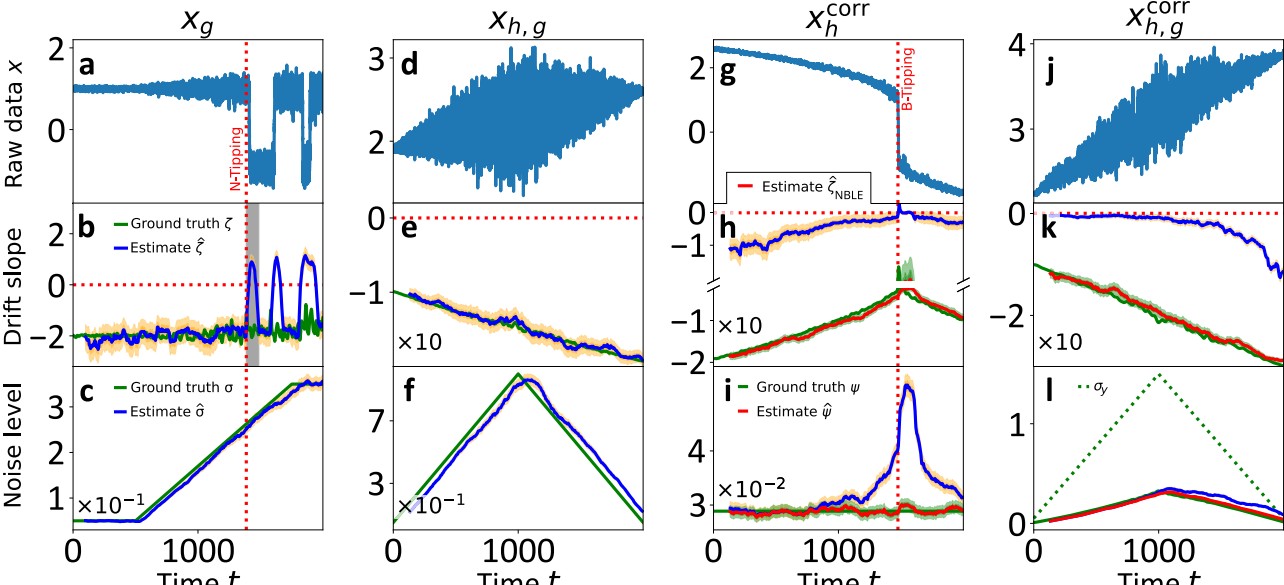

**Fig. 2 | (N)BLE applied to prototype synthetic examples.** Dark- and light-orange shadings represent 16% to 84% and 1% to 99% percentile credibility bands (CBs) for the BLE, respectively; the same applies in green for the NBLE. **a, d, g, j** The results of the (N)BLE applied to the four test sets. The red, dotted vertical lines in (**a–c, j–l**) indicate the approximate times when noise-induced tipping (N-tipping) into a flickering regime and bifurcation-induced tipping (B-tipping) take place, respectively. The example datasets are analysed within windows of size $N_w = 2 \cdot 10^3$ points with a shift of 100 points per window. The estimates' shift in time is due to the rolling window approach and ascribing the estimates to the last point of each window. Ascribing them to the midpoint of each window makes unbiased estimates match the true values almost perfectly (cf. SI S14, Fig. S12). In the examples (**d, g, j**), the data of each window are linearly detrended to account for the non-stationary trends in the mean. **b** At time $t = 1386.8$, the N-tipping causes artificial drift slope peaks with the width of one rolling time window as indicated by the grey-shaded area. **b, c, e, f** The BLE $\hat{\zeta}$ and $\hat{\sigma}$ are unbiased for the Markovian examples. **h, i, k, l** The BLE yields strongly biased estimates which work as qualitative leading indicators (due to correct trends), similar to AR1 $\hat{\rho}_1$ and STD $\hat{\sigma}$, apart from the noise level estimates in vicinity of the bifurcation point in (**i**). The BLE bias in (**l**) increases because of the increasing influence of the hidden process $y$ due to multiplicative coupling via $x$. The strong BLE bias is perfectly compensated by the NBLE estimates. It mirrors the constant noise before the bifurcation in (**i**) and also works under the imperfect model parameterisation in (**k, l**).

is fixed at $\nu \equiv 1$, the computed drift slopes $\hat{\zeta}$, shown in Fig. 2b, remain approximately constant prior to time $t_{N-tip}$. After this time, a sharp jump indicates sudden destabilisation caused by emerging flickering, leading to artificial peaks in the leading indicator $\zeta$. These artefacts $\hat{\zeta} \geq 0$ span the width of one window length, as marked by the grey-shaded area. The constant level of the leading indicator before time $t_{N-tip}$ aligns with the true drift slope values $\zeta$, shown as solid green line, which correctly indicates that no B-tipping occurs.

But if there is no B-tipping event, how can we identify the destabilising mechanism? In such cases, the BLE can yield valuable information about the ongoing phenomena. While the deterministic dynamics remain unchanged, the increasing noise level $\sigma$ is well estimated, as shown in Fig. 2c. The BLE results accurately follow both the noise plateaus and the linear ramp of the noise level. Small deviations from the true values are expected for two reasons. First, the rolling window approach introduces a time lag. Second, the online reasoning of the BLE requires assigning the estimates to the last time stamp of each window, in order to align the most recent stability information with the current time stamp. If the estimates were ascribed to the midpoints of their respective windows, they would almost perfectly match the true values (cf. SI S14). Even though the noise level estimates cannot definitively determine whether an N-tipping event will occur, they provide valuable information about increasing noise levels and, potentially, a higher chance for N-tipping occurring in multistable systems.

In the previous N-tipping example, $x_g$, the drift is unchanged. The slope estimates $\hat{\zeta}$ in Fig. 2e correctly mirror the stabilising effects of a decreasing control parameter $r$, showing a negative trend that aligns with the true drift slope values $\zeta$. As illustrated in Fig. 2f, changes in the noise level are precisely reproduced while accounting for deviations due to rolling windows and online reasoning. In summary, the BLE

analysis of the two synthetic uncorrelated Markovian examples demonstrates that using the BLE can be advantageous for quantifying local restoring rates and noise levels and for controlling systems in which both B- and N-tipping may play a role.

In principle, the BLE is limited to cases in which the general model ansatz of a Markovian Langevin equation holds. The analysis of the correlated B-tipping dataset $x_h^{corr}$ in Fig. 2g–i illustrates the potential BLE bias if the Markov assumption is not sufficiently fulfilled. The increasing drift slope estimates $\hat{\zeta}$ in Fig. 2h are strongly biased, but correctly suggest the approaching B-tipping event by a positive trend. Thus, they might be useful as qualitative indicators, similar to AR1 $\hat{\rho}_1$ and STD $\hat{\sigma}$, in this case. However, the noise level estimates $\hat{\sigma}$ exhibit an artificial positive trend in the vicinity of the bifurcation point and are not even qualitatively correct. The constant true noise level $\Psi_h^{corr} = q \cdot \sqrt{c} \cdot dt$ (cf. Eq. (20)) is shown by the green solid line. A BLE noise increase before a B-tipping event should be carefully interpreted in such cases, but at least the approaching B-tipping event is qualitatively mirrored by the BLE drift slopes $\hat{\zeta}$. The advanced NBLE turns out to be advantageous for the correlated B-tipping dataset $x_h^{corr}$ and related cases. In these basic examples, the NBLE completely corrects for the BLE bias. In particular, it does not exhibit the artificial increase in the noise levels that is observed for the BLE.

In order to complement these results, the generating process of the data $x_{h,g}$ is modified in a way that violates the NBLE model parameterisation. The analysis results of this case are shown in Fig. 2k, l: The BLE estimates are qualitatively—but not quantitatively—reasonable as in the previous example. Interestingly, the BLE noise level estimates $\hat{\sigma}$ still agree rather accurately with the ground truth $\Psi_{h,g}^{corr} = x(t) \cdot \sigma_y(t) \cdot dt$, especially in the first half of the simulated time range. The multiplicative coupling increases with the positive trend in the data $x_{h,g}^{corr}$ and the hidden correlated process $y$ becomes more

significant. This is probably the reason for the increasing bias in $\hat{\sigma}$ over the second part of the time range. Far away from a bifurcation, both BLE metrics are thus useful qualitative indicators. However, the analysis of the data $x_{h,g}$ reveals that the NBLE—despite its imperfect model parameterisation in this case—yields accurate results. The NBLE drift slopes $\hat{\zeta}_{\mathrm{NBLE}}$ and noise levels $\hat{\Psi}$ are still perfectly unbiased. The explanation for this is found by considering the explicit MCMC estimates of the NBLE $x$-coupling term $g_x(x) \equiv \theta_4$ and the OU parameter $\theta_5^2 = c^{-1}$. The varying diffusion $\sigma_y(t)$ is compensated by a suitable variation of $\theta_4$, whereas the OU parameter $\theta_5$ remains almost constant. The fact $\hat{\theta}_5 \approx \mathrm{const.}$ is in agreement with the constant correlation length $c^{-1}$ of the correlated drift-diffusion varying dataset $x_{h,g}^{\mathrm{corr}}$. This means that the drift slope and noise level estimates of the resulting process can be unbiased, although individual parameter estimates $\underline{\theta}$ do not agree with the data-generating process.

## North America Western Interconnection power outage on 10th August 1996

Keeping in mind the performance and robustness of the (N)BLE, demonstrated in "Studies on synthetic data", we apply the NBLE to two bus voltage frequency time series, $\omega_P(t)$ and $\omega_R(t)$, which cover a historic major cascading failure, namely the NAWI blackout on 10th August 1996. The NBLE is preferred to the BLE because it better reproduces key statistics of the datasets through inclusion of correlated fast-scale dynamics. For details, see SI S3 and S15.

**Pre-outage interval.** We start our discussion with bus voltage frequency data $\omega_P(t)$ from the pre-outage (index P) time interval, and conclude our considerations with the analysis of restoration (index R) frequency data $\omega_R(t)$ from the post-outage period. The results obtained from the NAWI frequency time series and from complementing model simulations presented in the following are robust to variation in window size (cf. SI S16).

In Fig. 1b–d, we present the NBLE results for the time series segment $\omega_P(t)$, covering the entire timeline of events leading up to the power outage. The analysed data $\tilde{\omega}_P(t)$, shown in Fig. 1(b), represent the original frequency time series $\omega_P(t)$ detrended using a Gaussian kernel-smoothed version with kernel bandwidth $\sigma_k = 5\,\mathrm{s}$, transforming the original data into a stationary version that is suitable for statistical analysis. The NBLE is applied in windows of $1 \cdot 10^3$ data points (i.e. 50 s), which are shifted by 100 points (i.e. 5 s). As shown in the inset of Fig. 1b, the data feature a periodicity of roughly 0.2 Hz to 0.4 Hz (cf. SI S15) that is not captured by the (N)BLE models. For details, please see SI S3, S8, and S15. In the following presentation of the results, note the high accuracy of the report's time stamps, as most of the grid components were satellite-synchronised.

The time series $\omega_P(t)$ was provided by Bonneville Power Administration (BPA) via a Freedom of Information Act (FOIA) request. Thanks to very cooperative correspondence, we could clarify the time series' metadata (cf. SI S11): The data record is sampled with equidistant time steps $\Delta t = 0.05\,\mathrm{s}$ and was measured in Tacoma. The record starts at 15:29:40 and ends at 15:48:54.95, which approximately coincides with the emergence of the first three islands (cf. Fig. 1a).

The historically significant cascading failure was triggered by the opening of the 500 kV Keeler-Allston line due to a *tree-related high-impedance fault* (THIF)[55,56], causing a flashover and subsequent tripping of the line at 15:42:03.139. This event, marked by the end of the first red time interval in Fig. 1b–d, led to the almost simultaneous tripping of the Keeler-Pearl line. The sharply pronounced positive frequency deviation peak at this time mirrors the line tripping event. Based on theoretical considerations (cf. Supplementary Boxes S1 and S2), variations in the NBLE metrics $\hat{\zeta}_{\mathrm{NBLE}}$ and $\hat{\Psi}$, are expected to yield a fingerprint of the Keeler-Allston line tripping. Indeed, a positive trend in the noise level estimates towards a new stable plateau (cf. vertical dimension line with double-headed arrows in Fig. 1d), along with

moderate oscillatory variations in the drift slopes $\hat{\zeta}_{\mathrm{NBLE}}$, can be observed starting roughly 2 min prior to the recorded line tripping event of the approved disturbance report[57] (cf. Data Availability section for access details). The approximate starting time of the noise level change is highlighted by the beginning of the first red time interval at 15:40:09.00. Simultaneously, there is a notable frequency dip in the original time series $\omega_P(t)$ (cf. SI S17), which may indicate a sharp load increase in the grid. This change in dynamics, preceding the actual line tripping, aligns well with technical aspects of THIFs and inspires two plausible explanations:

1.  There is a non-vanishing probability that the contact between the Keeler-Allston line and the tree was established around 15:40:09.00. The start time of the first red interval would thus correspond to the beginning of the THIF. A developing THIF typically leads to an immediate load increase on the directly affected line, and possibly surrounding grid components later, due to network dependencies. Although a THIF can evolve in various ways, it typically occurs stepwise. Initial tree-to-line contact, along with any resulting line damage, can significantly increase electrical resistance, leading to a greater load when the line is partially transferring energy. Eventually, if a permanent tree-to-line contact is established, progressive carbonisation can reduce the initially high electrical resistance of the wood[55,56,58,59]. Such a scenario is supported by the frequency dip at this time (cf. SI S17). Even if the THIF had caused a fairly sudden drop in load instead, such frequency dips are still possible due to the complex network response and control actions. This may lead to the greater impact of fast-scale phenomena, i.e. increasing noise level $\hat{\Psi}$. The emerging oscillations of the drift slope $\hat{\zeta}_{\mathrm{NBLE}}$ might thus resemble the interplay of alternating primary control actions and additional destabilising events (cf. timeline in Fig. 1a) under the increased noise stress. Of course, we cannot definitively deduce causal relations between noise level and drift slope, even though the explanation fits well into the overall picture. Deterministic changes could also contribute to the increased impact of fast-scale phenomena (cf. SI S3). However, following the THIF scenario, the grounded Keeler-Allston line led to modified drift-diffusion dynamics, characterised by drift slope oscillations and new noise level plateaus reached after 15:40:09.00. Once grounded via vegetation, it takes seconds up to several minutes for a THIF to be fully established before the line trips[58,59]. Thus, the actual tripping event at 15:42:03.139 falls within a reasonable time range. These findings suggest that THIFs could leave fingerprints in the (N)BLE metrics derived from bus voltage frequency data in some cases, providing a potential starting point for future research on THIF detection.

2.  Though it seems plausible that the changing dynamics are directly related to the actual Keeler-Allston line tripping due to a THIF, we cannot establish with certainty a causal relationship between the state change and an emerging THIF. However, even if we assume that the frequency dip at 15:40:09.00 is not caused by the THIF, it likely corresponds to a sudden load increase for some other reason. From an engineering perspective, such a load increase—especially during the hot weather documented for the 10th August 1996—favours the occurrence of THIFs. As the load increases, a power line heats up and elongates, leading to significant sagging. In consequence, the distance between the power line and the underlying vegetation decreases, heightening the risk of a THIF.

Even though the exact circumstances cannot be deduced from the NBLE, the method clearly identifies a significant change in the dynamics ~2 min before the actual line tripping event, and it appears likely to be related to the triggering event of the NAWI blackout on 10th August 1996.

The subsequent line trippings forced the McNary units to increase the reactive output to 494 MVAR, before they began tripping due to technical issues with the excitation equipment, which triggered actions of the system protection protocol. The NBLE identifies the lost reactive power of the McNary units by strongly increasing drift slopes $\hat{\zeta}$. This supports the depiction of the Langevin model for power grid frequency dynamics, as the McNary power units have a notable influence on the deterministic frequency dynamics (cf. Supplementary Box S1).

The NBLE mirrors the barely stable state with increasing power, voltage, and consequently, frequency oscillations following the McNary loss through the new drift slope level $\hat{\zeta}_{\text{NBLE}} \approx -1$, which is less stable than the state prior to the 500 kV Keeler-Allston line flashover. Ultimately, the drift slope estimates $\hat{\zeta}_{\text{NBLE}}$ tend to increase more gradually, reflecting the progressive destabilisation due to increasing frequency oscillations, and approach zero when the first two of the three island splittings occur. Moreover, in SI S3 we complement our empirical (N)BLE results, obtained from the frequency records, by modelling three key episodes of the NAWI blackout, namely

(i)  the NAWI pre-outage conditions, characterised by more dominant fast-scale stochastic dynamics and greater system stress compared to the post-outage interval (with the pre-outage noise levels being approximately one order of magnitude higher),

(ii)  the 500 kV Keeler-Allston line THIF,

(iii)  and the loss of the McNary power units, followed by insufficient damping.

We provide a simple model that qualitatively reproduces the key characteristics of the (N)BLE signatures over a wide range of window sizes (cf. SI S16) and consolidates the relationships between power grid components and macroscopic ASE frequency dynamics observed in this study, using a CSE mesoscale description. This further substantiates our real-world empirical observations. In particular, we find that key aspects of the CSE's mesoscale dynamics are principally accessible via the (N)BLE frequency model. The macroscopic (N)BLE noise levels follow the absolute noise level variations of the mesoscopic CSE model during the pre-outage period. Interestingly, the observed decoupling of BLE and NBLE drift slope estimates in the example of the Keeler-Allston line (cf. Fig. S13) is reproduced by including red instead of white noise in the model. Changing the deterministic line capacities of the CSE to model the Keeler-Allston line THIF is partially reflected in the (N)BLE drift slope of the ASE, as well as in the (N)BLE noise levels. In such cases, the noise level appears to quantify how rapidly noise propagates through the network and how strongly it enters into the dynamics of the nodes due to the coupling strength. The rate of change of the CSE parameters can influence the response of the drift slope. This is likely attributable to the network's intrinsic time scale for relaxing into a fully synchronised state after a disturbance. In this way, the abrupt loss of the McNary units is associated with a substantial increase in the drift slope. Reducing the damping in the CSE McNary model aligns with our empirical findings of increasing drift slopes and noise levels at the very end of the pre-outage interval. This idea is supported by the approved disturbance report[57] (cf. Data Availability section for access details), which states that several damping tools were not functioning at The Dalles and John Day hydroelectric power units due to control issues. This caused increasing frequency oscillations, resulting in the final grid separations. It is a prototypical example of a scenario similar to that presented in Supplementary Box S2. Not least, the relative influence of mesoscale CSE noise on the macroscopic ASE frequency noise diminishes as the grid approaches a B-tipping destabilization due to extreme power demands or insufficient line capacities, whereas the B-tipping itself continues to be characterised by drift slopes that approach zero (cf. Fig. S4).

**Grid restoration interval.** The second bus voltage frequency time series $\omega_R(t)$ was originally extracted from four DIN A4 pages of an analogue, printed frequency graph in the approved disturbance report[57] using image processing software comparable to DigitSeis[60]. The printed graph, originally provided by BPA for the disturbance report's task force, can be found in Exhibit 10 of the disturbance document[57]. In refs. 20,49, where the digitised version of the frequency scan first appeared, no absolute time stamps were reported; the studies used only relative times. After roughly 1.5 years of research and correspondence (cf. SI S11) to check the analogue source of the digital time series $\omega_R(t)$, there are compelling reasons to build the analysis on a time axis that differs significantly from the originally reported one.

Based on the reasons stated in SI S11, it can be concluded that the investigated bus voltage frequency time series $\omega_R(t)$ mostly covers the time after the grid separation in four islands, incorporates roughly 20 h with an approximate time step resolution of $\Delta t = 6.41$ s, and starts at 15:10:45 with an uncertainty of $\sigma_{t_s} = \pm 3$ min. This uncertainty, which is in the range of a few minutes, does not affect the conclusions drawn on a scale of hours. Additional details regarding the approximate reconstruction of the correct absolute time stamps, time interval, and the time step $\Delta t$ can be found in SI S12.

Similar to the pre-outage time series $\omega_P(t)$, the original frequency time series $\omega_R(t)$ is detrended by a slow trend version obtained through Gaussian kernel smoothing with the bandwidth $\sigma_k = 10.68$ min. This process results in the detrended frequency time series $\bar{\omega}_R(t)$, which is shown in Fig. 1e. For the analysis of the detrended time series $\bar{\omega}_R(t)$, time windows of $1 \cdot 10^3$ data points (i.e. 1.78 h) with a shift of 100 points (i.e. 10.68 min) are used.

Following the interpretation of the metadata from SI S12, the black-hatched time interval in Fig. 1e–g encompasses only pre-outage data. The end of the black-hatched interval coincides with the end of the high-resolution frequency data in Fig. 1b at 15:48:54.95. The emergence of the first three islands aligns with the first sawtooth-shaped peak. Roughly 6 min later, the final system separation state was reached when Alberta segregated from the Northern island. The separation of the Alberta island falls together with the second sawtooth-shaped peak, which is somewhat difficult to discern due to the time scale resolution.

The Alberta island was the last area to insulate from the NAWI, but it was also the first to re-synchronise with the Northern island. The full load in the Alberta area was already restored by 17:39, which, at first glance, seems to coincide with the zero-crossing of the drift slope estimates $\hat{\zeta}_{\text{NBLE}}$ and the rapid transient to stable noise level estimates $\hat{\Psi}$ in Fig. 1e, f, respectively. However, the estimates are not trustworthy, as the first windows incorporate both pre-outage and post-outage data. The zero-crossing of the drift slope estimates coincides with the exclusion of the initial dip and pronounced first sawtooth-shaped peak around 15:48. In this context, the zero-crossing is sensitive to the rolling window size due to the non-stationary nature of the data (i.e. mixed pre-outage and post-outage data) and does not reflect the real-world event of load restoration in Alberta after its re-synchronisation.

Only shortly after 21:42, when the full load was restored by recovering the Metropolitan Water District of Southern California, the drift slope estimates $\hat{\zeta}_{\text{NBLE}}$ indicate further stabilisation, exhibiting a negative trend towards a stable plateau around $\hat{\zeta}_{\text{NBLE}} \approx 2.3 \cdot 10^{-2}$ in the green-shaded time interval. During this interval, the last customers were reconnected to the grid by 01:00:00[61]. This is in good agreement with the theoretical model considerations, as over this time, the grid increasingly resembles the physical pre-outage state. Similarly, the rapid noise level transient prior to the orange-shaded key restoration interval slows down within the orange-shaded interval, eventually reaching a stable plateau in the green-shaded period. The complete restoration of the NAWI took several days, as mentioned previously.

The scaling differences between the NBLE estimates, derived from the frequency time series $\tilde{\omega}_P(t)$ and $\tilde{\omega}_R(t)$ in Fig. 1b–d, e–g, respectively, are at least partly attributable to the significantly different time resolution and quality of the data. The discussion concludes with a few remarks on this matter. In principle, generating the digitised frequency time series $\omega_R(t)$ of the restoration interval from a printed graph via an image processing tool yields reasonably approximated data for analysis. However, some outliers exist in the extracted raw data. The two most prominent outliers are shown as orange lines in the insets of Fig. 1d, along with the corrected raw data in blue. The effect of these outliers is demonstrated by the grey drift slopes $\hat{\zeta}_{NBLE}$ and noise level estimates $\hat{\Psi}$ in Fig. 1f, g, respectively. Essentially, they were concomitant with discontinuous jumps in the estimates to plateaus of approximately one window length. With access to a scan of the original printed frequency time series, these outliers are identified as artefacts of the data generation procedure. Specifically, the image processing algorithm that extracted the digitised time series from the printed version misinterpreted grid lines of the millimetre paper to be actual data points (cf. SI S18, Fig. S11 for more details). In light of this, the discontinuous jumps in the estimates make sense, as the outliers are completely independent of the assumed stationary data distributions in the individual windows. The unbiased results shown in Fig. 1f, g are accordingly computed on a corrected version of the frequency time series, represented in blue in Fig. 1e. Briefly, this correction consists of using a thinned version of the data, in which the two most prominent outliers are systematically replaced by random values. This approach diminishes their influence without making assumptions about their underlying frequency dynamics. See Methods for more details regarding the outlier correction procedure. As a result, the estimates from the outlier-corrected dataset exhibit only a weak increase with broader credible intervals (CIs) over one window length in the corrected intervals. This is an expected result, as the correction values were drawn from a distribution estimated in a symmetric $\epsilon$-environment around the outlier sequence. Therefore, they correct for the strong outlier magnitude; however, the increments between the random values do not resemble the actual frequency dynamics. In nine windows, broad CIs with skewed drift slope distributions are observed. These are most likely caused by the differing database of individual rolling windows. Some of them may include further outliers or extreme values that disrupt the NBLE and cause broader CIs. All in all, the power grid most likely remained in a similarly resilient state from 01:00 until 11:00.

## Discussion

We demonstrated the (N)BLE's potential to quantify varying restoring rates (as indicators of deterministic stability) and noise levels (capturing stochastic influences) related to B- and N-tipping, respectively, from time series data. The method significantly outperforms common non-parametric and parametric leading indicators, such as AR1, STD, direct drift-diffusion estimation, and the OUE (cf. SI S1 and S9) for synthetic and real-world datasets and is applicable across various disciplines (cf. SI S10). Additionally, we extend the BLE to the NBLE, which cancels the BLE bias observed for the non-Markovian synthetic examples. Incorporating memory effects and correlated noise, as in the NBLE, is fundamentally out of reach for the aforementioned state-of-the-art leading indicators.

Two bus voltage frequency time series, spanning the pre-outage to post-outage period of the NAWI cascading failure on 10th/11th August 1996, are analysed using the NBLE parameterisation, since it significantly improves the statistical model of frequency dynamics during the NAWI outage (cf. SI S15).

A detailed comparison of the NBLE results with the real timeline of events supports the presented theoretical ideas about macroscopic power grid frequency dynamics and the Langevin model to a large extent (cf. SI S6). The results disprove the simple idea of smooth changes in early warning metrics based on CSD for B-tipping[20,21,48,49],

even though the basic assumptions are still supported. Rather than observing continuous changes in frequency dynamics, the analysis reveals rapid variations into states of modified local stability and stochastic influences. These variations are closely interrelated to abrupt real events, such as line openings, the loss of power units, or failures of other grid components. These state changes are almost discontinuous in nature. However, the alternating states are reflected in alternating NBLE results. These findings align with the common intuition surrounding cascading failures, which are typically perceived as not continuous but rather as discrete sequences of destabilising events. Nevertheless, the analysed example demonstrates that these discrete changes can lead to operating conditions that indeed trigger more continuous destabilisation—specifically, a lack of damping options that favoured increasing frequency oscillations for ~1 min prior to the initial formation of grid islands. Even though a skilled power system controller might already extract a considerable amount of information about changes in the grid state from the raw frequency time series, the distinction of fast and slow dynamics, as well as the persistence of changes in the (N)BLE results, adds valuable information. In particular, the (N)BLE identifies a significant and persistent change in the frequency dynamics ~2 min before the outage's key triggering event, namely the 500 kV Keeler-Allston line tripping. While a trained system controller might have noticed a short dip in frequency at the time suggested by the (N)BLE, this dip does not provide information about the persistence of the system's state change. In the considered outage, the newly reached stable constant values persist for roughly 6 min, until the loss of the McNary units changes the state again. Moreover, the theoretical foundation of applying the (N)BLE to power grid frequency data is elaborated in SI S3. In stable operation, the frequency dynamics can be aggregated from the CSE on mesoscopic scales to the macroscopic scale of the ASE. The ASE represents an OU model, i.e. a Langevin-type equation. In this sense, using the (N)BLE, we directly estimate the macroscopic frequency dynamics of the ASE. Moreover, the formal relationship between the one-dimensional macroscopic ASE and the high-dimensional CSE, resolving network topologies on mesoscopic scales, allows for improved inference from the macroscopic (N)BLE to the mesoscopic scales of the power network. This partially closes the gap (cf. Supplementary Box S1) between the macroscopic ASE estimation and lower topological scales.

In this spirit, a simple CSE reconstruction is provided for three key events of the NAWI cascading failure. The simple model qualitatively reproduces (N)BLE signatures of the real-world scenarios, further substantiating our empirical findings and deepening our understanding of the NAWI outage event in relation to the response of the (N)BLE metrics. This simple topological model of the NAWI power outage serves as a starting point for further discussions, fostering a better understanding of the relationship between macroscopic frequency dynamics of the ASE and topological features of the CSE scale, and may be improved in future.

The overall modelling results leave several intriguing questions for future research: Notably, a clear direct (N)BLE response to actual line tripping events is not observed. However, in our model, cutting a line removes inertia from the grid, suggesting that considering such cases systematically in future studies could be beneficial. Moreover, the model results suggest that local disturbances are predominantly detected by the (N)BLE in signals from nodes located in the disturbance's immediate periphery. A more detailed consideration of these topological features is left for future investigations.

While this analysis focuses on frequencies as a key observable, other macroscopic quantities of power grids may also be suitable for the (N)BLE. For instance, the phase of the CSE represents an interesting candidate. The parameterisation of the (N)BLE could be extended to higher dimensions, potentially taking advantage of high-dimensional time series input. An ambitious approach could involve using frequency and phase time series of various grid locations in a CSE-like

high-dimensional parameterisation of the (N)BLE. Not least, the generality of the results should be tested on other real-world data or signals from network models with more complex topologies and alternative destabilisation scenarios.

The BLE is provided in the open-source Python package *antiCPy*[62,63]. Application to other fields and systems could improve the understanding of its scope and limitations. It lends itself for adaptations of the drift-diffusion parameterisation based on partial system knowledge and specific established models to improve the statistical inference.

Furthermore, we are considering the inclusion of prior information about multi-stability, specifically by restricting the parameter space to combinations that produce a double-well potential. This approach could potentially allow us to apply the formalism of Kramers' escape rate estimation[64–66] without needing data from both minima of the double-well potential. Assuming a priori that the system is bistable might enable us to extract the potential barrier height, necessary for computing the Kramers' rate, provided the noise is strong enough to resolve the inflection point of only one of the potential valleys.

## Methods
In "Estimation schemes", the basic and extended numerical procedures are presented in "Bayesian Langevin estimation" and "Non-Markovian Bayesian Langevin estimation", respectively. The outlier treatment is summarised in "Outlier correction in the post-outage interval".

### Estimation schemes
Key steps of the estimation protocols detailed below are illustrated in an algorithmic scheme in Box 1.

**Bayesian Langevin estimation.** The Bayesian Langevin estimation (BLE) procedure[37,38] is applied using rolling windows over the time series (cf. Box 1). Similar to the idea of Carpenter and Brock[12], the observed signal in each window is modelled by a stochastic differential Langevin equation. In general, the proposed modelling approach is not limited to one-dimensional time series. However, since we consider one-dimensional signals in the following, we reduce the general approach to the one-dimensional Langevin equation

$$\dot{x}(x,t) = h(x(t),t) + g(x(t),t) \cdot \Gamma(t). \tag{7}$$

This method implemented for the one-dimensional case can, in principle, be extended to address $N$-dimensional time series analysis problems. The noise $\Gamma(t)$ is assumed to be Gaussian and $\delta$-correlated, and does not significantly depend on the state $x(t)$, i.e. we can set $g(x(t),t) = \text{const.} = \sigma$ in each window. The stochastic process resembles the increments of a Wiener process, i.e. $\Gamma(t) = \frac{dW}{dt}$.

A change of the sign $(- \to +)$ in the slope

$$\zeta = \left.\frac{dh(x)}{dx}\right|_{x=x^*} \tag{8}$$

of the nonlinear drift at the fixed point $x^*$ indicates destabilisation of this fixed point through control parameter change and thus a bifurcation. The fixed point $x^*$ is estimated as mean in each window.

We develop $h(x, t)$ into a third-order Taylor series, which is sufficient to describe the normal forms of simple bifurcation scenarios[67]. Furthermore, in cases of strong noise, the first approximation of small disturbances, i.e. $\mathcal{O}((x - x^*)^2) \approx 0$ breaks down and the BLE is more reliable due to the higher-order Taylor expansion[38]. This results in

$$h(x(t),t) = \alpha_0(t) + \alpha_1(t) \cdot (x - x^*) + \alpha_2(t) \cdot (x - x^*)^2 + \alpha_3(t) \cdot (x - x^*)^3$$
$$+ \mathcal{O}((x - x^*)^4) \tag{9}$$

so that the information on the linear stability is incorporated in $\alpha_1$, assuming the fixed point is shifted to zero. For practical reasons, in the numerical approach, Eq. (9) is used in the form

$$h_{\mathrm{MC}}(x(t),t) = \theta_0(t;x^*) + \theta_1(t;x^*) \cdot x + \theta_2(t;x^*) \cdot x^2 + \theta_3(t;x^*) \cdot x^3, \tag{10}$$

where an arbitrary fixed point $x^*$ is incorporated in the coefficients $\underline{\theta}$ by algebraic transformation and comparison of coefficients. As we use the stationary window approximation, it holds that $h_{\mathrm{MC}}(x(t), t) \equiv h_{\mathrm{MC}}(x)$ within each window. In other words, the time dependence is translated into the subsequent shift of the window. The estimation of the model parameters $\theta_i$ and $\sigma$ is realised via a Markov Chain Monte Carlo (MCMC) method to reconstruct the full posterior distribution of the drift slope $\zeta$ and the noise level $\sigma$. The starting point is Bayes' theorem

$$p(\underline{\theta}|\underline{x}, \mathscr{I}) = \frac{p(\underline{x}|\underline{\theta}, \mathscr{I}) \cdot p(\mathscr{M}|\mathscr{I})}{p(\underline{\theta}|\mathscr{I})} \tag{11}$$

with the posterior probability density function (PDF) $p(\underline{\theta}|\underline{x}, \mathscr{I})$, the likelihood $p(\underline{x}|\underline{\theta}, \mathscr{I})$, the prior $p(\mathscr{M}|\mathscr{I})$, and the evidence $p(\underline{\theta}|\mathscr{I})$, which accounts for the normalisation of the posterior. The model parameters are denoted by $\underline{\theta}$, the time series data by $\underline{x}$, the background information by $\mathscr{I}$, and the model by $\mathscr{M}$.

The short-term propagator[68]

$$p(x, t = t' + \Delta t | x', t') = \frac{1}{\sqrt{2\pi g^2(x', t')\Delta t}} \exp\left(-\frac{[x - x' - h(x', t')\Delta t]^2}{2g^2(x', t')\Delta t}\right) \tag{12}$$

for subsequent times $t$ and $t'$ with $\tau = t - t' \longrightarrow 0$ can be derived from the Langevin equation if the difference $x - x'$ in the exponential expression is approximately defined by the first differences of a given time series. It represents the likelihood. The priors are chosen to reflect the situation of no or just poor prior information. This guarantees the determination of the posterior mainly due to the available data instead of strong prior assumptions. More restrictive priors would be ill-advised in the subsequent analyses, since we have limited information about the model parameters describing the frequency dynamics of the NAWI. For this reason, we assume an invariant prior[69] for a straight line, i.e. for the intercept $\theta_0$ and the slope $\theta_1$. It is given by

$$p_{\mathrm{prior}}(\theta_0, \theta_1) = \frac{1}{2\pi(1 + \theta_1^2)^{\frac{3}{2}}} \tag{13}$$

with broad parameter ranges. Note that we denote $p_{\mathrm{prior}}(\theta_0, \theta_1)$ even if there is no explicit dependency on the intercept $\theta_0$. For equally likely straight-line models, the angle and distance parameters in the Hesse normal form are assumed to be uniformly distributed. The prior in Eq. (13) then results from the transformation into Cartesian coordinates, which implicitly couples slope and intercept (cf. refs. 69–71). For the noise level $\sigma$, the invariant Jeffreys' scale prior[69]

$$p_{\mathrm{prior}}(\sigma) = \frac{1}{\sigma} \tag{14}$$

is chosen because it is almost uninformative. Furthermore, by using broad Gaussian priors

$$p_{\mathrm{prior}}(\theta_2) = \mathscr{N}(\mu = 0, \sigma_{\theta_2} = 4) \tag{15}$$

$$p_{\mathrm{prior}}(\theta_3) = \mathscr{N}(\mu = 0, \sigma_{\theta_3} = 8) \tag{16}$$

with mean $\mu$ and standard deviations $\sigma_{\theta_i}$, we ensure that the higher-order parameters can initially contribute to the deterministic

dynamics with a magnitude similar to that of the linear ones. The MCMC affine-invariant ensemble sampler of the *emcee*[72] Python package is used to compute the posterior PDF. Based on the estimated joint posterior PDF $p(\underline{\theta}|\underline{x}, \mathscr{I})$, the parameters $\underline{\theta}$ are sampled and corresponding drift slopes $\zeta$ in the fixed point $x^*$ are calculated by marginalisation:

$$p(\zeta|\underline{x}, \mathscr{I}) = \int p(\underline{\theta}, \sigma|\underline{x}, \mathscr{I})\delta\left(\zeta - \frac{\mathrm{d}h(x)}{\mathrm{d}x}\bigg|_{x=x^*}\right)\mathrm{d}\underline{\theta}\mathrm{d}\sigma. \quad (17)$$

The credible intervals (CIs) of the slopes and noise levels are defined as the 16% to 84% and 1% to 99% percentiles of the corresponding posterior PDFs. These are computed from kernel density estimates of these PDFs. The kernel density estimation is performed with *scipy. stats. gaussian_kde*[73] using Silverman's rule of thumb to determine the kernel bandwidth. More details on the specific prior ranges used for the analyses are provided in SI S19, Table S5.

**Non-Markovian Bayesian Langevin estimation.** The non-Markovian Bayesian Langevin estimation[74] (NBLE) builds on the previously sketched BLE (cf. Box 1). Inspired by Willers and Kamps[75], it involves a two-dimensional model

$$\dot{x} = h_x(x) + g_x(x) \cdot y \qquad \dot{y} = h_y(y) + g_y \cdot \Gamma(t) = -\frac{1}{\theta_5^2}y + \frac{1}{\theta_5} \cdot \Gamma(t), \quad (18)$$

with an observed Langevin-like process $x$ and a hidden OU process $y$. The hidden process may be interpreted as red noise $y(t) = \Gamma_{\mathrm{red}}(t)$ if it lives on a faster time scale $\tilde{\tau}_y$ than the observed process $x$, or, alternatively, as a slow external driver $y$ of the observed process $x$, which then evolves on the faster time scale $\tilde{\tau}_x$. Note that similar models are also treated under the name *hidden Markov models*[75]. Under the assumption of a hidden OU process, the likelihood of the two-dimensional system is adapted accordingly to be

$$p(x, t|x', x'', t', t'') = \frac{1}{\sqrt{2\pi\psi^2(x', x'')\Delta t}}\exp\left(-\frac{(x - x' - \varphi(x', x'')\Delta t)^2}{2\psi(x', x'')^2\Delta t}\right),$$

with primes and double primes denoting values one and two steps $\Delta t$ past in time. The non-Markovian drift is given as

$$\varphi(x', x'') = h_x(x') + g_x(x')y'' + h_y(y'')g_x(x')\Delta t \quad \text{with}$$
$$y''(x', x'') = \frac{x' - x'' - h_x(x'')\Delta t}{g_x(x'')\Delta t}. \quad (19)$$

The non-Markovian analogue to the constant noise level $g(x) = \text{const.} \equiv \sigma$ of the Langevin equation is given by the composite noise level[75]

$$\psi(x', x'') = g_x(x') \cdot g_y(y'') \cdot \Delta t. \quad (20)$$

The parameterisations of drift $h_x(x)$ and diffusion $g_x(x)$ remain unchanged compared to the one-dimensional BLE parameterisation, but the diffusion $g_x(x)$ serves as coupling function. An invariant prior of a straight line for the drift $h_y(y)$ and a scale prior for the constant diffusion $g_y \equiv \frac{1}{\theta_5}$ are multiplied:

$$p_{\mathrm{prior}}(\theta_5) = \frac{\theta_5}{2\pi\left(1 + \left(-\frac{1}{\theta_5}\right)^2\right)^{\frac{3}{2}}}. \quad (21)$$

Furthermore, we account for stability issues and ambiguities in the estimation scheme, discussed by Willers and Kamps[75], in two ways: First, analogous to Willers and Kamps[75], by including only one free

parameter $\theta_5$ in the OU process $y$, and second, by an a priori assumption of a notable time scale separation between the processes $x$ and $y$ with the time scales $\tilde{\tau}_x$ and $\tilde{\tau}_y$, respectively. Precisely, we require either $\tilde{\tau}_x > \gamma \cdot \tilde{\tau}_y$ or $\tilde{\tau}_y > \gamma \cdot \tilde{\tau}_x$ with a scale separation coefficient $\gamma = 2$. We denote characteristic time scales by $\tilde{\tau}$ to distinguish them from discrete time lags $\tau$ (without a tilde, as used in the SI). The characteristic time scales[67] are approximated by

$$\tilde{\tau}_\vartheta = \left|\left(\frac{\mathrm{d}h_\vartheta(\vartheta, t)}{\mathrm{d}\vartheta}\right)^{-1}\right|\bigg|_{\vartheta = \vartheta^*} \qquad \text{with } \vartheta \in \{x, y\}. \quad (22)$$

The priors for the model of the observed data $x(t)$ remain unchanged apart from the coupling term $g_x(x) = \text{const.} \equiv \theta_4$ for which we use a Gaussian prior, analogous to Eq. (15) with $\sigma_{\theta_4} = 4$. More details on the prior ranges used for the analyses are provided in SI S19 and Table S5.

**Outlier correction in the post-outage interval**
The time series $\omega_R(t)$ was carefully compared to its printed source in SI S18 and Fig. S11. The comparison revealed that the outliers were erroneously generated by the image processing tool, which misinterpreted grid lines of the scale paper as actual data points. For this reason, we first thinned the original time series by a factor of two, which already eliminated four out of six prominent outliers. Second, we corrected for the remaining two prominent outlier regions by using a heuristic approach. The detected outlier intervals, the first and second of which contain four and two points, respectively, were replaced with a sequence of random numbers in a range derived from the data observed before and after the intervals. Therefore, we considered the non-outlier samples in a symmetric $\epsilon$-environment of roughly 2 min centred around the outlier peaks. The peaks are subjectively defined by thresholds $T_{\mathrm{out}} = 0.01$ and $T_{\mathrm{out}} = 0.04$ for the first and second outlier intervals, respectively. The data $\omega \in \epsilon$ are used to construct Gaussian kernel density estimates with Silverman's rule. The correction values are drawn from the resulting probability densities. The approach is chosen to avoid prior assumptions about the frequency dynamics under analysis. In light of this, the corrections account for the magnitude of the outliers, but cannot reconstruct information about the underlying dynamics.

## Data availability
The simulated and empirical data and SI can be found in the folder "Quantifying_Local_Stability_and_Noise_Levels" of the GitHub repository[76] at https://github.com/MartinHessler/Disentangling_Tipping_Types. Both versions of the restoration time interval's frequency data $\omega_R(t)$, i.e. the densely sampled and the thinned one, are included. The additional data of a bacteria population collapse, analysed in SI S10, are openly available at https://lifesciences.datastations.nl/dataset.xhtml?persistentId=doi:10.17026/dans-ztg-93aw. Regarding the approved disturbance report[57], the text source without Appendices was provided by Bonneville Power Administration (BPA) via a Freedom of Information Act (FOIA) request thanks to James King (FOIA Public Liaison, BPA; P.O. Box 3621, CGI-7, Portland, OR 97208-3621; phone: 503-230-7621; email: FOIA@bpa.gov) and Brian Roth (FOIA Case Coordinator, BPA). The Appendices 2, 3, 5, and 9 of the timeline were provided thanks to Mary Schaff from the Washington State Library, currently operating under the Secretary of State, Steve Hobbs, through mail correspondence via the "Ask a Librarian" service (Washington State Library, Point Plaza East, 6880 Capitol Blvd. SE, Tumwater, PO Box 42460, Olympia WA 98504-2460; phone: (360) 704-5200; email: askalibrarian@sos.wa.gov). A low-quality scan of the restoration frequency time series from the approved report's Exhibit 10 was provided by WECC, and a readable version from the preliminary report[77] was made available thanks to Jeanie Fisher from the Seattle Municipal Archives (Seattle Municipal Archives, 600 Fourth Avenue,

Third Floor, Seattle, WA, 98104, PO Box 94728, Seattle, WA, 98124-4728; phone: (206) 684-8353; email: archives@seattle.gov). The full preliminary report[77] is available at the Seattle Municipal Archives. Contact with Jeanie Fisher of the Seattle Municipal Archives was established by Mary Schaff. Source data are provided with this paper.

## Code availability

The open-source Python package *antiCPy*[62,63] to perform the (N)BLE can be found at https://github.com/MartinHessler/antiCPy under a *GNU General Public License v3.0* and is documented at https://anticpy.readthedocs.io. A second GitHub repository[76] includes computation codes and plot scripts for the analyses presented in the main article and SI, published under a *GNU General Public License v3.0*.

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

## Acknowledgements

M.H. thanks James King (FOIA Public Liaison, BPA), Brian Roth (FOIA Case Coordinator, BPA), Mary Schaff (Librarian, Washington State Library), Jeanie Fisher (Reference Archivist, Seattle Municipal Archives), WECC, Eduardo Cotilla-Sanchez (Professor, Oregon State University), the anonymous employees of BPA and all the other people who took part in finding the metadata of the two frequency time series for their gentle, cooperative and very dedicated work (cf. SI S11). M.H. thanks Eduardo Cotilla-Sanchez for providing the densely sampled version of the restoration time interval data $\omega_R(t)$ and for fruitful discussions about the missing metadata. M.H. thanks the Studienstiftung des deutschen Volkes for a scholarship including financial support. O.K. thanks Frank Ehebrecht for co-working to gain the thinned restoration interval dataset $\omega_R(t)$ during previous projects (under the old time scale assumptions). We thank colleagues and friends for proofreading the manuscript.

## Author contributions

M.H. and O.K. have designed research. M.H. performed research, analysed data, designed graphics and modelling thoughts, wrote the manuscript, created and maintains contents of the supplementary GitHub repository, and searched for the frequency time series metadata. O.K. provided thinned restoration interval data $\omega_R(t)$ from previous works.

## Funding

## Competing interests

The authors declare no competing interests.
