## [Transparent Peer Review file · Nature Communications]

Quantifying Local Stability and Noise Levels From Time Series in the US Western Interconnection Blackout on 10th August 1996

Corresponding Author: Mr Martin Heßler

Version 0:

Reviewer comments:

Reviewer #1

(Remarks to the Author)

What are the noteworthy results?

The manuscript proposes an Early-Warning Indicator (EWI) and applies it to test examples and a times series from historic frequency data around a power grid blackout 1996.

I see the proposal & testing of the new method, a Bayesian estimator for a generic Langevin model (called BL estimation in the paper), as the main noteworthy novelty of the manuscript. Proposing and demonstrating feasibility of a computational/statistical method may not count as a result or discovery in the sense considered by Nature's science journals. However, the authors claim that the BL estimate is a significant improvement of classical EWIs (such as time-step aurocorrelation, variance) that were introduced in high-profile articles in Nature, PNAS, Science etc (refs 1,7,9,17,18,54) and are repeatedly and recently discussed there (see [R1,2023] for a striking recent example in Nat. Comm.).

Hence, a strong argument can be made that a significant improvement of EWI estimation should be published to the same audience. In my view the proposed BL estimate is certainly a methodological improvement over the above-mentioned classical EWIs. However, the current version does not convince me yet if the improvement is significant.

[R1] Ditlevsen, P., & Ditlevsen, S. (2023). Warning of a forthcoming collapse of the Atlantic meridional overturning circulation. Nature Communications, 14(1), 1-12.]

Will the work be of significance to the field and related fields? How does it compare to the established literature? If the work is not original, please provide relevant references.

The method as proposed would have widespread relevance for providing evidence of potential tipping from observations in climate science, ecology, social science and complex engineering systems.

Regarding originality: the abstract uses the word "propose a BL approach", leaving open how original this is. The description of the method is only brief and comes at the very end. It refers to [12] (however, [12] does not employ a Bayesian approach). The numerical ingredients of the BL estimate (estimating probability distributions sampled using MCMC for coefficients of a polynomial model with additive Gaussian noise) may not be new, and in the article the authors do not claim that this part is novel. However, introducing a rigorous statistical method into science has value.

Does the work support the conclusions and claims, or is additional evidence needed?

There are severe problems with the presentation: the paper spends space introducing and discussing tipping, has a detailed introduction of the test examples, followed by an extensive discussion of the behaviour of the BL estimate on these test examples, then a minute description of events and data at the 1996 blackout (using power grid specific jargon), without ever

describing the BL estimation proposed. The term is used throughout the paper without explanation or context until the brief description in "Methods".

The description in "method" makes various ad-hoc choices that are not discussed (the supplementary material would have been the place to do this): for example (and apart from the Bayesian priors), it includes 2 higher-order expansion terms of the right-hand side beyond the linear term, while only the linear term is used. The linear term is referred to as ζ in all figures but as θ_1 in the method. A Bayesian approach iteratively incorporates additional data to update the current belief about the probability distribution of an unknown. The paper does not describe what is iterated over. Is it time? That is, are the priors assumed to be the coefficients at the earliest time and then the measured time series points for increasing times are "fed into" Bayes' theorem to update the distribution, such that we obtain time dependent coefficients simply because of the iteration process? Or do we have a prior distribution of time-dependent coefficient functions $\theta_i(t)$ and then we apply all time series data points at once? This makes a difference: in the former approach the estimates at the beginning of the times series are more strongly influenced by the prior, while in the latter approach all estimates are influenced equally.

The power grid example, also does not fully support the claim of prediction of tipping. The narrative suggests that the BL estimates \gg detect \ll an event that occurred at 15:40 (start of first red interval in figure 5). If that change of $\hat{\zeta}$ and $\hat{\sigma}$ had any predictive significance for the following spike at 15:42 is less clear to me (the authors claim that). However, a similar several subsequent sharp change of $\hat{\zeta}$ and increase of $\hat{\sigma}$ at 15:43 did not lead to a spike.

The strongly increased variance of the frequency time series pre-outage (std increased by a factor of >10), compared to the apparently standard variance (see post-outage) is a strong EWI on its own.

* Are there any flaws in the data analysis, interpretation and conclusions? Do these prohibit publication or require revision?

The authors refer to bifurcations for systems with noise, even though it is not clear in which sense they mean this. For example, for their test example Eq(2), Crauel [R2] showed that additive noise destroys the pitchfork bifurcation: one has a single globally stable attractor for all parameters.

[R2] Crauel, Hans, and Franco Flandoli. "Additive noise destroys a pitchfork bifurcation." Journal of Dynamics and Differential Equations 10 (1998): 259-274.

The introduction spends time expounding on the importance of distinguishing between a change in decay rate in the "deterministic part" of the (assumed-to-exist) Langevin model and a change of the noise amplitude. However, this distinction is artificial as every change in the noise amplitude can be transformed into a change of the deterministic part (as long as the noise amplitude does not go to 0, see [R3]).

[R3] <https://math.stackexchange.com/questions/4673542/transformation-of-ito-diffusion-such-that-the-diffusion-coefficient-becomes-cons>

The argument that autocorrelation and variance of the time series do not permit to conclude separately about input noise amplitude σ^2 and decay rate $-\zeta$ is also not quite correct. Assuming that in each sliding window the time series is approximately governed by an Ornstein-Uhlenbeck process, the one-step autocorrelation AR1 determines ζ through $\zeta = (\log AR1) / \Delta_t$ (Δ_t is the time step) and the variance V determines the σ^2 through $\sigma^2 = -2V (\log AR1) / \Delta_t$. The estimates $\hat{\rho}$ and $\tilde{\sigma}$ in Figure 3 look consistent with the above expressions. So, Figure 3 should plot the decay rate and noise amplitude implied by AR1 and V , and compare that to the quantities determined by the BL estimate. (They may still be less accurate, which would demonstrate superiority of the BL estimate.)

Is the methodology sound? Does the work meet the expected standards in your field?

The analysis methodology is sound as far as I could judge. The presentation is poor: the article is too long, spends too little space on the main method/result proposed in the abstract, and far too much on tangents (general tipping background, details of the power grid event).

Is there enough detail provided in the methods for the work to be reproduced?

The code for BL estimation and reproducing the results is available according to references. However, as the description is very short, one would have to apply the code in a black-box fashion without understanding what is happening.

Reviewer #2

(Remarks to the Author)
Please check the PDF attached.

Best regards,
Leonardo Rydin

Reviewer #3

(Remarks to the Author)

In this manuscript it is argued that statistical early warning signals (EWS) can be obtained from measurements of frequency fluctuations in the AC power net preceding a power disruption. It is argued that such a disruption can be modelled as a noise-induced tipping similar to standard transitions in stochastically driven bifurcating dynamical systems.

The paper is obscure, with a mixture of qualitative (inaccurate) descriptions of electric power grids, poor descriptions of the mathematics of tipping scenarios and specific description of the North American Interconnection power blackout in 1996.

I cannot recommend publication, and I cannot offer much advice guiding the authors in directions of revisions to make this a relevant paper for publication.

A few remarks:

Inforboxes 1,2: There is very obscure connection between the stochastic differential equation ("Langevin model") and the diagram below in (a). The cartoons with mountains and houses are really not helpful for describing the transitions in a potential. Pdf's in b.2 seems to be taken out of the blue and the time-independent potential $V(x)$ in the text is an integral of an explicitly time-dependent drift function $h(x(t),t)$.

Figure 3 shows (quite arbitrary) sets of simulations of pitchfork and fold bifurcation scenarios with constant and changing noise intensities. The EWS are calculated without mentioning of running windows, with "credibility bands", without specifying anything about the Bayesian-model part of the BL-estimation (BL meaning Bayesian-Langevin). It is very difficult to get any useful information from this figure.

The descriptions in Section 2.1 of the conceptual models is poor. Eqs. (5) and (8) does not describe noise intensities as shown in Figure 4 f, i.

Statements as in lines 250-252: "Note also that for monitoring uprising tipping risk factors the exact estimates are much less important than their relative time evolution which encodes the desired information." Is completely obscure to me, and this is just an example.

Other examples from the conclusions: "Nonetheless, the qualitative evolution of the stability and the noise level is also possible if the assumption is violated which is an important implication for many real world applications". (line 492) I simply cannot make sense of such a sentence.

I recommend that the authors, firstly, specify which bifurcating or non-bifurcating model is relevant for describing the observed power grid. (I am not an expert). Secondly, they should specify, which part of the fluctuations can be modelled as a drift and which part can be modelled as stochastic. (the h - and the g -functions). Finally, they should specify what is the Bayesian part of the analysis. (It seems to me that the "credibility bands" are mere statistical "confidence bands", which is also fine, but for a mathematically oriented paper to be relevant to the reader, it must be precise and correct).

I am sorry that I cannot be more positive this time.

Version 1:

Reviewer comments:

Reviewer #1

(Remarks to the Author)

The revision responds to several of my requests. In particular, the authors now describe their statistical method advertised in the abstract in more detail on p16 (second half) to p18. They also clarify what the Bayesian updating mechanism iterates over. They ignored my mathematical nitpickings (that one strictly speaking cannot have a bifurcation in these systems with Gaussian noise), but that is less relevant for NatComm.

The revision makes much clearer that the paper considers as its main contribution the dissemination and advertisement of (N)BLE to the scientific community. The discussion of the power grid outage, the Greenland Ice Sheet and the cyanobacteria light stress experiments by Veraart et al (the latter two added in the revision) serve merely as demonstration examples picked from the available pool of high-profile debates on tipping. The new title, now containing the two disparate examples, confirms this.

The proposed (N)BLE method is undoubtedly an improvement over the common indicators such as one-step autocorrelation (AR1) or variance (VAR) of the time series in terms of systematic time series analysis. (N)BLE fits the time series data to coefficients of a Langevin equation such that its outcome are probability distributions for coefficients in this equation, namely for the restoring rate "zeta" toward the equilibrium and the amplitude (std) "psi" of the internal or external fluctuations assumed to influence the dynamics in the Langevin equation.

The fitting procedure is Bayesian updating using Markov Chain Monte Carlo (MCMC) sampling. The only cause preventing the scientific community from using the proposed method (instead of AR1 and VAR) is lack of acceptance or awareness.

I have two major concerns, both leading me to recommend not publishing the article in Nature Communications.

(1) Being a methodological improvement over reporting of AR1 and VAR is a low bar to cross for a time series analysis method. The authors demonstrate that (N)BLE has (far) less bias than AR1 and VAR for synthetic data from examples with known ground truths. However, the examples in the main article are Langevin equations such that there is very little (or no) model bias.

Demonstrating less bias when fitting with a more complex model is in itself not a significant step toward "Quantifying tipping risks" as claimed in the title. At no point in the paper do the authors provide an estimate for the tipping risk. For "noise-induced tipping", which this paper focuses on, this estimate could be provided in the form of a rate (in the spirit of Kramers' escape rate): the probability of tipping per time unit.

So, the title is misleading: the paper proposes a method to estimate the restoration rate and the noise amplitude of an assumed Langevin model for a time series. The method has much less bias than widely used but basic indicators when tested on Langevin models. No tipping risk is quantified.

The proposed (N)BLE method still contains a large number of ad-hoc choices such as

- * degree of polynomial in Langevin equation
- * choice of Langevin equation as base model (or LE+OU for NBLE)
- * priors for polynomial coefficients enforcing an intercept of the right-hand side
- * fixed time scale separation between hidden OU process and observed process in NBLE.
- * window sizes (which have a severe effect on the results as reported in several graphs)

I am not sufficiently expert on statistical time series analysis to judge if the choices made by the authors improve the state of the art in this area (and not just AR1 and VAR estimates), and I am not sure if NatComm is the appropriate venue to evaluate improvements of statistical estimation methods.

(2) The writing style of the original article was criticised by all reviewers as too unfocused, loaded with unnecessary tangents. The revision has improved some aspects, but made other aspects worse.

For example, the revision now advertises a different method than the original one. Now the method fits a model with a non-Markovian term. It also includes another demonstration using the Greenland Ice sheet melt rates, requiring a new round of background explanation. The paper now introduces and discusses the distinction between non-parametric (qualitative) and parametric estimations. The authors call, for example, the AR1 autocorrelation estimate non-parametric, even though (by its definition) the estimate models the time series's underlying process as an AR-1 process and estimates the one-step memory coefficient. So, the model is just simpler than the Langevin model fitted by (N)BLE and the distinction is artificial.

These changes in focus and scope confirm that the aims of the paper are still evolving rapidly and are not yet in a state settled enough to be reviewed with confidence.

Other presentation deficiencies are still present:

- * The text spends space introducing tipping rather poorly: Infobox (1) is full of details that play no role in the analysis. It contains metaphorical landscapes symbolising a potential, but what forms the barrier for tipping (the metaphorical hill in (b,d)) is never explained: what is the hypothesized positive feedback effect that decreases the restoring force for larger deviations? The presumed goal of the paper is "quantifying tipping risks", so the height, shape and location of this barrier needs to be estimated. However, the corresponding quantities are not discussed at all anywhere. Instead the pictures distract the reader with little suns (?) in the landscape. The discussion admits that the power grid failure cascade was triggered by a few discrete events, resulting in discontinuous changes for which estimates of restoring forces and Gaussian Noise amplitudes are inappropriate.
- * Axis scaling in graphs is different between closely related figures. For example, in Figure 1 the estimates for ζ_{NBLE} differ by three orders of magnitude ($\zeta_{\text{pre}} \sim -3$ vs $\zeta_{\text{post}} \sim -2 \times 10^{-3}$) between c (pre-) and f (post-outage), contradicting the text which states that ζ returns to pre-outage levels.
- * The overly extensive description of the power outage event sequence does not advance the main message of the paper. This was criticized already in the original version.
- * The interpretation of the graphs obtained in Fig1 and Fig3 is vague when translating the results into discussions of tipping risk. This is natural because the quantities obtained are not sufficient to estimate tipping risk, but it leads to hard-to-follow speculative writing.

(Remarks on code availability)

The code is well documented and not overly complex. I have no concerns that it works to the specification given in the methods section of the paper.

Version 2:

Reviewer comments:

Reviewer #1

(Remarks to the Author)

The most recent version of the manuscript is in my view suitable for publication if the authors address a few final points, listed below. The editor may decide whether the points have been addressed in a subsequent revision (but I am happy to have another look if needed).

My main reason for supporting publication for the revision (changing from previous recommendations) is that the presentation has now much better focus on the main contribution with fewer distractions than previous revisions. In my view the main contribution is the demonstration of the (N)BLE estimators for first-order resilience, often used as "Early-Warning Indicators" (EWIs) for tipping. It is instructive to see how including higher-order terms in the estimator improves the robustness of the first-order terms (not only in synthetic examples but also in credible real-world examples).

The contribution is mostly a methodological advance. Nat. Comm. is not the ideal venue for proposing and demonstrating methods. However, the methodology is an obvious improvement over the EWIs used frequently in high-profile general-interest science journals to discuss imminence of tipping, such that a case for publication in Nat. Comm. exists.

I am not convinced that the real-world example (the NAWI power outage in 1996) is a strong case for using these EWIs: the time series before the outage has highly coherent periodic oscillations (at about 0.2-0.3Hz) with an amplitude 10 times larger than post-outage. The authors admit in the SI that their methods are not designed to capture such periodicity, resulting in large deviations from the known ground truth in the resulting autocorrelation estimates (Fig S12 left). In addition to these oscillations, large spikes corresponding to a tree-related fault of the power line serve as clear problem-specific EWIs, as the paper's discussion reveals. However, the paper demonstrates that the (N)BLEs behave consistent with theoretical expectations in such cases.

The following remaining concerns and objections can be addressed without major changes or additional analysis.

[*1: Remove overclaim] My main objection is still that the paper claims to "quantify tipping risks" in the title, even though their analysis does not provide an estimate for the tipping risk. The usage of "tipping risk" is an unsupported overclaim. As Nat. Comm. is a general interest journal the word risk should be used close to what the general public would understand, which is in this case of noise-induced tipping a rate: what is the probability of tipping per time unit? The authors refer to [11], which claims to "quantify resilience". The word resilience is sometimes used in the weak sense as in this manuscript and [11], but "risk", especially in relation to tipping, should involve the probability of tipping.

The proposed (N)BLE method automatically implies an estimate for the tipping rate, because the authors approximate the potential "landscape" shown in their infobox S1 up to fourth order such that they can check if their estimated landscapes have local maxima ("tipping barriers"). Together with their estimated noise levels the authors' statistical method predicts the tipping rate and its uncertainty. However, the authors choose not to report this estimate for the power failure or for the synthetic examples. For the examples they could have then objectively checked if the estimate is realistic.

If the authors expect this rate estimate to be extremely uncertain (because it requires extrapolation), such that they advise against using this estimate, then they also imply that knowledge of the restoring rate and noise level around the equilibrium leaves the tipping rate (risk) equally uncertain.

In summary, please remove any claim that the (N)BLE is able to quantify tipping risk or resilience.

[*2: Sensitivity to window size] Provide a comprehensive sensitivity analysis of dependence on window size for the NAWI power outage and the synthetic examples in the SI. Especially the NAWI pre-outage $\hat{\Psi}$ looks as if it is affected by window size dependent artifacts. The rapid increase of $\hat{\Psi}$ appears to be caused by a single spike, the subsequent plateauing looks as if caused by the 50s window size artifact. So, the argument that there is a "sustained sudden increase in noise level" depends on appropriately addressing of this sensitivity.

[*3: Reproducibility] My original impression about availability of code was overly positive: the cited source https://github.com/MartinHessler/Disentangling_Tipping_Types only provides the time series shown in the paper and SI, not the code reproducing the figures. The source should contain scripts that one can call to reproduce the figures from the original time series, clearly listing the method parameters chosen. If necessary, a snapshot of the library version used should be included.

[*4: Oscillatory nature of NAWI pre-outage time series] Clearly highlight that the pre-outage data has coherent oscillations (e.g., by an inset similar to post-outage in Fig.1) and alert the reader to this feature prominently the text. Also, for the analysis

of the mesoscopic network model S.5-S.6 clearly demonstrate that these models reproduce the coherent oscillations with similar frequency and amplitude as the original data to make them credible. This essential feature is hard to discern from Figure S.3.

Smaller issues:

* Define THIF also in main text

* I do not understand the binning procedure behind the blue curve in Fig S5: the data is highly oscillatory with amplitude about 0.02.

* extend the x axis in the left column of FigS12 (autocorrelation of original time series and (N)BLE) to include at least the first maximum of the autocorrelation (to about $\tau=60$?). This clearly illustrates the mismatch between the dynamics underlying the real data and the (N)BLE models.

(Remarks on code availability)

My original impression about availability of code was overly positive: the cited source

https://github.com/MartinHessler/Disentangling_Tipping_Types only provides the time series shown in the paper and SI, not the code reproducing the figures. The source should contain scripts that one can call to reproduce the figures from the original time series, clearly listing the method parameters chosen. If necessary, a snapshot of the library version used should be included.

Version 3:

Reviewer comments:

Reviewer #1

(Remarks to the Author)

I was already supportive of publication for the previous revision, requesting only several changes to presentation, removal of overclaims, clarifications and improved code availability for reproducibility. These requests have been comprehensively addressed by the authors, such that I have no further comments.

(Remarks on code availability)

The code is stored on a publicly accessible repository github. It relies on a library antiCPy that implements the authors' methodology and is not mirrored in the same place such that there is some danger of future versions of the library breaking compatibility. The README file does not provide detailed instructions for the sequence in which order the scripts are to be run. However, the scripts are not overly complex such that it is possible to follow. The scripts serve mostly to specify precisely the method parameters used.

Overall, reproducibility is ensured to a satisfactory extent in my view.

Introductory Notes

First of all, we thank the three Referees for spending their time on the manuscript, for their differentiated judgments and the detailed constructive comments. We start with a general comment: In the course of the revision process, we were able to compute results with an extended version of the Bayesian Langevin estimation (BLE) which accounts for a second hidden Markov process in the data. It is better suited to treat non-Markovian time series of processes driven by correlated noise as the synthetic test cases in Figure 2(g-l). We call it the non-Markovian Bayesian Langevin estimation (NBLE). In the figure, it is shown in red with green credibility bands. Based on a statistical comparison of the suitability of the BLE and NBLE parameterisation to describe the bus voltage frequency time series (cf. Supplementary Information (SI) S15) we decided to show the NBLE results instead. Nevertheless, the overall results of both methods are largely comparable and do not notably affect the results (cf. Figure S13). Only the slight negative BLE drift slope trend might be better explained by an increasing BLE model misfit which we thus discuss in the SI instead. Furthermore, In the following, we discuss all concerns to our best knowledge.

Response to Referee 1

(1) What are the noteworthy results? The manuscript proposes an Early-Warning Indicator (EWI) and applies it to test examples and a times series from historic frequency data around a power grid blackout 1996. I see the proposal and testing of the new method, a Bayesian estimator for a generic Langevin model (called BL estimation in the paper), as the main noteworthy novelty of the manuscript. Proposing and demonstrating feasibility of a computational/statistical method may not count as a result or discovery in the sense considered by Nature’s science journals. However, the authors claim that the BL estimate is a significant improvement of classical EWIs (such as time-step autocorrelation, variance) that were introduced in high-profile articles in Nature, PNAS, Science etc (refs 1,7,9,17,18,54) are repeatedly and recently discussed there (see [R1,2023] for a striking recent example in Nat. Comm.). Hence, a strong argument can be made that a significant improvement of EWI estimation should be published to the same audience. In my view the proposed BL estimate is certainly a methodological improvement over the above-mentioned classical EWI’s. However, the current version does not convince me yet if the improvement is significant. [R1] Ditlevsen, P. and Ditlevsen, S. (2023). Warning of a forthcoming collapse of the Atlantic meridional overturning circulation. Nature Communications, 14(1), 1-12.

We agree with Referee 1 that a significant improvement compared to the statistical EWIs would be a strong argument for publication. In this spirit, we add new results to Figure 4: Additionally to the Markovian estimator which is biased in case of the non-Markovian time series, we have made major enhancements, restructured the manuscript, and provide several additional results:

1. We add new results to Figure 2: Additionally to the Markovian Bayesian Langevin estimation (BLE) which is biased in case of the non-Markovian time series in (g,j), we add a stable numerical non-Markovian version (NBLE) which cancels out the bias completely (cf. red estimates with green credibility intervals in Figure 2). The NBLE theory is provided in Material and Methods 4.1. It is also implemented in the provided open-source Python package and well-documented [1,2] to be easily applicable by independent researchers. Since we show in SI S15 that the NBLE fits the power grid frequency data better than the BLE, we modified the presentation in the main article accordingly.
2. In the following, we consider the comparison to AR1 and variance. Inspired by the Referee’s comment (1) and (10), we made several additions considering the approximation of the data via an Ornstein-Uhlenbeck (OU) process and its relations to AR1 and variance V . We found these derivations of the Referee’s comment (10) again in an article by Morr and Boers [3] and call the approach OU estimation (OUE). Since in our opinion, the comparison significantly improves the quality of the manuscript, we want to give the discussion adequate space. However, we need to keep in mind the length of the main article. For this reason, we added two new sections in the SI and summarize the results in the main article with references to the corresponding SI sections. Additionally, we claim that it is important to distinguish two lines of argumentation in the context of EWIs. The Referee’s proposal of the OUE as well as the (N)BLE consider specific parameterised models. However, there are several articles which avoid the assumption of any specific parameterisation [4–7]. Therefore, we define “parametric” and “non-parametric” or “qualitative” indicators in the main text. In a next step, we moved the comparison of the BLE to the qualitative non-parametric measure to SI S1 to address also the non-parametric school of reasoning. However, in SI S10 we perform the suggested comparison of the OUE and the BLE. The results show that the OUE yields strongly biased estimates over the four synthetic examples. Furthermore, the OUE results diverge to $\pm\infty$ in almost one half of the time interval in three out of four test cases due to an $AR1\hat{\rho}_{\tau-1} < 0$ which is, where the assumption of an OU process breaks down.

3. We demonstrate that this result is not an artefact of pathological synthetic examples. Quite in contrary, the application of the OUE to Greenland Ice Sheet Melt rates [8] in SI S10 and S16 reveals that these problems can occur in potential real world applications.
4. Finally, in anticipation of the Referee’s comment (2), we demonstrate the widespread potential of the (N)BLE for other fields of research by the analysis of the further real world examples. The most prominent one, the analysis of Greenland Ice Sheet (GrIS) Melt rates is placed in the main article Section 2.3 because, apart from being an application example, the performed analysis has interesting implications. The BLE results provide further evidence for the previously claimed destabilization of the GrIS (cf. ref. [8]), but also suggest a common driver of increasing fast scale dynamics in the late 20th century. Furthermore, a quantitative comparison of the resilience levels over melt rates from a coastal and a continental location hints to an ongoing assimilation process. Following this hint, the dynamics and stability of the the main GrSI might tend to slowly approach the ones of coastal lower regions. In Section 2.3, we briefly discuss also further examples to outline the widespread potential and rather general applicability of the (N)BLE. Apart from the GrIS melt rates, we analyse a biological living system, i.e., a bacteria culture’s collapse due to increasing light stress, documented in SI S5. The experiment was originally performed by Veraart et al. [4]. The drift slope yields a significant destabilizing trend. A last example is only briefly mentioned and we integrate it in the response letter, because it is better placed in another manuscript in preparation. The BLE indicates also the destabilizing trend from an ancient greenhouse earth state into an icehouse earth state (cf. response letter, Figure A1). In summary, we demonstrate by our additions that the (N)BLE has real world application potential from paleoclimatology and climate science to technological and biological/ecological systems like power grids or the bacteria culture experiment.

However, we want to point out that the article is not only intended to propose the (N)BLE to a broader audience. (This is, why we decided to sketch the method briefly in the Materials and Methods section (which we corrected in the revised manuscript according to the Referee’s concerns, cf. later comments)). We claim that, apart from providing a robust statistical time series analysis tool with potential wide range applicability, there are several other important implications:

1. The method does not focus on the deterministic restoring rate ζ alone. Instead, we start by giving several synthetic examples, where the method is able to extract information about both the deterministic and the stochastic part which is connected to bifurcation induced and noise induced tipping mechanisms. (Note that we answer the concerns about the distinction in comment (9)) These information are not reliably extracted by simpler methods like the OUE, and not even accessible if we stay in the non-parametric reasoning of AR1 and variance. The importance of changing noise levels is moreover evident in three real world examples, namely, the pre-outage NAWI frequency data and the GrIS melt rates.
2. The presented results relate the (N)BLE measures to real world power outage scenarios and the interpretation of the modelling approach. Most of the literature about power grid analysis treats network dynamics and synthetic models. Only a few articles (for example [9]) follow the idea of summarizing power grid dynamics by a macroscopic key observable like the bus voltage frequency. This is directly inspired by the practical engineers’ and system controllers’ point of view. However, these works propose that power outages might be accompanied by a rather smooth increase in statistical EWIs. In consequence, they transmit the impression, these phenomena would be mostly caused by a bifurcation in the macroscopic observable’s state space. In our article, we demonstrate that things are much more complex and, in particular, not as smooth as suggested by previous studies (not least due to the significant novelties concerning the metadata, commented on in detail in SI S11). An outage may appear in a data-driven modelling approach entirely caused by a bifurcation, by noise, or by a mixture of both. These theoretical considerations about the potential shape and the grid topology are supported, e.g., by the strong increase of the drift slope $\zeta_{\text{N}BLE}$ when the McNary units are lost in the second red interval of Figure 1(b) and the negative drift slope trend towards the physical pre-outage state in Figure 1(b). Moreover, we complement the analyses by mesoscale modelling attempts and considerations of theoretical frequency models in the added SI S4. This further confirms our reasoning and deepens the overall understanding of our results. Finally, the results are obtained from empirical real world data of a major outage with the possibility of comparing the results directly to real world events which is a very rare case study to our knowledge. This assessment is supported by Referee 2.
3. The advantage of a more reliable noise level estimation and a more quantitative comparability of the estimates is further supported by the GrIS analysis.
4. Not least, the (N)BLE is rather straightforward to adapt to specific modelling needs in different fields.

Figure A1: BLE analysis of the greenhouse-icehouse transition in a $CaCO_3$ proxy time series.

(2) Will the work be of significance to the field and related fields? How does it compare to the established literature? If the work is not original, please provide relevant references. The method as proposed would have widespread relevance for providing evidence of potential tipping from observations in climate science, ecology, social science and complex engineering systems.

We fully agree with Referee 1 regarding the widespread potential of the proposed algorithm. Please note that we answered to this comment already in comment (1). Briefly summarised, we added several analyses of real world examples, ranging from paleoclimatology, climatology, to technological and living biological systems to the manuscript to confirm the widespread applicability of the (N)BLE.

(3) Regarding originality: the abstract uses the word "propose a BL approach", leaving open how original this is. The description of the method is only brief and comes at the very end. It refers to [12] (however, [12] does not employ a Bayesian approach). The numerical ingredients of the BL estimate (estimating probability distributions sampled using MCMC for coefficients of a polynomial model with additive Gaussian noise) may not be new, and in the article the authors do not claim that this part is novel. However, introducing a rigorous statistical method into science has value.

We thank the Referee for the correct evaluation regarding originality. We do not claim the single ingredients of the procedure to be new, but can confirm that the composed algorithm, is our original work. The formulation "propose a BL approach" shall not doubt that originality, but underline that we introduce it for the first time in the context of real power grid resilience analysis (and further examples in the revised manuscript). In particular, the frequency data turned out to be a rather convincing application example for the expanded version, the NBLE.

(4) Does the work support the conclusions and claims, or is additional evidence needed? There are severe problems with the presentation: the paper spends space introducing and discussing tipping, has a detailed introduction of the test examples, followed by an extensive discussion of the behaviour of the BL estimate on these test examples, then a minute description of events and data at the 1996 blackout (using power grid specific jargon), without ever describing the BL estimation proposed. The term is used throughout the paper without explanation or context until the brief description in "Methods".

After reading the manuscript again under the impression of the Referee's comments we fully agree about that point. Following the Referee's objections, we decided to reorganize the manuscript: As already explained above, we relocated the discussion of the introductory synthetic datasets including AR1, variance and OUE to the SI to shorten the introduction. The gained space is used to sketch the (N)BLE in the introduction and refer to the majorly advanced Materials and Methods section. We stay with documenting the Methods in the end, since, as stated above, they are not the main focus. This streamlines the presentation.

(5) The description in "method" makes various ad-hoc choices that are not discussed (the supplementary material would have been the place to do this): for example (and apart from the Bayesian priors), it includes 2 higher-order expansion terms of the right-hand side beyond the linear term, while only the linear term is used.

The linear term is referred to as zeta in all figures but as θ_1 in the method. A Bayesian approach iteratively incorporates additional data to update the current belief about the probability distribution of an unknown. The paper does not describe what is iterated over. Is it time? That is, are the priors assumed to be the coefficients at the earliest time and then the measured time series points for increasing times are "fed into" Bayes' theorem to update the distribution, such that we obtain time dependent coefficients simply because of the iteration process? Or do we have a prior distribution of time-dependent coefficient functions $\theta_i(t)$ and then we apply all time series data points at once? This makes a difference: in the former approach the estimates at the beginning of the times series are more strongly influenced by the prior, while in the latter approach all estimates are influenced equally.

We understand that the description of the method is too brief in the end of the article. Therefore, we provide further details to avoid misunderstandings. In particular, we summarize the included details following the points of Referee 1:

1. The Bayesian part of the method refers to assuming the parameters θ_i to be variables, whereas the measured data $x(t)$ are assumed to be fixed observations. Additionally, we start from Bayes theorem which in principle accounts for prior information. Both points are explicitly based on non-frequentist statistics' reasoning.
2. The priors are chosen to be almost uninformative: Only the Gaussian priors for the higher order coefficients $\theta_{2,3}$ are slightly informative. They only express that the chosen prior boundaries of these parameters should be assumed to be a priori less probable. The almost uninformative priors lead to a higher coincidence of the maximum posterior and a maximum likelihood estimation. However, the weak prior assumptions are chosen for good reasons which we also include in the method description. In most real world cases, only poor prior information about the system are available, contraindicating strong prior assumptions. One example is the exact model for the frequency dynamics. Nevertheless, note that the framework allows for easily including prior knowledge if available for the system at hand. In our case, we prefer to not restrict the estimation procedure in favor of too strong prior assumptions.
3. It holds $\zeta \neq \theta_1$. We are sorry for the misunderstanding due to the short description of the method. Since we assume to start our observations in a fixed point x^* , the drift term is Taylor expanded around this fixed point x^* . Under these circumstances, we expect that $x - x^* < 1$ and $(x - x^*)^4 \ll 1$ so that truncating the expansion after the third order is justified. Taking into account these higher order terms tends to stabilize the method also for moderate to high noise levels. A linear stability analysis of the Taylor expanded drift term h_{MC} results in

$$\zeta = \left. \frac{dh(x)}{dx} \right|_{x=x^*} = (\theta_1 + 2\theta_2 x + 3\theta_3 x^2)|_{x=x^*}.$$

4. Referee 1 is right. There are numerous possibilities for the iteration procedure and initialization of the priors. We simply iterate over the rolling windows in time without updating the priors from window to window for several reasons (however, the principle implementation would allow to do so with some adaptations): One of the most straight forward incorporations of prior knowledge would be to center the prior probabilities of the current window around the estimated values of the previous one. However, since we expect a state change we do not want to introduce a prior bias of "no change". Furthermore, the results in the manuscript illustrate that no stronger priors are necessary in our analyses, since they are rather smooth, i.e., the estimations of subsequent windows converge nearly to the same values. In our approach the coefficients are estimated with all data per window at once. The time dependence of the parameters θ_i follows from the locally stationary rolling window approach.
5. We appreciate the Referee's idea of updating the estimation scheme by incoming data points. Indeed, we also tested such an implementation. A disadvantage for our purpose is that we expect the coefficients to change over time (since we expect a transition). This makes it necessary to assume an explicit time dependence of the parameters. In consequence, the algorithm becomes much more rigid with regard to the chosen time dependence model. Not least, the resulting estimation is numerically notably less stable.

(6) The power grid example, also does not fully support the claim of prediction of tipping. The narrative suggests that the BL estimates $\hat{\zeta}$ detect an event that occurred at 15:40 (start of first red interval in figure 5). If that change of $\hat{\zeta}$ and $\hat{\sigma}$ had any predictive significance for the following spike at 15:42 is less clear to me (the authors claim that). However, a similar severe subsequent change of $\hat{\zeta}$ and increase of $\hat{\sigma}$ at 15:43 did not lead

to a spike.

We appreciate the constructive doubts and are sorry about the confusion, probably caused by comparison with statistical EWIs. We agree with Referee 1 that the example does not support the claim of tipping prediction, but emphasize that we do not use the word "prediction" in the direct context of our results for good reasons. The manuscript title also underlines that we focus on a quantitative inference of tipping risks in complex systems, i.e., the changes of the system dynamics such as the restoring force/drift slope ζ (related to B-tipping) and the internal noise level σ (related to N-tipping).

The official triggering of the cascading failure is the Keeler-Allston line tripping around 15:42, caused by a tree-to-line contact. The peak at 15:42 marks only the actual line tripping. However, the tripping is caused by the line touching a tree which leads to a short circuit due to grounding. This is a process which does not happen instantaneously. First of all, the line has to elongate due to increasing heat, probably caused by heavier loads in the grid. Otherwise, the line would not hit the vegetation. Secondly, even if a tree-to-line contact is established, it is not permanent. The electric current changes already, but the progressing carbonization of the woods completes the permanent contact. These thermo-chemical processes take place in order of seconds up to a few minutes [10, 11]. This is the reason why it is likely that the permanent state change detected by the BL estimation is related to the actual line tripping event. Nevertheless, Referee 1 is right to question the increase of the metrics around 15:43. It is not related to the dynamics of the system, but has numerical reasons: The missing estimates in the beginning of Figure 5 account for the rolling window size. If this is compared to the increase of the metrics around 15:43, it becomes obvious that the increase of the metrics is caused by including and excluding the tripping spike about one window length. In contrast, the dynamical change around 15:40 is persistent and independent from the window length which makes it very likely to be a significant state change. We clarified these points in the revised manuscript.

(7) The strongly increased variance of the frequency time series pre-outage (std increased by a factor of ≈ 10), compared to the apparently standard variance (see post-outage) is a strong EWI on its own.

We thank Referee 1 for the accurate observation, included a remark in the manuscript, and used the information in the modelling SI S4. However, we guess that the Referee hints to the (N)BLE noise level $\hat{\sigma}$, not the variance $\hat{\sigma}^2$. We checked the variances of both data as well. The variances on interval 1 and 3 (cf. Figure S14) are almost equal ($3.865 \cdot 10^{-5}$ and $3.721 \cdot 10^{-5}$, respectively). Nevertheless, we also hint to the fact that the two empirical time series are taken from very different sources (analogue paper scan vs. digital measurement) with completely different time resolutions. Differences between the datasets might be (partially) also caused by that fact.

(8) Are there any flaws in the data analysis, interpretation and conclusions? Do these prohibit publication or require revision? The authors refer to bifurcations for systems with noise, even though it is not clear in which sense they mean this. For example, for their test example Eq(2), Crauel [R2] showed that additive noise destroys the pitchfork bifurcation: one has a single globally stable attractor for all parameters. [R2] Crauel, Hans, and Franco Flandoli. "Additive noise destroys a pitchfork bifurcation." Journal of Dynamics and Differential Equations 10 (1998): 259-274.

We thank the referee for the interesting comment and enjoyed reading the recommended article. The authors of the reference stick to a very formal distinction of a "static" and a "dynamical" bifurcation. They write:

No two (not even random) regions of the state space stay physically distinguishable for all times. The "dynamic" bifurcation is destroyed. However, the "static" bifurcation may be argued to survive. The distribution of the "random attractor" a_α , (which moves in a stationary way) has a one-peak density for $\alpha < 0$ and a two-peak density for $\alpha > 0$.

Apart from the classical static bifurcation (i.e., the emergence of two stable states and destabilization of the previously stable one) they define the dynamical bifurcation to be characterized by two close initial conditions of different sign which converge to a fixed distance from each other. It makes sense that this does not hold for noisy systems.

However, the referee is right that we did not state in which sense we mean it. We include the reference and hint to the static bifurcation defined therein. Stating this, Crauel et al.'s dynamical bifurcation definition which is compromised by noise does not touch the basic reasoning of our study example in Figure S1. Moreover, in Figure 2(a) we consider one single realisation with one single initial condition in the static regime of the double-peaked state probability density (PDF) without tuning the control parameter α . In that case, there is no bifurcation included in the simulations, but only the random jumps between the two branches of the

double-peaked PDF. We hint also there to the double-peaked PDF and the recommended reference.

(9) *The introduction spends time expounding on the importance of distinguishing between a change in decay rate in the "deterministic part" of the (assumed-to-exist) Langevin model and a change of the noise amplitude. However, this distinction is artificial as every change in the noise amplitude can be transformed into a change of the deterministic part (as long as the noise amplitude does not go to 0, see [R3]). [R3] <https://math.stackexchange.com/questions/4673542/transformation-of-ito-diffusion-such-that-the-diffusion-coefficient-becomes-cons>*

We thank the reviewer for the interesting comment. We remark that we have strong theoretical background for the assumption of a Langevin description of the macroscopic frequency dynamics. We hint to several references [12–14] and included an SI section S4 in which we further outline the relationships of macroscopic and mesoscopic descriptive levels for power grids.

Anyways, indeed, the Lamperti transform [15] is designed to normalize the variance of a stochastic differential equation (SDE) to one. Nevertheless, this normalization transformation is mainly used to

1. facilitate and stabilize numerical simulations of arbitrary SDEs,
2. successfully employ, e.g., Kalman filters for statistical inference,

and does not touch the formal nature of the Langevin equation. The original Langevin equation describes Brownian motion, i.e., the apparently random movement of a particle in a fluid due to random collisions with the molecules around. In the original scenario, the macroscopic velocity change of the particle is given by the deterministic Stokes friction and the stochastic random collisions. This distinction is an important feature of our statistical inference approach because it considers two time scales relative to the resolved macroscopic bus voltage frequency time series. This time scale separation of the model in a slow deterministic, macroscopic evolution and fast stochastic microscopic contributions is not touched by the Lamperti transformation. In particular, applying the Lamperti transformation

$$\Lambda(x(t), t) = \int \frac{1}{g(x(t), t)} dx$$

to the model parameterization

$$dx = h(x(t), t)dt + g(x(t), t)dW = (\theta_0 + \theta_1 x + \theta_2 x^2 + \theta_3 x^3)dt + \sigma dW \equiv h(x)dt + g_{\text{const.}}dW$$

leads to the transformed dynamics of the variable Z :

$$\begin{aligned} dZ &= \left(\frac{\partial \Lambda}{\partial t} + \frac{\partial \Lambda}{\partial x} h(x) + \frac{1}{2} \frac{\partial^2 \Lambda}{\partial x^2} g_{\text{const.}}^2 \right) dt + \frac{\partial \Lambda}{\partial x} g_{\text{const.}} dW \\ &= \left(0 + \frac{h(x)}{\sigma} + 0 \right) dt + 1 \cdot dW \\ &= \frac{1}{\sigma} (\theta_0 + \theta_1 x + \theta_2 x^2 + \theta_3 x^3) dt + dW. \end{aligned}$$

The above equation could be used for the Bayesian inference instead, but the coefficients θ_i and the noise level σ are present as before. Furthermore, summarizing the variable in terms of $\tilde{\theta}_i = \theta_i/\sigma$ only hides the information about the noise level in the new parameters $\tilde{\theta}_i$, but does not cancel it in terms of the Langevin equation's separation of deterministic and stochastic dynamics.

(10) *The argument that autocorrelation and variance of the time series do not permit to conclude separately about input noise amplitude σ^2 and decay rate $-\zeta$ is also not quite correct. Assuming that in each sliding window the time series is approximately governed by an Ornstein-Uhlenbeck process, the one-step autocorrelation AR1 determines ζ through $\zeta = (\log(\text{AR1}))/\Delta_t$ (Δ_t is the time step) and the variance V determines the σ^2 through $\sigma^2 = -2V(\log(\text{AR1}))/\Delta_t$. The estimates $\hat{\rho}$ and $\hat{\sigma}$ in Figure 3 look consistent with the above expressions. So, Figure 3 should plot the decay rate and noise amplitude implied by AR1 and V , and compare that to the quantities determined by the BL estimate. (They may still be less accurate, which would demonstrate superiority of the BL estimate.)*

We thank the referee for the valuable suggestion and followed his comment. Please note that we already brought the answer forward in comment (1).

(12) *Is the methodology sound? Does the work meet the expected standards in your field? The analysis methodology is sound as far as I could judge. The presentation is poor: the article is too long, spends too little space on the main method/result proposed in the abstract, and far too much on tangents (general tipping background, details of the power grid event).*

We agree about the points and made already several notes to corrections. We briefly summarize them again.

1. The significantly shortened the introduction by relocating the comparison to statistical EWIs and the OUE to the SI. Only the valuable main results are stated in place.
2. We sketch the method briefly in the main text to accomodate the reading flow.
3. We include several important details of the method in the corresponding Materials and Methods section.

(13) *Is there enough detail provided in the methods for the work to be reproduced? The code for BL estimation and reproducing the results is available according to references. However, as the description is very short, one would have to apply the code in a black-box fashion without understanding what is happening.*

We confirm the Referee's comment that the method and code is freely available and understand the concerns regarding black-box fashioned algorithms. However, our implementation is not a black-box for several reasons:

1. The whole code is written following best practices (<https://geo-python.github.io/site/notebooks/L1/gcp-1-variable-naming.html>, <https://www.appacademy.io/blog/python-coding-best-practices>) with readable class, function, and variable names and completely open to study. In that sense, this is the most detailed documentation of an algorithm on a numerical level which one could think of.
2. Furthermore, the whole package is documented in text form at <https://anticpy.readthedocs.io/en/latest/>. This includes short descriptions of the mathematical methods.
3. The github repository allows for customizing the package to individual needs.
4. Additionally, we include a more detailed description of the numerical procedure in the introduction and Materials and Methods section of the manuscript.

Response to Referee 2

Major remarks

(1) *The manuscript is very well written, but it is simply too long. The 1996 blackout case is discussed too extensively, with a lot of information being presented sometimes thrice! The example is exceptional – as is the work performed here – but it becomes too cumbersome for the reader. I would actually suggest one big addition, which could solve a lot of issues: Both a map of the affected areas, with the 3 and then 4 island formation, with a detailed timeline below, as a long left-to-right arrow, which each marking of the time stamps referred in the text. This would greatly increase the readability of the text – an image is a thousand words here. Overall, the case study needs to be streamlined for readability.*

We thank the Referee for the very constructive comment. We followed the suggestions regarding the time line and the illustration of the islanding process. Additionally, we combine the new figure with the presentation of the main results to streamline the overall results related to the real world event. Dublicate information are cancelled from the main text body.

(2) *As commented above, the manuscript lacks a certain congruence. The introduction focus heavily on tipping, the limitations of the two seminal CSD metrics, then we move to the Langevin model, and later the case-study of the 1996 power outage. The Discussion and Outlook focuses on the case study and steps forward to improve estimation using BL, yet we never arc back to discussing CSD metrics and only a few mentions are given to R-tipping. An outlook on the method is given, with suggestions to the Kramers' escape rate, which is a new topic in this regard.*

After reading the manuscript again, we agree with the Referee's comment and made mainly three adaptations:

1. Since the comparison was intended to briefly motivate our Bayesian Langevin approach (BLE) and the non-Markovian extension (NBLE) (which we additionally consider in the revised version), but occupied too much space, we moved it into the supplementary information (SI) S1 and S10. This balances the length of the introduction and augments the focus on the more important theoretical description of the (N)BLE, the macroscopic power grid frequency model estimation, and the subsequent real world analyses.
2. The conclusion is adapted accordingly and summarizes first the main findings of our study. Afterwards we briefly outline possible future research attempts.
3. Note that we added several further analyses of real world examples ranging from paleoclimatology, climate science, and biological living systems into the main body or SI with respect to the Referees' comments. In particular, the Greenland Icesheet is a prominent further example of increasing noise levels in the main text. For more detailed comments on this we refer to the answers to Referee 1. Moreover, in SI S15, we show that the NBLE yields a better description of the frequency time series. The overall results are not altered at all, but the provided comparison provides valuable additional insides in our opinion. We adapted the results accordingly. Not least, we enrich the SI with naive modelling attempts of the NBLE fingerprints during the NAWI outage (cf. SI S4). The addition addresses comments of Referee 3 concerning the theoretical background. We show that the real world NBLE fingerprints' qualitative behaviour can be recovered in a naive model of three key outage events in a mesoscale model guided by the approved disturbance report [16]. It might be seen as base for further discussion and developments. Nevertheless, we consider the real outage analysis with a direct comparison to real events as the key results.

Major questions

(1) In Eq. 1, we find a vector formulation with underbar x . Strictly speaking, the diffusion \underline{g} is a matrix and not a vector, as it is multiplied by the noise vector $\underline{\Gamma}$. Have the authors chosen the current nomenclature for a particular reason? Furthermore, for consistency, should the left-hand side not read $x\text{-dot}(x(t),t)$ instead of $x\text{-dot}(x,t)$? The authors are careful in including the time dependence of $x(t)$ on each function/vector/matrix on the right-hand side.

We appreciate the precise comment and corrected the notation.

(2) Continuing on this issue. The authors present, in the most general form an N -dimensional Langevin equation, but the work and most subsequent examples are 1 dimensional, as is the case study. The justification for power-grid frequency seems solid, as it is generally almost perfectly identical regardless of where in a power grid it is measured. The work in this manuscript relies heavily on antiCPy, which, to my understanding, is also only a 1 dimensional time series analysis tool. The authors offer a convincing argument to the study of 1-D time series, where the presented N -D Langevin equation might be misleading. It is not straightforward, for voltage analysis across a power grid, to reduce the problem to a 1-D time series. Much less so for climate or ecological studies. Could the authors comment on this? I would push for a less ambitious case herein only a 1-D case is presented.

It is true that we deal with one dimensional time series and antiCPy is designed only for that purpose up to now. Therefore, we adapted the Langevin equation in the introduction to a one-dimensional version depending on ω . However, we have preliminary results in which a two-dimensional BLE version announces the firing spike of a two-dimensional time series from a Neuron Hindmarsh-Rose model. Therefore, we point out in the main body that the basic framework of our estimation procedure—in particular because of the open-source nature of the code—is rather flexible and can be adapted to higher dimensions or specific needs of other research fields. Furthermore, the Infoboxes are intended to provide mainly conceptional information and were intended to be kept as general as possible, but with context to the main topic, i.e., power grids. We hint therein also to $\omega \neq \underline{x}$ to clarify that the power grid is a highly complex system with high-dimensional states \underline{x} which might be treated on various coarse-grained levels. These may be thought to be a solution of an assumed-to-exist N -dimensional lower scale model of a power grid. The crucial step is to outline the partial aggregation of these information in a macroscopic observable which is the frequency ω in this case. We further elaborate on that in SI S4 where we discuss in detail the theoretical justification of the (N)BLE applied to power grid frequencies and relations to less coarse-grained descriptions.

Not least, in course of the revision we added several real world examples which confirm that a heuristic approach of the one-dimensional (N)BLE can provide interesting insights in complex systems over a wide range of research fields (cf. in particular the answer to comment (1) for Referee 1; cf. also further comments to Referee 1). In the context of bifurcation induced tipping, we want to hint to Haken's slaving principle [17], which basically means that fast scale degrees of freedom are forced to follow slower ones. These slower ones should be long term stable order parameters. The frequency is not the order parameter of the system, but nevertheless a

slow key observable. When the system approaches a bifurcation, the dynamics are progressively slowed down (critical slowing down [6]) which results in increasing dominance of the slow variables due to the slaving principle. Against that background, a high dimensional system implodes and a low dimensional heuristic description becomes approximately correct for observables near a bifurcation if they represent the order parameter of the system or are at least reasonably related to it.

(3) Eq. 19 has two parameters θ_0 and θ_1 on the left-hand side, yet depends only on θ_1 on the right-hand side. Is there something amiss here?

We agree that the notation might be confusing without further explanation. For a straight line $y = ax + b$ in 2 dimensions, the chosen prior is invariant under a variable transformation, i.e., unbiased with regard to the choice of the representation. It can be shown (cf. <https://jakevdp.github.io/blog/2014/06/14/frequentism-and-bayesianism-4-bayesian-in-python/#The-Prior>) that the prior is therefore uniform in b and $a = \sin(\alpha)$ with the angle $\alpha = \tan^{-1}(a)$ between the x-axis and the line y . Uniformity of a in $\sin(\alpha)$ instead of α is a result of the influence of b . In congruence with these considerations, we added some sentences to clarify the notation.

(4) What is the time series Gemini?

We thank the referee for the valuable feedback. It should underline that the presented time series of Figure S1 are rather similar, i.e., "time series twins". However, the question highlights the potential confusion and we cancelled the unprecise wording.

(5) It is hard for me to follow why Eq. 7 diffusion term is referred to a "coupled" noise amplitude. Coupled noise would seem to me to pertain to the non-diagonal diffusion matrix terms of a 2- dimensional stochastic process, which couples noise across the dimensions of a 2-d process. Is there a reason for this choice of wording?

We agree with the referee that the wording might not be precise enough. The wording should only clarify that the noise amplitude σ_{coupled} corresponds to the noise amplitude of the process y which basically "couples" as a correlated noise term into the x dynamics. We changed σ_{coupled} to σ_y to avoid confusion.

(6) For the correlated B-tipping model, the authors refer to Eq. 4 being coupled via the constant $q = 0.5$, but there is no constant q in Eq. 4. Subsequently, the noise term from Eq. 7 is $\sigma = \sqrt{c}$, which is not related to the noise but to the drift term. I find this quite confusing. Could the authors try to alleviate this and include directly an equation for this process?

We appreciate the clear hint and agree that including the whole set of equations for the correlated B-tipping set facilitates the reading flow significantly. The coupling constant q was thought to replace the multiplicative coupling of y into x in Eq. 4. Indeed, the OU process was kept as simple as possible for demonstration purposes with only one independent parameter c for drift and diffusion. However, the method applies independent from that choice as shown by the correlated drift-diffusion varying dataset $x_{g,h}^{\text{corr}}$. Please note also the additional results of the NBLE. It completely cancels out the bias of the hidden OU process in the non-Markovian examples in Figures 2(g,j).

Remarks and smaller questions

(7) In the second paragraph of the introduction, we find "[...] in connection with uprising bifurcation induced transitions [...]". What is the meaning of "uprising" here?

We are happy about the comprehensive minor corrections and English style comments. Indeed, this is basically a wrong wording. It was meant to mean "emerging". However, we improve the reading flow by deleting the word without replacement.

(8) In the third paragraph of the introduction, we find "Furthermore, the currently very important trend to an environmentally sustainable grid architecture will probably augment the amount of noisy grid participants significantly in future". This is a claim that requires a citation. I suggest the authors look at Milano et al. (2018) doi:10.23919/PSCC.2018.8450880.

That is a correct claim and a great literature suggestion. We enjoyed reading the article and included the citation.

(9) *The remark “Nevertheless, industrial and scientific stakeholders only report activations or deactivations up to a certain extent, and railway traffic can be heavily disrupted, for example, due to damage to the railway infrastructure caused by a storm or unforeseen strikes” seems out of place. I cannot follow what is the relevance of this statement in the context of the narrative. Could the author elaborate on this? Maybe best removed? – I would also like to point out to the authors that, to the extent of my knowledge, this might not be the best example to use. Various EU countries operate their train systems and 16.6 Hz instead of 50 Hz. Admittedly the power source is still the same, but for the argument the authors are seeking, there are better examples to pick. (Some Sbahnen are even DC powered).*

Considering the text passage again, we follow the Referee’s advice and removed the example. Originally, the example should illustrate that individual components of the power grid might act on various time scales. In consequence, changes of their dynamics may influence also the grid frequency on different scales, i.e., $h(\omega(t), t)$ and $g(\omega(t), t)$. Since industry and power units are often involved in response and demand agreements, they partially act on pushing frequency deviations back to the reference frequency. At the same time industry might act on faster scales by partially stochastically energy demand. The example’s intention was to clarify that. However, since it led to confusion instead, we reduce the example to industrial consumers and power units. This might suffice for to illustrate the idea.

(10) *The use of the term “mountainside”, although clear, seems like more added baggage. A much more common term – which the authors also use – is “landscape”. This is a term used almost uniformly from psychology to ecology and physics. I suggest the authors stick to “landscape” only.*

We agree with the referee that “landscape” is used over multiple disciplines and is the base for every figurative explanation of critical transitions. However, to our knowledge “mountainside” and “landscape” are not synonyms. In the manuscript, the word “mountainside” is used to refer to the slope of the mountain below the mountain top following the figurative style of the Infoboxes. A quick search in the Oxford English Dictionary supports this meaning (<https://www.oed.com/search/dictionary/?scope=Entries&q=mountainside>):

mountainside, n.

the sloping surface of a mountain below the summit. Also figurative.

Thus, for now we stick to the wording. However, if we misinterpret the English meaning, we appreciate a hint to the correct term.

(11) *Pertaining to the sentence “Note that power outages as well as control mechanisms live on time scales of seconds up to several minutes, e.g. global grid primary control actions in Germany react to a maximum frequency deviation of ± 0.2 Hz in times up to 30 s and have to be stable under full load up to 15 min before the simultaneously starting secondary control in regional grids should replace the primary control completely.” Firstly, this seems to refer to article 31 which seems to point to regulations from 2003. Are these still the statutory limits in Europe/Continental Synchronous area?*

We appreciate that the reference is checked carefully by the referee. It was difficult to find a reliable citation and we were initially happy with that one. Anyways, we checked up again and found a revised version from 2022 which confirms the stated requirements for primary control power units in Germany. Of course, we updated the citation accordingly.

(12) *Secondly, it might be better to refer to the regulation of the Continental Synchronous area in general. Germany is but one of various country in this supergrid.*

That is a good point. Including these information improves the quality of the manuscript. To this end, we read the legal text from the European Union which deals with the regulation of the European Transmission Grid. The guidelines for the Continental Synchronous Area are congruent to the German ones (as it is part of the continental synchronous area). However, we included also information about the interconnected grids with the corresponding citation. All grids’ primary control mechanisms operate in the same order of magnitude regarding frequency deviations and activation times, i.e., ± 0.2 Hz to 0.5 Hz and 10 s to 30 s (cf. main text of the manuscript for more detailed information).

(13) *The (although commonly used) abbreviation “pdf” in the last paragraph in page 2 is not introduced. For consistency, introduce it.*

We agree that the abbreviation is not introduced at the first place in the main text. It was erroneously introduced in the Infobox 1 which is referenced first. The referee is right that this is against the convention of defining the abbreviation in first place of the main text and might not be seen if the reader prefers following the main text only. We corrected the issue by introducing it additionally in the main text and keeping the cumbersome definition in the Infobox, since the boxes are designed to stand alone in the sense of illustrative explanations of the basic mechanisms.

(14) *The sentence “Since a sustainable adaptation of our power grids is indisputable” at the end of page 2, reads weirdly. I believe the authors intend to imply that renewable energies will become ubiquitous. The way it is written implies that there has been a steady adaptation of the power grid, which is not necessarily the same. I believe rephrasing this would bring the point to the readers in a clearer fashion.*

We are glad to correct for the imprecise formulation. The referee is completely correct regarding our intention and we hope our alternative formulation is clearer.

(15) *Comparing the titles of Infobox 1 and Infobox 2, Infobox 1 should include a mention to “tipping”, as this discussed therein.*

We modified the headline accordingly.

(16) *In Infobox 1, Figure 1, we read “[...] be captured by the diffusion $g(x(t),t)\Gamma(t)$ [...]” but further on, e.g., in page 4, bottom, “[...] in principle contribute to the diffusion [sic] $g(x(t)t)$ [...]” (note the missing comma!). In my view, g is the diffusion, not $g\Gamma$. It would be best to be consistent here.*

We agree with the referee that $g(x(t),t)$ only should be denoted as the diffusion and modified the manuscript. The comma is also included. Note that we changed the formulation in the box slightly to avoid the wrong usage of “diffusion”, but keeping the whole term $\underline{g}(x(t),t) \cdot \underline{\Gamma}(t)$ as “intrinsic stochastic dynamics” to stay congruent with the orange color-coding of the Infobox.

(17) *In Infobox 1, “[...] If the restoring force (light green arrow line) given by the steepness of the potential mountainsides is high enough [...]”. I believe the correct English form here is “[...] is large enough [...]”, but do check as I am not a native speaker.*

We checked for English language examples which confirm that the referee is right. We corrected the wording.

(18) *In Infobox 2, “[...] but under stationary noise level if the potential landscape is negatively affected”. I believe is this a bit of an abuse of language. This landscape is not an actual landscape, thus it cannot be negatively affected (in a sense of a loss of vegetation, flora, of fauna). This is still a function, which cannot be “negatively affected”. Maybe there is a better wording to be found here?*

We are glad to suggest an alternative with a more precise wording during keeping the figurative language of the Infoboxes (cf. highlighted manuscript).

(19) *Eq. 7 includes, not erroneously, a noise term dW . I would advise against this. In the current formulation, the authors use x -dot, which implies a $dx(t)/dt$ (more conventionally a physics notation) on the left-hand side. The drift on the right-hand side is correct, yet the final term of the equation is a differential, which is incongruent with the notation. I think it is best to stick with $\Gamma(t)$.*

The referee is right. We changed the notation to $\Gamma(t)$ and included a short hint to the Wiener process assumption $\Gamma(t) = \frac{dW}{dt}$ in the method section. Note also that we basically forgot the denominator in the original manuscript’s notation.

(20) *Eqs. 2, 4, and 6 all have their respective control parameters as $-\nu$, $-r$, and $-\alpha$, yet, their values are also all negative. Not critical, but feels a bit pointless. Would it not be more elegant to just have these are positive values and remove the minus signs?*

That is a good point. Actually, we decided consciously to change from positive to negative notation, since it is convenient for plotting the control parameter trends within the corresponding figures. We thought including

the actual control parameter trends directly in the figure helps a lot to understand what is going on behind the scenes. For example, readers which are less familiar with the topic might expect the whole time series to be stationary. That is, e.g., due to the control parameter changes, not the case. Nevertheless, this information is basically given by the ground truths of the drift slope. Together with the referee’s comments on the figures we decided that it would be best to remove the control parameter plots. Thus, we also changed to the simple positive notation.

(21) Eqs. 2, 4, and 6 all introduce new notation for h and g with various different subscripts, like h_{pf} . This seems superfluous as there are never used in the text. The only terms mentioned are X_g and so on. I suggest alleviating the notation, as there is little gain here in separating h and g . They are all the same mathematical objects, just with different functional values in the various examples.

We are happy for that hint. Indeed, it reduces the notation’s overhead.

(22) Subsequently, the use of the term set in N -tipping set X_g might be misleading. These are not sets, just stochastic processes. Note that the last one is denoted correlated B -tipping model and not set.

Thank you for the valuable comment. The error in the manuscript is due to imprecise wording. We modified “set” to “dataset”, since we denote specific realisations $X_{...}$ of the general stochastic process equations.

(23) There is also an inconsistency in the presentation of Eq. 6 and 7, which give the SDEs directly, instead of the drift and diffusion term. I think it would be more elegant to stick to one notation only and report as in the previous examples.

We adapted the notation accordingly.

(24) In the same line as above, the last example, denoted correlated B -tipping model is not given the honour of an equation, just inline passages. It would be best to also give this model an equation, for presentational purposes.

We thought it might shorten the main text, but we agree with the referee that a systematic presentation has to be preferred. We included the fourth model with explicit equations.

(25) Fig. 4 and Fig. 5 are missing the y -labels for the plots. It is best to include those. Moreover, the sub-plot labels need work: Fig. 4 has no parentheses and the labels are in the subplots, Fig.5 is the opposite. For Fig. 5 I suggest removing the legends entirely and instead include the terms in the y -label, i.e., $\hat{\sigma}$ and $\hat{\zeta}$. The rest can be mentioned in the caption. For Fig. 4 all legends can be removed, $\hat{\sigma}$ and $\hat{\zeta}$ added accordingly in the y -labels, and the rest referred to in the caption.

We are happy to adapt the figures to improve the overall quality. Originally, we have omitted the actual y -labels, since the figures show *different* quantities, e.g., the drift slope ground truth, the drift slope estimate, and the control parameters in one axis. Using one y -label would have been inconsistent. However, we decided to remove the control parameter plots because the crucial information about non-stationarity of the process is already included in the varying ground truths. In that way we achieve consistent y -labels and can also account for the double minus notation of the control parameters which was only chosen because of the space-efficient plot. To solve the inconsistency issue regarding the subfigure labels we propose to keep the positions (since they are the most space efficient ones), but make them identical, i.e., we removed the parentheses and wrote all subfigure labels in bold type. The changes enabled also a more precise usage of the legends.

(26) In section 2.2.1, the authors write “[...] grid components were satellite-synchronized.” I suggest the authors write “GPS-synchronized”. This is more appropriate for it clearly states what timing method was used (if that was indeed the case!)

The approved disturbance report underlines that the Portland Area Dispatch SCADA system was not satellite synchronized. This is the reason given in the report why it cannot be aligned precisely in the time line of events. We agree with the referee that the wording probably refers to the GPS satellite system. However, there is no further explanation or more specific technical term than that included in the approved report. We were not able to find a more specific hint to the actual satellite system. It might also be that a combination of the GPS (USA) and GLONASS (Russia) system was used at that time (even if it might be unlikely due to the vicinity to the end of the Cold War). At least today, private parties have access to both (and more) GNSS systems. The crucial information for the manuscript is that the time stamps are well-aligned without technical issues

which makes the interpretation more robust. Since we are not experts in time synchronization methods, their history, and their technical implementation, we would prefer to keep the same wording as in the approved report.

(27) In section 2.2.1, page 11, the authors include “Additionally, power grids are commonly modeled by coupled Kuramoto oscillators which contain a linear damping term. This damping can be interpreted in reality as control actions in order to stabilize the deterministic dynamics of the system.” This seems misplaced. The narrative is about actual events. The way power grids are modelled is perfectly unrelated to the question at hand.

We agree that the focus of the article lies on the analysis of the actual NAWI power outage event which is a well-documented case to compare individual real world events with the coarse-grained drift-diffusion coarse-grained frequency dynamics. However, in order to address concerns of Referee 3 regarding the theoretical model base we constructed a simple mesoscale model of the pre-outage period, the Keeler-Allston line tripping, and the McNary units with subsequently insufficient damping based on the classical swing equations (cf. SI S4). For suddenly decreased damping as indicated by the approved report due to failing equipment, we observe an increase of drift slopes and noise levels in line with the very end of the real world time series analysis. We accordingly adapted the paragraph and added a reference to the SI, since it provides further insights into relations between different scale models and the NBLE.

(28) In Section 3, page 14, the reference to Kramers’ rate requires a citation, as this might not be known to all readers.

We agree that citations are necessary. We include the Kramers’s original work, a didactic example including a comparison between Markovian and non-Markovian models, and an article that introduces a correction of Kramers’s escape rate for multiplicative noise.

(29) In Section 5, between Eq. 10 and Eq. 11, “of the corresponding Fokker-Planck equation of the time series (a), with the potential”. What is the “(a)” here?

We included the missing reference to the figure.

(30) Below Eq. 17, the authors use “ $p(\cdot)$ ”. Better to be explicit. This notation is used only once here.

We agree that it might be best to avoid the implicit notation in the manuscript. Accordingly, we restructured the respective sentences.

(31) Section 7 should just be part of a Supplemental Material.

We agree with the referee that it is important to define a precise criterion to decide what should be included in the main article and what should be part of a supplementary information. In this case, the outlier correction paragraph is directly related to the presented results of the restoration time series interval. The grey lines show the results without the outlier correction. Furthermore, the outliers are shown in the insets of the figure. We believe that it is more convenient for the reader to include the short paragraph about the outlier correction in the main article to dispel any doubts about the proper treatment of these points. Following the referee’s argument more details are discussed in the supplementary information.

(32) Reference 33 includes various acknowledgements that I believe should be included in the acknowledgements, wherein I hope they are more visible, and thus pay homage to the people involved accordingly.

We are glad for the detailed comments on the reference section and adapt the acknowledgments in the designated section. Nevertheless, we want to point out why it might be reasonable to keep the additional “acknowledgments” also in the reference section: Apart from the fact that without the help of these people the very intricate search for the missing metadata would have been without success after almost two years, the specific references are not easily available online. If researchers want to read them, they might contact us or even the institutions and persons which were involved in the search. This is the most appropriate way to keep the sources accessible to everyone over the coming decades. Of course, we have asked for permission to include the mentioned persons in the acknowledgments and the reference section for this purpose.

Minor issues, typos, and typography

(33) All references to the use of Gaussian kernels for filtering data indicate the bandwidth chosen with a σ , which naturally pertain to the standard deviation of the Gaussian filter. In this particular paper, this adds a bit of confusion, since the noise term in the various models is also a σ , but that pertains to the magnitude of the noise, not the standard deviation of the pdfs of each model! I strongly advise using a different term for one or the other, to avoid confusion.

That is a very good point. We modified the bandwidth σ and σ_{kernel} (there was some inconsistency) to the short and clear notation β .

(34) The use of the dot in the equations feels a bit unnecessary, and sometimes confusing. Eq. 1, which is written in vectorial form, implies a matrix operation on a vector, g on Γ , without the use of a product, which reads perfectly fine for me. From hereon, we find a consistent use of the dot to multiply scalar and functions, which seems excessive for a standard mathematical operation. I suggest the authors remove the dot, apart from the separation of scientific notation for, e.g. in Eq. 3, where we read $2.4 \cdot 10^{-4}$. [From a typographical standpoint in latex, consider using “.” instead of “.”]. Check, e.g., Eq. 15 vs Eq. 16 and the use of the dot.

We thank for the practical latex advice and applied it to the scientific notation. Furthermore, we introduced a consistent use of the dot in all equations for better readability, i.e., dots are included for multiplication of alphabetic variables and function symbols, but excluded for (de)nominators with more than one summand and for explicit numerical factors like 2 or 2π .

(35) I dislike that Infobox 1 has figure 1, and therein, figure 1 has a caption. It seems best to simply refer to it as Infobox 1. There is no difference between “Infobox 1” and “Infobox 1, figure 1”, it is just a mouthful the last one. Idem from Infobox 2.

We agree with the referee. Originally, it was only introduced because of problems with resolving the Infobox references in latex. We worked on that and could define a customized box that allows for proper referencing.

(36) “Supplemental Material” and “Supplemental Information” are both use. At best, stick to one style.

We now use consistently “supplementary information”. This is the term used in the author guidelines of the journal.

(37) Second, I would exempt from writing “Supplemental Material, section S1”. “Supplemental Material S1” or “Supplemental Information S1” is clear.

We moved to the shorter expression.

(38) The term MCMC is never explained.

Definition is inserted.

(39) The use of “o’clock” should be removed. It reads too colloquial. I would simply suggest adding a sentence on the first reference to the time of the event to start that all measurements are in local, PDT time, and then only use the time, without “o’clock”.

We inserted a footnote for the first time stamp and deleted the “o’clock”.

(40) Please double check the term “1:00”. It should read “01:00”.

Corrected.

(41) Various times one find “on 11th August 1996” or “at the 11th August 1996”. This should always be “on the 11th August 1996”. Either “the” is missing or “at” should be “on”.

Corrected.

(42) The use of $\delta[\omega](t)$ seems cumbersome to me. This reads as an operator on ω , where it is just the detrended timeseries ω . I think a tilde above would suffice and be cleaner.

We believe that the “detrending” can be seen as kind of an operator on the raw time series object. However, we followed the referee’s suggestion for a more lightweight notation.

(43) *The font size of the figures is generally too large. This can be amended later once placed in the final template.*

We will let this point open as indicated by the referee.

(44) *The sub-sub-labels in the Infoboxes are excessive. “Infobox 1, figure 1 a.1”. This could easily just be “Infobox 1 b)”.*

We changed the labels accordingly.

(45) *In Sec. 1: “bifurcation-induced transitions (B-tipping)” is not italicised; “rate-dependent tipping (R-tipping)” is italicised; “noise-induced tipping” is partially italicised. Please be consistent. Remove “events” after the last, for consistency.*

We corrected for that.

(46) *In last sentence of page 1, we find “ – as they are needed in general for leading indicator analyses –”. This seems superfluous and can be removed without any loss of information.*

Removed.

(47) *Add the abbreviation “BL” in the abstract.*

We added BLE in the whole manuscript.

(48) *In page 4 we find both American English in “favored” and British English in “favourable”. Best choose one style only.*

We read through the manuscript and adapted the language to our best knowledge to British English.

(49) *There is a mixed used of “hydro plant” and “hydroelectric power units”. Stick to one style.*

We use now consistently “hydroelectric power units”.

Response to Referee 3

(1) *In this manuscript it is argued that statistical early warning signals (EWS) can be obtained from measurements of frequency fluctuations in the AC power net preceding a power disruption. It is argued that such a disruption can be modelled as a noise-induced tipping similar to standard transitions in stochastically driven bifurcating dynamical systems.*

The Referee is right that the stochastic noise contributions play a crucial role in the analysis. Around 15:40:09.00 the fast scale process’s noise increases. Nevertheless, we want to point out that we do not argue for one specific route to destabilization, i.e, pure B-tipping or N-tipping. The overall analysis rather suggests a complex interplay of changing dynamics on the slow and the fast scale of the aggregated swing equation (ASE) (cf. following comments) which are related to different tipping risks. Apart from the increasing noise amplitude, the loss of the McNary power units is resembled in a decreasing restoring force (cf. Figure 1) which is in favour of B-tipping. Furthermore, the analysis reveals that cascading failures cannot be seen as smoothly destabilizing B-tipping events as argued in previous studies [9, 18]. Instead, individual events can have severe impact in a short period of time, e.g., the mentioned loss of the McNary units, or they are stretched over longer periods. Real world examples for the latter are indicated by the continuous positive drift slope trend after the McNary loss because of insufficient damping and the continuous negative trend over the restoration interval’s

green shaded period.

Note that in the following we changed the order of the Referee’s comments below for a more streamlined argumentation. Of course, all comments are included. However, we prefer to start with the Referee’s major suggestions.

(2) I recommend that the authors, firstly, specify which bifurcating or non-bifurcating model is relevant for describing the observed power grid. (I am not an expert).

We believe that the Referee’s advice largely improves the quality of the article. We include much more theoretical background of related frequency and power grid models on various descriptive levels and in close relation to the NAWI real world scenarios analysed in the main article to adequately address the Referee’s suggestion. In particular:

1. We sketch the (N)BLE briefly in the Introduction and outline the theoretical model motivation of its use for analysing power grid frequency time series ω . Briefly, the aggregated swing equation (ASE)

$$\dot{\omega} = -\bar{\gamma} \cdot \bar{\omega} + \bar{\sigma} \bar{\xi}(t) \quad (1)$$

with the center of inertia or bulk frequency $\bar{\omega}$, restoring rate $\bar{\gamma}$, and a noise process $\bar{\xi}(t)$ with amplitude $\bar{\sigma}$ can in a first approximation describe the power grid frequency in stable operation. However, the linear approximation leads to diverging frequencies for $\bar{\gamma} > 0$ without an alternative stable state which is not observed in reality. Furthermore, real frequency data hint to nonlinear terms present in the drift. An example are frequency signals from the Ireland grid [19], but also the analysed NAWI pre-outage time series (cf. SI S6). Compared to the (N)BLE drift parameterisation, the nonlinear drift derivative ζ and the (N)BLE noise level σ are thus closely related to the restoring rate $\bar{\gamma}$ and the noise level $\bar{\sigma}$ of eq. 1, respectively.

2. The equation can be derived from the mesoscopic, i.e., less coarse-grained, classical swing equation (CSE) that models power grids as networks of coupled nonlinear oscillators. The CSE is mainly used to study self-synchronisation and topological effects in power grid dynamics. In this description, nodes represent basically generators and consumers connected by graph lines which mirror electric transmission lines, such as the Keeler-Allston line. In contrast to the macroscopic ASE frequency description, the mesoscopic CSE can directly involve bifurcation- and noise-induced transitions. Since the ASE and CSE are closely related we discuss their connections in more detail in SI S4. To this end, we employ a simple mesocale CSE model which reproduces the observed BLE and NBLE signatures of the real world scenario including the the pre-outage region (with increased fast scale system stress), the Keeler-Allston line tree-related high impedance fault (THIF), and the loss of the McNary power units with subsequently insufficient damping. Note that the model is oriented on the documented conditions and real cascading episodes outlined in the approved report and kept as simple as possible. In that way, we yield a better understanding of how the (N)BLE estimation of the ASE is related to disturbances from the lower descriptive scale of the CSE in the context of the historic NAWI cascading failure (cf. following answer). The results strengthen the interpretation in the main article.
3. Finally, we also remark that, if needed, the ansatz of the (N)BLE drift-diffusion parameterisation is rather flexible and straightforward to be adapted to specific systems in other fields or might be applied in a black box fashion to systems that lack a proper model description.

(3) Secondly, they should specify, which part of the fluctuations can be modelled as a drift and which part can be modelled as stochastic. (the h- and the g-functions).

That is valuable feedback and we guess that the comment partially implies that Infobox 1 is not introduced well. Therefore, we address this comment twofold. Firstly, we have revised the Infobox explanation and include more details, such as different possible levels of coarse-grained model descriptions in which the states \underline{x} might be represented. Furthermore, we describe more precisely its relation to the frequency ω . Furthermore we state in the revised version:

1. It shall predominantly explain the basic concepts on an intuitively accessible level. These conceptual descriptions cover the studied system (power grids), related components, and the modelling thoughts (Langevin equation).
2. The starting point of our estimation is the Langevin equation for macroscopic observables which can be derived from the system components’ microscopic dynamics, e.g., by the well-established Mori-Zwanzig

formalism [20, 21]. This results generally in a separation of resolved slow deterministic dynamics and fast dynamics which are described in a stochastic fashion instead. We include a remark on that in the revised version.

3. Furthermore, in the revised manuscript we emphasize that we model the frequency dynamics and not individual power grids' components. Nevertheless, the components are related to the frequency dynamics which is the mentioned macroscopic observable. The rare case of a well-documented cascading failure like the one under study is a valuable direct source to better understand such relations. Moreover, please note that the contributions to the drift h and diffusion g depend on the resolution of the time series that we model. We include a remark on this in the manuscript. In particular, we study the frequency dynamics on a second to subsecond scale on which the stabilizing dynamics, such as primary control, and grid disturbances as the Keeler-Allston line THIF live.

Secondly in this time domain, we want to answer the question what is contributing to which function from two perspectives:

1. From a formal point of view, the drift $h(\omega(t), t)$ models the trend components of the studied data, i.e., the force that attracts the frequency towards its stationary reference value. The residuals of that trend correspond to the remaining fast stochastic dynamics $g(\omega(t), t)\Gamma(t)$.
2. From a more technical point of view, the contributions relate to specific grid components. There is no precise microscopic model that can state for a given time series resolution which component will go into the deterministic and stochastic part and we highlight in the manuscript that, e.g., Infobox 1, is only an approximative illustration of the modelling ideas. However, please note that these approximations are not taken out of the blue. They are supported by the analysis of the real world NAWI power outage and previous studies [12, 22]. In our opinion, this makes the NAWI outage an exceptional example of how real world disturbances can affect the drift and diffusion of the power grid frequency and their time evolution. In particular, the discussed primary control of power units lives on a time scale of seconds and the loss of the McNary power units directly goes into the estimated drift slopes. Additionally, the restoration of the physical grid components and resupply of the customers influences the deterministic part. That is why we tend to assign the traditional power units to affect mostly the deterministic part for the given time series resolution. At the same time, several studies deal with the strong influence of the much faster and unpredictable power supply of renewable energy sources (cf. ref. [13] as an example). For that reason, we approximately attribute them to the stochastic part. In the end, most of the consumer decisions take place randomly on fast time scales. An example might be the extensive use of air conditioning because of the heat wave prior to the considered power outage or the mentioned high energy exports due to good conditions for the hydroelectric power units. They might be related to the much higher noise level of the pre-outage time series compared to the post-outage time series which covers mostly the morning of the next day in which the weather and trading conditions might already differ from the past day. Further evidence is provided by the mesoscale NAWI outage model based on the CSE which is closer related to individual components (but still a significant simplification) (cf. SI S4). Therein, we show that increased mesoscopic noise levels are captured by the (N)BLE noise levels. Furthermore, the Keeler-Allston line's THIF model adds further insights into the contributions to $h(\omega(t), t)$ and $g(\omega(t), t)$. The mesoscale noise amplitude of the CSE model remains constant, but the line's transmission properties change, i.e., the coupling between two nodes. In consequence, the NBLE noise level of the ASE increases. That means a modified signal propagation on lower scales can affect the coarse-grained ASE $g(\omega(t), t)$. We denote that as a relative noise increase which quantifies the dominance of fast scale fluctuations from lower scales in the ASE. In contrast, the load imbalance of the McNary units is captured by $h(\omega(t), t)$ which is also supported by the NAWI model. Not least, in the very end of the pre-outage interval, we find increasing drift slopes and noise levels. The report states that signal oscillations were not sufficiently damped by the equipment directly before the outage. Following equation 1, the missing damping on the less-coarse grained CSE level goes into the restoring rate $\bar{\gamma}$, i.e., $h(\omega(t), t)$ of the ASE. However, on the descriptive level of the ASE, it also affects the noise level $g(\omega(t), t)$, since slow and fast signal contributions are less damped. These additional results provide a more complete picture of the ongoing processes during the cascading failure and help to better interpret the (N)BLE metrics related to different model scales as provided by the ASE and CSE. We modified the respective paragraphs and added several sentences and remarks to clarify that we understand the discussion as an illustrative and approximate reasoning which is supported by our empirical findings.

(4) Finally, they should specify what is the Bayesian part of the analysis. (It seems to me that the "credibility bands" are mere statistical "confidence bands", which is also fine, but for a mathematically oriented paper to be

relevant to the reader, it must be precise and correct).

We thank the Referee for the constructive comment. We addressed this issue by including an overall much more detailed description of the method in the revised Methods section and a brief sketch in the introduction. Briefly, the employed Markov Chain Monte Carlo sampled joint parameter posterior distribution allows for the derivation of percentiles from the drift slope and marginal noise level distributions, such that we end up with Bayesian credibility bands. We added also some information in the Method section to clarify that.

(5) The paper is obscure, with a mixture of qualitative (inaccurate) descriptions of electric power grids, poor descriptions of the mathematics of tipping scenarios and specific description of the North American Interconnection power blackout in 1996.

We agree that there were several points to streamline the article’s presentation. We followed carefully the suggestions of the referees to improve the overall quality of the manuscript in this regard. We hope that this addresses the Referee’s criticism conveniently. For example, we emphasize in the manuscript that the general description of power grids is simplified and mainly used for illustration purposes. The goal is to provide the necessary basic understanding (fixed frequency over the whole grid for balanced power input-output, basic ideas of Langevin model) to follow the analysis of the real world example which is basically the main topic of the article. The description of tipping scenarios follows mainly the common sense of practically oriented research articles in the field (cf. [6, 7]) which is why we neglected going into further mathematical detail. However, we include some references which deal with the description of tipping phenomena in a more formal fashion (e.g. ref. [3]).

(6) Infoboxes 1,2: There is very obscure connection between the stochastic differential equation (“Langevin model”) and the diagram below in (a). The cartoons with mountains and houses are really not helpful for describing the transitions in a potential. Pdf’s in b.2 seems to be taken out of the blue and the time-independent potential $V(x)$ in the text is an integral of an explicitly time-dependent drift function $h(x(t),t)$.

As outlined in previous comments we largely improved the description of the theoretical background and related models for the (N)BLE analysis of the NAWI cascading failure. A detailed description on the fine-grained level of the diagram with several individual components, which exhibit complex dynamics on their own, and connected layers of different grids might be out of reach at the current state of scientific knowledge and would probably test the limits of currently available hardware. However, such a detailed description is just not the idea of the Langevin ansatz. The strength of the Langevin description is to reduce the overall modelling effort by keeping the most relevant part of the dynamics. Therefore, the equation in Infobox 1 should be seen as summarising “headline” of the otherwise too complex microscopic dynamics. We explain this better in the revised version. Nevertheless, our specific analysis of a real world event provides deeper understanding of how the single components may go into the deterministic and stochastic contributions of the Langevin model. For example, the power units and restoration actions of the physical grid act on the deterministic scale, whereas the short circuit of the Keeler-Allston line influences mainly the stochastic part of the better fitting NBLE. These real world findings for the NAWI outage are further confirmed by first modelling attempts on differently coarse-grained scales (cf. SI S4). This fills at least partially the gap between individual components and the macroscopic Langevin model of the grid frequency. Concerning the description of the transitions in a potential, we refer to several examples which use similar illustrative explanations. In the manuscript we adapt them to the specific power outage and the involved mixed tipping scenarios. We are conscious about the figurative style of our versions which are adapted to the overall illustrative diagram style. We agree with the Referee that they do not resemble reduced mathematical figures. For this reason, we decided to place them apart into Infoboxes. The simple reduced figures of this type are well-known and we cite related articles. In that sense, a reader can also stick to the reduced versions of other articles. The Referee is right that the pdfs do not resemble a specific scenario. Since the Infoboxes aim to give an intuition for the dynamics and mathematics rather than precise derivations, the pdfs are reasonably designed to fit the N-tipping description in the potential landscape of Infobox 1(d). We added a sentence to avoid confusion about that fact. At last, we thank the Referee for the hint to the missing notation of the time dependence of the potential which is added in the revised version. Basically, this time dependence is mirrored by the rolling windows of the estimation method.

(7) Figure 3 shows (quite arbitrary) sets of simulations of pitchfork and fold bifurcation scenarios with constant and changing noise intensities. The EWS are calculated without mentioning of running windows, with “credibility bands”, without specifying anything about the Bayesian-model part of the BL-estimation (BL meaning Bayesian-Langevin). It is very difficult to get any useful information from this figure.

The Referee is right about the missing hint to the rolling windows and the Bayesian method details in the introductory part of the article. We added those in the introductory part and the figure caption. Please note that Figure 3 is now placed in SI S1. We restructured this part to complement the (N)BLE comparison to AR1 and STD by the related parametric Ornstein-Uhlenbeck estimator (OUE) suggested by Referee 1. Following the comments of the referees, we also include a more detailed description of the methods in the Method section. We refer to the detailed description in the appropriate positions in the introduction and the figure captions. The main advantages of the data-driven parametric BLE in contrast to relying exclusively on the AR1, STD statistics or the OUE are as follows:

1. The parametric Langevin measure gives quantitative estimates of the drift slope and the noise level. Based on that, the distance of the drift slope to zero is a quantitative measure of resilience that is comparable to results for data of equal time resolution from a congruent system.
2. The parametric Langevin measure is able to resolve the different dynamics of the four examples, whereas the statistical measures AR1 and standard deviation are not able to do so.
3. Furthermore, the statistical leading indicators are not applicable in the example data of Figure 1(b) due to an ambiguous trend in the standard deviation.
4. The OUE is much less robust and exhibits significantly higher deviations from the ground truth than the BLE counterparts. Not least, they diverge for AR1 $\rho_{\tau_1} < 0$ (cf. SI S10)

(8) *The descriptions in Section 2.1 of the conceptual models is poor. Eqs. (5) and (8) does not describe noise intensities as shown in Figure 4 f, i.*

We thank the referee for the careful vision of the mathematics. Indeed, we forgot to adapt the t -dependence in the second part of the intervals. The issue is solved by correcting the dependence to $(t - 1000)$.

(9) *Statements as in lines 250-252: “Note also that for monitoring uprising tipping risk factors the exact estimates are much less important than their relative time evolution which encodes the desired information.” Is completely obscure to me, and this is just an example.*

We are sorry for the misunderstanding. We modified and complemented the respective formulations and added precise definitions of the chosen wording in the introduction:

1. We denote leading indicators based on significant positive trends (such as AR1 ρ_{τ_1} and STD $\tilde{\sigma}$), but without employing a parametric model as *qualitative* or *non-parametric* [4–7] ones.
2. Alternative measures which rely on parameterisations of the data generating processes to derive reliable resilience measures are denoted as *parametric* [3,23] leading indicators. For example, (N)BLE, OUE.
3. Finally, we denote estimates that deviate from a known ground truth generally as *biased* ones throughout the article.

What we intended to say was the following: The commonly discussed leading indicators AR1 and STD are essentially qualitative measures for an increasing tipping risk, i.e., there is no threshold and only a weak connection to a parametric model (like the previously mentioned OUE [3] (cf. also SI S1)). Instead analyses with these indicators have to rely on trend changes without definition of a destabilization threshold. These trend changes are termed “qualitative” in the manuscript. In contrast, the Langevin estimation is shown to yield quantitatively unbiased results in the examples with precisely defined destabilization threshold of zero in the case of Markovian dynamics (cf. ground truth vs. estimates in Figure 2). It also reconstructs a reasonable trend in non-Markovian cases, i.e., the measures become comparable to the qualitative reasoning of statistical leading indicators AR1 and STD. This holds for drift slope and noise level estimates apart from the noise amplitude in the B-tipping scenario which we already discussed separately in the original version of the manuscript. In addition, we present the advanced NBLE procedure which yields unbiased estimates also for the non-Markovian datasets, i.e., it clearly outperforms the basic statistical measures. We added the expansion of the method in the corresponding Method section and the quantitatively correct results for the non-Markovian examples in figure 2 of the revised version.

(10) *Other examples from the conclusions: “Nonetheless, the qualitative evolution of the stability and the noise level is also possible if the assumption is violated which is an important implication for many real world*

applications”. (line 492) I simply cannot make sense of such a sentence.

This goes back to the answer to the comment above. We corrected the imprecise wording accordingly.

References

- [1] Martin Heßler. antiCPy. <https://github.com/MartinHessler/antiCPy>, 2021.
- [2] Martin Heßler. antiCPy’s documentation. <https://anticpy.readthedocs.io>, 2021.
- [3] Andreas Morr and Niklas Boers. Detection of approaching critical transitions in natural systems driven by red noise. [arXiv](https://arxiv.org/abs/2010.12345), oct 2023.
- [4] Annelies J. Veraart, Elisabeth J. Faassen, Vasilis Dakos, Egbert H. van Nes, Miquel Lurling, and Marten Scheffer. Recovery rates reflect distance to a tipping point in a living system. *Nature*, 481(7381):357–359, dec 2011.
- [5] Vasilis Dakos, Stephen R. Carpenter, William A. Brock, Aaron M. Ellison, Vishweshha Guttal, Anthony R. Ives, Sonia Kéfi, Valerie Livina, David A. Seekell, Egbert H. van Nes, and Marten Scheffer. Methods for detecting early warnings of critical transitions in time series illustrated using simulated ecological data. *PLoS ONE*, 7(7):e41010, jul 2012.
- [6] Marten Scheffer, Jordi Bascompte, William A. Brock, Victor Brovkin, Stephen R. Carpenter, Vasilis Dakos, Hermann Held, Egbert H. van Nes, Max Rietkerk, and George Sugihara. Early-warning signals for critical transitions. *Nature*, 461(7260):53–59, sep 2009.
- [7] Marten Scheffer, Stephen R. Carpenter, Timothy M. Lenton, Jordi Bascompte, William Brock, Vasilis Dakos, Johan van de Koppel, Ingrid A. van de Leemput, Simon A. Levin, Egbert H. van Nes, Mercedes Pascual, and John Vandermeer. Anticipating critical transitions. *Science*, 338(6105):344–348, 2012.
- [8] Niklas Boers and Martin Rypdal. Critical slowing down suggests that the western greenland ice sheet is close to a tipping point. *Proceedings of the National Academy of Sciences*, 118(21), May 2021.
- [9] Eduardo Cotilla-Sanchez, Paul D. H. Hines, and Christopher M. Danforth. Predicting critical transitions from time series synchrophasor data. *IEEE Transactions on Smart Grid*, 3(4):1832–1840, dec 2012.
- [10] Daqing Hou. High-impedance fault detection — field tests and dependability analysis. <https://api.semanticscholar.org/CorpusID:142456396> (Retrieved: 24 October 2023), 2015.
- [11] John Goodfellow and Paul Appelt. How trees cause outages. <https://www.eci-consulting.com/wp-content/uploads/2017/10/How-Trees-Cause-Outages-ECI-Research-Summary.pdf> (Retrieved: 24.10.2023), 2023.
- [12] Benjamin Schäfer, Christian Beck, Kazuyuki Aihara, Dirk Witthaut, and Marc Timme. Non-gaussian power grid frequency fluctuations characterized by lévy-stable laws and superstatistics. *Nature Energy*, 3(2):119–126, January 2018.
- [13] Andreas Ulbig, Theodor S. Borsche, and Göran Andersson. Impact of low rotational inertia on power system stability and operation. *IFAC Proceedings Volumes*, 47(3):7290–7297, 2014.
- [14] Prabha Kundur. *Power system stability and control*. McGraw-Hill, New York [u.a.], [nachdr.] edition, 2007.
- [15] S.G. de Boer. The lamperti transform. applications to stochastic local volatility models. mathesis, TU Delft, 2020.
- [16] Don Watkins (BPA), Bernie Alfonso (SDGE), Dean Athow (WAPA), Don Badley (NWPP), Jim Beck (NCPA), Sharon Blair (BPA), Mark Bonsall (SRP), Steve Brockschink (COE), Jim Burns (BPA), Gene Byars (NERC), Paul Carrier (DOE), Douglas Cave (BCHA), Bill Comish (WSCC), Jon Daume (BPA), Robert Dintelman (WSCC), Jim Eden (PGE), Christine Elliott (NWPP), Michael Flores (TEP), John Forman (TANC), Kevin Graves (WAPA), Dharam Gupta (PAC), Mark Hanson (IPC), Bob Harshbarger (PSPL), Jack Kelley (SCL), Eric Law (PG&E), Lou Leffler (NERC), Eddie Lim (SMUD),

Mike Lopez (SCE), Wayne Mays (PGE), Al McCuen (CEC), Don McInnis (NERC), Bill Mittelstadt (BPA), Bernie O’Connell (PAC), Brad Osburn (WAPA), Leslie Padilla (SDGE), Vern Parry (COE), Dan Sauter (IPC), Ron Schelberg (IPC), Joe Seabrook (PSPL), Bob Smith (APS), Bob Stuart (PG&E), Brian Theaker (LDWP), Yu Wang (PAC), and Scott Waples (WWP). Western Systems Coordinating Council. Disturbance Report. For the Power System Outage that Occurred on the Western Interconnection. August 10, 1996. 15:48 PAST. Approved by the WSCC Operation Committee on October 18, 1996, oct 1996. The text source without Appendices was provided by Bonneville Power Administration via a Freedom of Information Act request thanks to James King (FOIA Public Liaison, BPA) and Brian Roth (FOIA Case Coordinator, BPA). The Appendices 2,3,5,9 of the timeline were provided thanks to Mary Schaff from the Washington State Library, currently operating under the Secretary of State, Steve Hobbs, via mail correspondence with the "Ask a Librarian". (Washington State Library, Point Plaza East, 6880 Capitol Blvd. SE, Tumwater, PO Box 42460, Olympia WA 98504-2460, Phone: (360) 704-5200, Email: askalibrarian@sos.wa.gov). A low quality scan of the restoration frequency time series from the approved report’s Exhibit 10 was provided by WECC, a readable version from the preliminary report by Jeanie Fisher from the Seattle Municipal Archives (Seattle Municipal Archives, 600 Fourth Avenue, Third Floor, Seattle, WA, 98104, PO Box 94728, Seattle, WA, 98124-4728, Phone: (206) 684-8353, Email: archives@seattle.gov). The contact with Jeanie Fisher of the Seattle Municipal Archives was established by Mary Schaff.

- [17] H. Haken. Synergetics: Introduction and Advanced Topics. Physics and astronomy online library. Springer, 2004.
- [18] P Hines, E Cotilla-Sanchez, and S Blumsack. Topological models and critical slowing down: Two approaches to power system blackout risk analysis. In 2011 44th Hawaii International Conference on System Sciences. IEEE, 2011.
- [19] Ulrich Oberhofer, Leonardo Rydin Gorjão, G. Cigdem Yalcin, Oliver Kamps, Veit Hagenmeyer, and Benjamin Schäfer. Non-linear, bivariate stochastic modelling of power-grid frequency applied to islands. arXiv, June 2023.
- [20] Yen Ting Lin, Yifeng Tian, Marian Anghel, and Daniel Livescu. Data-driven learning for the mori-zwanzig formalism: a generalization of the koopman learning framework. arXiv, jan 2021.
- [21] Eric Darve, Jose Solomon, and Amirali Kia. Computing generalized langevin equations and generalized fokker–planck equations. Proceedings of the National Academy of Sciences, 106(27):10884–10889, July 2009.
- [22] Benjamin Schäfer, Leonardo Rydin Gorjão, G. Cigdem Yalcin, Ellen Förstner, Richard Jumar, Heiko Maass, Uwe Kühnapfel, and Veit Hagenmeyer. Microscopic fluctuations in power-grid frequency recordings at the subsecond scale. Complexity, 2023:1–13, March 2023.
- [23] S. R. Carpenter and W. A. Brock. Early warnings of unknown nonlinear shifts: a nonparametric approach. Ecology, 92(12):2196–2201, dec 2011.

Revision II

We thank Referee 1 for spending time on the manuscript.

Response to Referee 1

(1) The revision responds to several of my requests. In particular, the authors now describe their statistical method advertised in the abstract in more detail on p16 (second half) to p18. They also clarify what the Bayesian updating mechanism iterates over. They ignored my mathematical nitpickings (that one strictly speaking cannot have a bifurcation in these systems with Gaussian noise), but that is less relevant for NatComm.

We thank the Referee for the differentiated feedback. We kindly hint to Comment (8) of Revision I, in which we discussed the mathematical nitpickings mentioned by Referee 1. Of course, we are open for further discussions.

(2) The revision makes much clearer that the paper considers as its main contribution the dissemination and advertisement of (N)BLE to the scientific community. The discussion of the power grid outage, the Greenland Ice Sheet and the cyanobacteria light stress experiments by Veraart et al (the latter two added in the revision) serve merely as demonstration examples picked from the available pool of high-profile debates on tipping. The new title, now containing the two disparate examples, confirms this.

This goes back to a misunderstanding on our side. Based on Comment (2), we thought Referee 1 would not be convinced about the generality of the approach. Therefore we analysed further examples. However, in the second revision, these examples are not focused on because, aside from the key advantages of the (N)BLE over standard leading indicators, the power outage analysis is a second equally worth contribution of the article as already emphasized during the first revision. We provide some examples:

- “However, we want to point out that the article is not only intended to propose the (N)BLE to a broader audience. [...] We claim that, apart from providing a robust statistical time series analysis tool with potential wide range applicability, there are several other important implications.” (Comment (1) from Referee I). After this, we list several points connected to power grids and N-tipping.
- “In particular, the frequency data turned out to be a rather convincing application example for the expanded version, the NBLE.” (Comment (3) from Referee I)
- “We stay with documenting the Methods in the end, since, as stated above, they are not the main focus.” (Comment (4) from Referee I)
- “Nevertheless, we consider the real outage analysis with a direct comparison to real events as the key results.” (Comment ?? from Referee II)

(3) The proposed (N)BLE method is undoubtedly an improvement over the common indicators such as one-step autocorrelation (AR1) or variance (VAR) of the time series in terms of systematic time series analysis. (N)BLE fits the time series data to coefficients of a Langevin equation such that its outcome are probability distributions for coefficients in this equation, namely for the restoring rate "zeta" toward the equilibrium and the amplitude (std) "psi" of the internal or external fluctuations assumed to influence the dynamics in the Langevin equation. The fitting procedure is Bayesian updating using Markov Chain Monte Carlo (MCMC) sampling. The only cause preventing the scientific community from using the proposed method (instead of AR1 and VAR) is lack of acceptance or awareness.

The (N)BLE is a significant improvement for several reasons. The NBLE incorporates correlated noise, which is a key advantage compared to the common indicators. The estimation schemes of these standard metrics do not allow to include correlated fast-scale processes that are evident across various applications from different fields, such as in Refs. [1–6] .

Additionally, we thank Referee 1 for his proposal of comparing the Ornstein-Uhlenbeck estimation (OUE) to the Bayesian Langevin estimation (BLE). The OUE breaks down in synthetic and in real-world examples due to the oversimplified linear drift used in the OUE. Additionally, where it can be computed, it is heavily biased, and partly diverges to $\pm\infty$ (cf. SI S9 for the OUE and SI S1 for AR1 and STD).

Furthermore, we kindly ask Referee 1, whether the second argument regarding “lack of acceptance or awareness” is really crucial or rather of relative importance for the following reason. Early warning signs as AR1 and VAR are discussed in high-profile journals. These approaches combine statistics like AR1 and VAR, rolling window

approximations, and complex system dynamics, which are basic concepts from statistics, econometrics [7], and nonlinear dynamics [8], respectively. In other words, the novelty of the (N)BLE’s ingredients behaves similar to that of the above-mentioned statistics, while the algorithm itself is a novelty, including undoubtedly a significant improvement over the common leading indicators. Therefore, we would like to offer another perspective: For several reasons, our advanced methods, such as Bayesian estimation via Markov Chain Monte Carlo sampling, including nonlinearities and correlated fast-scale dynamics for anticipating critical transitions, are not introduced before. This is likely not only due to lack of awareness. The suitability of the approximations and estimation formalisms must be established. Furthermore, the implementation of these methods is comparably complex. For example, a serial version can easily result in computation times from hours to days. In contrast, in our manuscript we provide implementations that terminate in seconds to hours. Our technically advanced implementations come along with detailed documentation and open-source availability. Furthermore, inclusion of correlated noise is mathematically complex and an enhancement of methods that are only recently available [9, 10].

(4) I have two major concerns, both leading me to recommend not publishing the article in Nature Communications. (1) Being a methodological improvement over reporting of AR1 and VAR is a low bar to cross for a time series analysis method. The authors demonstrate that (N)BLE has (far) less bias than AR1 and VAR for synthetic data from examples with known ground truths. However, the examples in the main article are Langevin equations such that there is very little (or no) model bias.

We point out that we consider a twofold model misfit in the synthetic examples: fast-scale dynamics with time-dependent amplitudes (process y) and multiplicative coupling in x . In the framework of stochastic differential equations (SDE), these model misfits are noteworthy. However, there are reasons to stay with SDEs as benchmark models. First, they provide a powerful heuristic description of stochastic time series across various disciplines. Second, they provide a theoretical foundation for the commonly applied leading indicators, such as OUE, AR1, and variance. Third, using a completely different modelling framework for methodological tests would be ill-advised as the BLE is applied to data, where a heuristic Langevin description is reasonable. This said, the estimates are perfectly unbiased, although we introduce a noteworthy bias. Furthermore, even smaller bias than that strongly limits the applicability of the commonly used OUE as shown in the SI examples. Furthermore, the (N)BLE has not only far less bias than AR1 and VAR, but there is no estimation bias at all in the examples considering the (N)BLE’s soundly defined credibility intervals.

(5) Demonstrating less bias when fitting with a more complex model is in itself not a significant step toward “Quantifying tipping risks” as claimed in the title.

We believe, this is due to a misunderstanding. The different risks of bifurcation- and noise-induced tipping (B- and N-tipping) are addressed by the Langevin equation’s drift and diffusion components, respectively. Furthermore, the title follows the common wording in the community’s literature. For example, we cite a PNAS article’s title “Quantifying resilience in humans and other animals” [11], which considers indicators based on critical slowing down such as AR1 and STD. These statistical measures are not even transformed into the OUE. In this qualitative reasoning, they remain very loosely connected to restoring rates. Furthermore, they cannot address noise level estimates, such as the comparably robust (N)BLE. Therefore, we believe our manuscript’s title to be well justified both by its content and by the common wording in the field.

(6) At no point in the paper do the authors provide an estimate for the tipping risk.

The (N)BLE provides a robust estimate of the restoring rate ζ and its distance to zero is approximately a quantitative measure of resilience. This is mathematically sound [8, 12].

(7) For “noise-induced tipping”, which this paper focuses on, this estimate could be provided in the form of a rate (in the spirit of Kramers’ escape rate): the probability of tipping per time unit. So, the title is misleading: the paper proposes a method to estimate the restoration rate and the noise amplitude of an assumed Langevin model for a time series. The method has much less bias than widely used but basic indicators when tested on Langevin models. No tipping risk is quantified.

We agree with Referee 1 that the wording “risk” could be understood in this way. This goes back to the misunderstanding described in Comment (5). Using “risk” in terms of B- and N-tipping risk factors is intended (cf. the article’s main text). An alternative title may be “Quantifying destabilizing factors in...”. We are open for further suggestions.

The B-tipping risk is quantified by the distance of the (N)BLE’s restoring rate estimate $\hat{\zeta}$ to zero. That is mathematically sound [8, 12].

Additionally, the (N)BLEs noise level estimates $\hat{\sigma}$ and $\hat{\Psi}$ relate to N-tipping probability in multistable systems. This is demonstrated by the synthetic x_g example. Moreover, the Referee’s proposal of the Kramers’ rate is already included in the original draft’s Section 3. The issue of Kramers’ rate estimation in the absence of data from various stable states of a multistable system is an unsolved task in the numerical sciences. However, with the (N)BLE we provide a promising pathway for further enhancements in this direction and make first steps with the submitted manuscript. Additionally, there is no doubt that fast-scale dynamics significantly changed during the outage cascade, which is not quantifiable with standard approaches.

(8) The proposed (N)BLE method still contains a large number of ad-hoc choices such as...

We kindly state that no choice is unmotivated or “ad-hoc”, but well justified for the research task. Therefore, we answer point-by-point:

*(9) * degree of polynomial in Langevin equation*

To our knowledge, this is already answered in the original manuscript: “We develop $h(x, t)$ into a Taylor series up to order three which is sufficient to describe the normal forms of simple bifurcation scenarios.”

Moreover, the parameterisation is directly motivated for the power grid’s frequency data based on the aggregated swing equation:

1. Furthermore, the third-order polynomial drift better fits the real frequency dynamics (cf. SI S4).
2. The noise of the frequency data is intrinsically correlated, which is reflected by the NBLE model. In particular, compare the green dashed line of the NBLE to the blue ground truth in the lower left plot in Figure S12. Note that no standard leading indicator, including the OUE is able to include such noise types in a straightforward manner.

*(10) * choice of Langevin equation as base model (or LE+OU for NBLE)*

The Langevin equation is a well justified starting point for stochastic time series from various nonlinear systems, since the foundation of Synergetics by Hermann Haken [12].

In particular, we hint to the original article: “It would be worthwhile to have the opportunity to monitor changes in both resilience and noise levels simultaneously at a given time resolution to effectively control the system, better understand the frequency dynamics before power outage events, and, in rare cases, possibly avoid power outages. In principle, a model based on a Langevin equation

$$\dot{\underline{x}}(\underline{x}, t) = \underline{h}(\underline{x}(t), t) + \underline{g}(\underline{x}(t), t)\underline{\Gamma}(t), \quad (1)$$

with the drift $\underline{h}(\underline{x}(t), t)$, the diffusion $\underline{g}(\underline{x}(t), t)$ and stochasticity $\underline{\Gamma}(t)$ is able to do so. It provides access to B-tipping risk factors which are included in the drift term and N-tipping risk factors which are modelled by the diffusion term for a given time scale resolution.”

And furthermore we cite the first revised version: “The hidden process may be interpreted in terms of red noise $y(t) = \Gamma_{\text{red}}(t)$ if it lives on a faster time scale τ_y than the observed process x or alternatively as a slow external driver y of the observed process x on the faster time scale τ_x .”

In the manuscript, we employ a model with correlated fast-scale dynamics because we demonstrate in SI S15 that it significantly better describes the data.

*(11) * priors for polynomial coefficients enforcing an intercept of the right-hand side*

No prior enforces an intercept. Zero is always included within the prior ranges $[-50, 50]$ and $[0, 50]$ for the drift and diffusion coefficients, respectively.

*(12) * fixed time scale separation between hidden OU process and observed process in NBLE.*

We included the missing explanation already in the first revision: “Furthermore, we account for stability issues and ambiguities in the estimation scheme, discussed in Willers and Kamps [9], in two ways: [...] and second, by an a priori assumption of a notable time scale separation of the processes x and y with the time

scales τ_x and τ_y .”

(13) * *window sizes (which have a severe effect on the results as reported in several graphs)*

This is due to a simple misunderstanding. We never emphasized a crucial role of the window size and clarified that in the article. The window size only plays a role for distinction of permanent and artificial trend changes in the BLE results. A numerical artefact is characterised by a stepwise change *over one window length* in the drift slope ζ or noise levels because incorporation of an outlier disturbs the estimation procedure. However, the key results are *permanent* (which means significantly longer than one window length) and therefore robust.

(14) *I am not sufficiently expert on statistical time series analysis to judge if the choices made by the authors improve the state of the art in this area (and not just AR1 and VAR estimates), and I am not sure if NatComm is the appropriate venue to evaluate improvements of statistical estimation methods.*

The choices made must be adequate for the research task, i.e., the data analysed. This is ensured by the choices explained above.

(15) (2) *The writing style of the original article was criticised by all reviewers as too unfocussed, loaded with unnecessary tangents. The revision has improved some aspects, but made other aspects worse.*

For example, the revision now advertises a different method than the original one. Now the method fits a model with a non-Markovian term. It also includes another demonstration using the Greenland Ice sheet melt rates, requiring a new round of background explanation. The paper now introduces and discusses the distinction between non-parametric (qualitative) and parametric estimations. The authors call, for example, the AR1 autocorrelation estimate non-parametric, even though (by its definition) the estimate models the time series’s underlying process as an AR-1 process and estimates the one-step memory coefficient. So, the model is just simpler than the Langevin model fitted by (N)BLE and the distinction is artificial. These changes in focus and scope confirm that the aims of the paper are still evolving rapidly and are not yet in a state settled enough to be reviewed with confidence.

Most of the criticized changes addressed concerns of the Referees during the first revision or were based on misunderstandings on our side. Referee 3 wanted a more detailed definition of “qualitative” and “quantitative”. Furthermore, Referee 1 claimed that we should rather compare the OUE to the (N)BLE instead of AR1 and STD, which was a good point. Referee 1 is right regarding the AR1 definition, which can be seen as coefficient of an autoregressive model. We approached the AR1 from the statistical perspective. In statistics, the AR1 is equivalent to the autocorrelation function at a lag $\tau = 1$. Therefore, we termed it a non-parametric estimator, since the autocorrelation function can be estimated without assuming a specific model. However, since the term was not precise we added a clarifying sentence to SI S9. Additionally, we significantly shortened and clarified the presentation.

Other presentation deficiencies are still present:

(16) * *The text spends space introducing tipping rather poorly: Infobox (1) is full of details that play no role in the analysis. It contains metaphorical landscapes symbolising a potential, but what forms the barrier for tipping (the metaphorical hill in (b,d)) is never explained: what is the hypothesized positive feedback effect that decreases the restoring force for larger deviations? The presumed goal of the paper is “quantifying tipping risks”, so the height, shape and location of this barrier needs to be estimated. However, the corresponding quantities are not discussed at all anywhere. Instead the pictures distract the reader with little suns (?) in the landscape.*

We agree with Referee 1 that the Infoboxes may cause distraction from the main findings. During the second revision, we significantly shortened the text and moved the Infoboxes from the main text to the SI to highlight the key contributions. As stated in several answers during revision I, the potential picture is a common illustrative intuition in the field (cf. [13,14]).

(17) *The discussion admits that the power grid failure cascade was triggered by a few discrete events, resulting in discontinuous changes for which estimates of restoring forces and Gaussian Noise amplitudes are inappropriate.*

The referee summarizes one of the key contributions in our article, which presents a crucial correction of previous results. These previous results characterized cascading failures to be accompanied by rather continuous

destabilization trends in statistical leading indicators [15–18].

Moreover, the estimates are only not appropriate in the small intervals including such discontinuous changes. However, before and after such changes, the estimates reflect the dynamics of the stationary aggregated swing equation. Thus, a comparison of the resilience and noise level is indeed mathematically sound and enables a first intuition for the connection between real-world events and modelling of the frequency dynamics.

(18) * *Axis scaling in graphs is different between closely related figures. For example, in Figure 1 the estimates for ζ_{NBLE} differ by three orders of magnitude ($\zeta \sim -3$ vs $\zeta \sim -2 \times 10^{-3}$) between c (pre-) and f (post-outage), contradicting the text which states that zeta returns to pre-outage levels.*

We are sorry for the article’s wording that might have caused some misunderstanding. The referee is right about the different axis scales. These are due to different time resolution of the time series in the pre- and post-outage region, which was already stated in the *original* draft: “The scaling differences of the BL-estimates from the frequency time series $\delta[\omega_{\text{P}}](t)$ and $\delta[\omega_{\text{R}}](t)$ in figure (a–c) and (d–f), respectively, are caused by the significantly different time resolution and quality of the data.”

Furthermore, we never stated that the drift slope ζ returns to pre-outage levels, but that it reaches a new stable plateau when the *physical* pre-outage state is approached. *Physical* means the configuration of the lines, power plants, and loads is resumed. To emphasize this fact, we cite the first revised version: “After the load restoration, the system approaches steadily the physical pre-outage configuration and the drift slope reaches a new stable plateau in the end of the green shaded period.”

However, since the sentence caused a misunderstanding, we added “...the physical pre-outage configuration, including lines, power plants, and loads,...” during the second revision.

(19) * *The overly extensive description of the power outage event sequence does not advance the main message of the paper. This was criticized already in the original version.*

We accounted for that by moving all details to the SI. In this way, the side information is easily accessible for more technically interested readers.

(19) * *The interpretation of the graphs obtained in Fig1 and Fig3 is vague when translating the results into discussions of tipping risk. This is natural because the quantities obtained are not sufficient to estimate tipping risk, but it leads to hard-to-follow speculative writing.*

To our understanding, the writing style and argumentations are well supported by Bayesian reasoning. In this framework, we interpret the results relying on the available data, theoretical models, and background information [19] (such as those from the disturbance reports). The BLE results for the blackout are well justified by the aggregated swing equation and mesoscopic modelling attempts of key outage episodes using the classical swing equation (cf. SI S3).

Furthermore, we transparently communicate the left uncertainties. To illustrate our perspective, we provide an example from the original draft: “There is a non-vanishing probability, that the contact between the Keeler-Allston line and the tree was established around 15:40:09.00. [...] Though, it seems to be plausible that the changing dynamics are directly related to the actual Keeler-Allston line tripping by a THIF, we cannot deduce with certainty a causal relation between the state change and an emerging THIF.” The abbreviation THIF refers to the tree-to-line contact.

Reviewer #1 (Remarks on code availability):

(20) *The code is well documented and not overly complex. I have no concerns that it works to the specification given in the methods section of the paper.*

We thank the referee for examining the code as well. Additionally, we kindly hint to our answer to Comment (3) for Referee 1 during the second revision to highlight further advantages of the code documentation and availability. The implemented methods provide far more than the simple serial versions of the presented algorithms. Code and documentation of the object-oriented estimation schemes including strong parallelisation result in roughly 3700 lines. The package is therefore not only a backup for the study, but enables researchers in the field to notably benefit from several advantages of the (N)BLE compared to the standard leading indicators without losing time due to complex programming and coding tasks.

References (Revision II)

- [1] Michael Schulz and Manfred Mudelsee. Redfit: estimating red-noise spectra directly from unevenly spaced paleoclimatic time series. Computers & Geosciences, 28(3):421–426, 2002.
- [2] István Matyasovszky. Estimating red noise spectra of climatological time series. Időjárás (Quarterly Journal of the Hungarian Meteorological Service), 117:187–200, 04 2013.
- [3] Zhengyu Liu, Peng Gu, and Thomas L. Delworth. Strong red noise ocean forcing on atlantic multidecadal variability assessed from surface heat flux: Theory and application. Journal of Climate, 36(1):55–80, January 2023.
- [4] Christian Franzke. Long-range dependence and climate noise characteristics of antarctic temperature data. Journal of Climate, 23(22):6074–6081, November 2010.
- [5] Meng Gao, Xiaoyu Fang, Ruijun Ge, You-ping Fan, and Yueqi Wang. Multiple serial correlations in global air temperature anomaly time series. PLOS ONE, 19(7):e0306694, July 2024.
- [6] Andreas Morr and Niklas Boers. Detection of approaching critical transitions in natural systems driven by red noise. Physical Review X, 14(2):021037, June 2024.
- [7] Eric Zivot and Jiahui Wang. Modeling Financial Time Series with S-PLUS, volume 2. Springer-Verlag New York, 2006.
- [8] Steven H. Strogatz. Nonlinear dynamics and chaos. With Applications to Physics, Biology, Chemistry and Engineering. Westview Press, second edition edition, 2015.
- [9] Clemens Willers and Oliver Kamps. Non-parametric estimation of a langevin model driven by correlated noise. The European Physical Journal B, 94(7), jul 2021.
- [10] Clemens Willers and Oliver Kamps. Efficient bayesian estimation of a non-markovian langevin model driven by correlated noise, 2022.
- [11] Marten Scheffer, J. Elizabeth Bolhuis, Denny Borsboom, Timothy G. Buchman, Sanne M. W. Gijzel, Dave Goulson, Jan E. Kammenga, Bas Kemp, Ingrid A. van de Leemput, Simon Levin, Carmel Mary Martin, René J. F. Melis, Egbert H. van Nes, L. Michael Romero, and Marcel G. M. Olde Rikkert. Quantifying resilience of humans and other animals. Proceedings of the National Academy of Sciences, 115(47):11883–11890, October 2018.
- [12] H. Haken. Synergetics: Introduction and Advanced Topics. Physics and astronomy online library. Springer, 2004.
- [13] Marten Scheffer, Jordi Bascompte, William A. Brock, Victor Brovkin, Stephen R. Carpenter, Vasilis Dakos, Hermann Held, Egbert H. van Nes, Max Rietkerk, and George Sugihara. Early-warning signals for critical transitions. Nature, 461(7260):53–59, sep 2009.
- [14] Marten Scheffer, Stephen R. Carpenter, Timothy M. Lenton, Jordi Bascompte, William Brock, Vasilis Dakos, Johan van de Koppel, Ingrid A. van de Leemput, Simon A. Levin, Egbert H. van Nes, Mercedes Pascual, and John Vandermeer. Anticipating critical transitions. Science, 338(6105):344–348, 2012.
- [15] P Hines, E Cotilla-Sanchez, and S Blumsack. Topological models and critical slowing down: Two approaches to power system blackout risk analysis. In 2011 44th Hawaii International Conference on System Sciences. IEEE, 2011.
- [16] Eduardo Cotilla-Sanchez, Paul D. H. Hines, and Christopher M. Danforth. Predicting critical transitions from time series synchrophasor data. IEEE Transactions on Smart Grid, 3(4):1832–1840, dec 2012.
- [17] Hui Ren and David Watts. Early warning signals for critical transitions in power systems. Electric Power Systems Research, 124:173–180, jul 2015.
- [18] Frank Ehebrecht. Anticipation of critical transitions in complex systems. Master’s thesis, Westphälische Wilhelms-Universität Münster, 2017.
- [19] Wolfgang von der Linden, Volker Dose, and Udo von Tussaint. Bayesian Probability Theory. Applications in the Physical Sciences. Cambridge University Press, 2014.

Revision I

Introductory Notes

First of all, we thank the three Referees for spending their time on the manuscript, for their differentiated judgments and the detailed constructive comments. We start with a general comment: In the course of the revision process, we were able to compute results with an extended version of the Bayesian Langevin estimation (BLE) which accounts for a second hidden Markov process in the data. It is better suited to treat non-Markovian time series of processes driven by correlated noise as the synthetic test cases in Figure 2(g-l). We call it the non-Markovian Bayesian Langevin estimation (NBLE). In the figure, it is shown in red with green credibility bands. Based on a statistical comparison of the suitability of the BLE and NBLE parameterisation to describe the bus voltage frequency time series (cf. Supplementary Information (SI) S15) we decided to show the NBLE results instead. Nevertheless, the overall results of both methods are largely comparable and do not notably affect the results (cf. Figure S13). Only the slight negative BLE drift slope trend might be better explained by an increasing BLE model misfit which we thus discuss in the SI instead. Furthermore, In the following, we discuss all concerns to our best knowledge.

Response to Referee 1

(1) What are the noteworthy results? The manuscript proposes an Early-Warning Indicator (EWI) and applies it to test examples and a times series from historic frequency data around a power grid blackout 1996. I see the proposal and testing of the new method, a Bayesian estimator for a generic Langevin model (called BL estimation in the paper), as the main noteworthy novelty of the manuscript. Proposing and demonstrating feasibility of a computational/statistical method may not count as a result or discovery in the sense considered by Nature's science journals. However, the authors claim that the BL estimate is a significant improvement of classical EWIs (such as time-step autocorrelation, variance) that were introduced in high-profile articles in Nature, PNAS, Science etc (refs 1,7,9,17,18,54) are repeatedly and recently discussed there (see [R1,2023] for a striking recent example in Nat. Comm.). Hence, a strong argument can be made that a significant improvement of EWI estimation should be published to the same audience. In my view the proposed BL estimate is certainly a methodological improvement over the above-mentioned classical EWI's. However, the current version does not convince me yet if the improvement is significant. [R1] Ditlevsen, P. and Ditlevsen, S. (2023). Warning of a forthcoming collapse of the Atlantic meridional overturning circulation. Nature Communications, 14(1), 1-12.

We agree with Referee 1 that a significant improvement compared to the statistical EWIs would be a strong argument for publication. In this spirit, we add new results to Figure 4: Additionally to the Markovian estimator which is biased in case of the non-Markovian time series, we have made major enhancements, restructured the manuscript, and provide several additional results:

1. We add new results to Figure 2: Additionally to the Markovian Bayesian Langevin estimation (BLE) which is biased in case of the non-Markovian time series in (g,j), we add a stable numerical non-Markovian version (NBLE) which cancels out the bias completely (cf. red estimates with green credibility intervals in Figure 2). The NBLE theory is provided in Material and Methods 4.1. It is also implemented in the provided open-source Python package and well-documented [1, 2] to be easily applicable by independent researchers. Since we show in SI S15 that the NBLE fits the power grid frequency data better than the BLE, we modified the presentation in the main article accordingly.
2. In the following, we consider the comparison to AR1 and variance. Inspired by the Referee's comment (1) and (10), we made several additions considering the approximation of the data via an Ornstein-Uhlenbeck (OU) process and its relations to AR1 and variance V . We found these derivations of the Referee's comment (10) again in an article by Morr and Boers [3] and call the approach OU estimation (OUE). Since in our opinion, the comparison significantly improves the quality of the manuscript, we want to give the discussion adequate space. However, we need to keep in mind the length of the main article. For this reason, we added two new sections in the SI and summarize the results in the main article with references to the corresponding SI sections. Additionally, we claim that it is important to distinguish two lines of argumentation in the context of EWIs. The Referee's proposal of the OUE as well as the (N)BLE consider specific parameterised models. However, there are several articles which avoid the assumption of any specific parameterisation [4–7]. Therefore, we define “parametric” and “non-parametric” or “qualitative” indicators in the main text. In a next step, we moved the comparison of the BLE to the qualitative non-parametric measure to SI S1 to address also the non-parametric school of reasoning. However, in SI S10 we perform the suggested comparison of the OUE and the BLE. The results show that the OUE yields strongly biased estimates over the four synthetic examples. Furthermore, the

OUE results diverge to $\pm\infty$ in almost one half of the time interval in three out of four test cases due to an $\text{AR1}\hat{\rho}_{\tau-1} < 0$ which is, where the assumption of an OU process breaks down.

3. We demonstrate that this result is not an artefact of pathological synthetic examples. Quite in contrary, the application of the OUE to Greenland Ice Sheet Melt rates [8] in SI S10 and S16 reveals that these problems can occur in potential real world applications.
4. Finally, in anticipation of the Referee’s comment (2), we demonstrate the widespread potential of the (N)BLE for other fields of research by the analysis of the further real world examples. The most prominent one, the analysis of Greenland Ice Sheet (GrIS) Melt rates is placed in the main article Section 2.3 because, apart from being an application example, the performed analysis has interesting implications. The BLE results provide further evidence for the previously claimed destabilization of the GrIS (cf. ref. [8]), but also suggest a common driver of increasing fast-scale dynamics in the late 20th century. Furthermore, a quantitative comparison of the resilience levels over melt rates from a coastal and a continental location hints to an ongoing assimilation process. Following this hint, the dynamics and stability of the the main GrSI might tend to slowly approach the ones of coastal lower regions. In Section 2.3, we briefly discuss also further examples to outline the widespread potential and rather general applicability of the (N)BLE. Apart from the GrIS melt rates, we analyse a biological living system, i.e., a bacteria culture’s collapse due to increasing light stress, documented in SI S5. The experiment was originally performed by Veraart et al. [4]. The drift slope yields a significant destabilizing trend. A last example is only briefly mentioned and we integrate it in the response letter, because it is better placed in another manuscript in preparation. The BLE indicates also the destabilizing trend from an ancient greenhouse earth state into an icehouse earth state (cf. response letter, Figure A1). In summary, we demonstrate by our additions that the (N)BLE has real world application potential from paleoclimatology and climate science to technological and biological/ecological systems like power grids or the bacteria culture experiment.

However, we want to point out that the article is not only intended to propose the (N)BLE to a broader audience. (This is, why we decided to sketch the method briefly in the Materials and Methods section (which we corrected in the revised manuscript according to the Referee’s concerns, cf. later comments)). We claim that, apart from providing a robust statistical time series analysis tool with potential wide range applicability, there are several other important implications:

1. The method does not focus on the deterministic restoring rate ζ alone. Instead, we start by giving several synthetic examples, where the method is able to extract information about both the deterministic and the stochastic part which is connected to bifurcation induced and noise induced tipping mechanisms. (Note that we answer the concerns about the distinction in comment (9)) These information are not reliably extracted by simpler methods like the OUE, and not even accessible if we stay in the non-parametric reasoning of AR1 and variance. The importance of changing noise levels is moreover evident in three real world examples, namely, the pre-outage NAWI frequency data and the GrIS melt rates.
2. The presented results relate the (N)BLE measures to real world power outage scenarios and the interpretation of the modelling approach. Most of the literature about power grid analysis treats network dynamics and synthetic models. Only a few articles (for example [9]) follow the idea of summarizing power grid dynamics by a macroscopic key observable like the bus voltage frequency. This is directly inspired by the practical engineers’ and system controllers’ point of view. However, these works propose that power outages might be accompanied by a rather smooth increase in statistical EWIs. In consequence, they transmit the impression, these phenomena would be mostly caused by a bifurcation in the macroscopic observable’s state space. In our article, we demonstrate that things are much more complex and, in particular, not as smooth as suggested by previous studies (not least due to the significant novelties concerning the metadata, commented on in detail in SI S11). An outage may appear in a data-driven modelling approach entirely caused by a bifurcation, by noise, or by a mixture of both. These theoretical considerations about the potential shape and the grid topology are supported, e.g., by the strong increase of the drift slope $\zeta_{\text{N}BLE}$ when the McNary units are lost in the second red interval of Figure 1(b) and the negative drift slope trend towards the physical pre-outage state in Figure 1(b). Moreover, we complement the analyses by mesoscale modelling attempts and considerations of theoretical frequency models in the added SI S4. This further confirms our reasoning and deepens the overall understanding of our results. Finally, the results are obtained from empirical real world data of a major outage with the possibility of comparing the results directly to real world events which is a very rare case study to our knowledge. This assessment is supported by Referee 2.
3. The advantage of a more reliable noise level estimation and a more quantitative comparability of the estimates is further supported by the GrIS analysis.

Figure A1: BLE analysis of the greenhouse-icehouse transition in a $CaCO_3$ proxy time series.

4. Not least, the (N)BLE is rather straightforward to adapt to specific modelling needs in different fields.

(2) Will the work be of significance to the field and related fields? How does it compare to the established literature? If the work is not original, please provide relevant references. The method as proposed would have widespread relevance for providing evidence of potential tipping from observations in climate science, ecology, social science and complex engineering systems.

We fully agree with Referee 1 regarding the widespread potential of the proposed algorithm. Please note that we answered to this comment already in comment (1). Briefly summarised, we added several analyses of real world examples, ranging from paleoclimatology, climatology, to technological and living biological systems to the manuscript to confirm the widespread applicability of the (N)BLE.

(3) Regarding originality: the abstract uses the word "propose a BL approach", leaving open how original this is. The description of the method is only brief and comes at the very end. It refers to [12] (however, [12] does not employ a Bayesian approach). The numerical ingredients of the BL estimate (estimating probability distributions sampled using MCMC for coefficients of a polynomial model with additive Gaussian noise) may not be new, and in the article the authors do not claim that this part is novel. However, introducing a rigorous statistical method into science has value.

We thank the Referee for the correct evaluation regarding originality. We do not claim the single ingredients of the procedure to be new, but can confirm that the composed algorithm, is our original work. The formulation "propose a BL approach" shall not doubt that originality, but underline that we introduce it for the first time in the context of real power grid resilience analysis (and further examples in the revised manuscript). In particular, the frequency data turned out to be a rather convincing application example for the expanded version, the NBLE.

(4) Does the work support the conclusions and claims, or is additional evidence needed? There are severe problems with the presentation: the paper spends space introducing and discussing tipping, has a detailed introduction of the test examples, followed by an extensive discussion of the behaviour of the BL estimate on these test examples, then a minute description of events and data at the 1996 blackout (using power grid specific jargon), without ever describing the BL estimation proposed. The term is used throughout the paper without explanation or context until the brief description in "Methods".

After reading the manuscript again under the impression of the Referee's comments we fully agree about that point. Following the Referee's objections, we decided to reorganize the manuscript: As already explained above, we relocated the discussion of the introductory synthetic datasets including AR1, variance and OUE to the SI to shorten the introduction. The gained space is used to sketch the (N)BLE in the introduction and refer to the majorly advanced Materials and Methods section. We stay with documenting the Methods in the end, since, as stated above, they are not the main focus. This streamlines the presentation.

(5) The description in "method" makes various ad-hoc choices that are not discussed (the supplementary material would have been the place to do this): for example (and apart from the Bayesian priors), it includes 2

higher-order expansion terms of the right-hand side beyond the linear term, while only the linear term is used. The linear term is referred to as ζ in all figures but as θ_1 in the method. A Bayesian approach iteratively incorporates additional data to update the current belief about the probability distribution of an unknown. The paper does not describe what is iterated over. Is it time? That is, are the priors assumed to be the coefficients at the earliest time and then the measured time series points for increasing times are "fed into" Bayes' theorem to update the distribution, such that we obtain time dependent coefficients simply because of the iteration process? Or do we have a prior distribution of time-dependent coefficient functions $\theta_i(t)$ and then we apply all time series data points at once? This makes a difference: in the former approach the estimates at the beginning of the times series are more strongly influenced by the prior, while in the latter approach all estimates are influenced equally.

We understand that the description of the method is too brief in the end of the article. Therefore, we provide further details to avoid misunderstandings. In particular, we summarize the included details following the points of Referee 1:

1. The Bayesian part of the method refers to assuming the parameters θ_i to be variables, whereas the measured data $x(t)$ are assumed to be fixed observations. Additionally, we start from Bayes theorem which in principle accounts for prior information. Both points are explicitly based on non-frequentist statistics' reasoning.
2. The priors are chosen to be almost uninformative: Only the Gaussian priors for the higher order coefficients $\theta_{2,3}$ are slightly informative. They only express that the chosen prior boundaries of these parameters should be assumed to be a priori less probable. The almost uninformative priors lead to a higher coincidence of the maximum posterior and a maximum likelihood estimation. However, the weak prior assumptions are chosen for good reasons which we also include in the method description. In most real world cases, only poor prior information about the system are available, contraindicating strong prior assumptions. One example is the exact model for the frequency dynamics. Nevertheless, note that the framework allows for easily including prior knowledge if available for the system at hand. In our case, we prefer to not restrict the estimation procedure in favor of too strong prior assumptions.
3. It holds $\zeta \neq \theta_1$. We are sorry for the misunderstanding due to the short description of the method. Since we assume to start our observations in a fixed point x^* , the drift term is Taylor expanded around this fixed point x^* . Under these circumstances, we expect that $x - x^* < 1$ and $(x - x^*)^4 \ll 1$ so that truncating the expansion after the third order is justified. Taking into account these higher order terms tends to stabilize the method also for moderate to high noise levels. A linear stability analysis of the Taylor expanded drift term h_{MC} results in

$$\zeta = \left. \frac{dh(x)}{dx} \right|_{x=x^*} = (\theta_1 + 2\theta_2 x + 3\theta_3 x^2)|_{x=x^*}.$$

4. Referee 1 is right. There are numerous possibilities for the iteration procedure and initialization of the priors. We simply iterate over the rolling windows in time without updating the priors from window to window for several reasons (however, the principle implementation would allow to do so with some adaptations): One of the most straight forward incorporations of prior knowledge would be to center the prior probabilities of the current window around the estimated values of the previous one. However, since we expect a state change we do not want to introduce a prior bias of "no change". Furthermore, the results in the manuscript illustrate that no stronger priors are necessary in our analyses, since they are rather smooth, i.e., the estimations of subsequent windows converge nearly to the same values. In our approach the coefficients are estimated with all data per window at once. The time dependence of the parameters θ_i follows from the locally stationary rolling window approach.
5. We appreciate the Referee's idea of updating the estimation scheme by incoming data points. Indeed, we also tested such an implementation. A disadvantage for our purpose is that we expect the coefficients to change over time (since we expect a transition). This makes it necessary to assume an explicit time dependence of the parameters. In consequence, the algorithm becomes much more rigid with regard to the chosen time dependence model. Not least, the resulting estimation is numerically notably less stable.

(6) The power grid example, also does not fully support the claim of prediction of tipping. The narrative suggests that the BL estimates $\hat{\zeta}$ detect an event that occurred at 15:40 (start of first red interval in figure 5). If that change of $\hat{\zeta}$ and $\hat{\sigma}$ had any predictive significance for the following spike at 15:42 is less clear to me (the

authors claim that). However, a similar severe subsequent change of $\hat{\zeta}$ and increase of $\hat{\sigma}$ at 15:43 did not lead to a spike.

We appreciate the constructive doubts and are sorry about the confusion, probably caused by comparison with statistical EWIs. We agree with Referee 1 that the example does not support the claim of tipping prediction, but emphasize that we do not use the word "prediction" in the direct context of our results for good reasons. The manuscript title also underlines that we focus on a quantitative inference of tipping risks in complex systems, i.e., the changes of the system dynamics such as the restoring force/drift slope ζ (related to B-tipping) and the internal noise level σ (related to N-tipping).

The official triggering of the cascading failure is the Keeler-Allston line tripping around 15:42, caused by a tree-to-line contact. The peak at 15:42 marks only the actual line tripping. However, the tripping is caused by the line touching a tree which leads to a short circuit due to grounding. This is a process which does not happen instantaneously. First of all, the line has to elongate due to increasing heat, probably caused by heavier loads in the grid. Otherwise, the line would not hit the vegetation. Secondly, even if a tree-to-line contact is established, it is not permanent. The electric current changes already, but the progressing carbonization of the woods completes the permanent contact. These thermo-chemical processes take place in order of seconds up to a few minutes [10, 11]. This is the reason why it is likely that the permanent state change detected by the BL estimation is related to the actual line tripping event. Nevertheless, Referee 1 is right to question the increase of the metrics around 15:43. It is not related to the dynamics of the system, but has numerical reasons: The missing estimates in the beginning of Figure 5 account for the rolling window size. If this is compared to the increase of the metrics around 15:43, it becomes obvious that the increase of the metrics is caused by including and excluding the tripping spike about one window length. In contrast, the dynamical change around 15:40 is persistent and independent from the window length which makes it very likely to be a significant state change. We clarified these points in the revised manuscript.

(7) *The strongly increased variance of the frequency time series pre-outage (std increased by a factor of ≈ 10), compared to the apparently standard variance (see post-outage) is a strong EWI on its own.*

We thank Referee 1 for the accurate observation, included a remark in the manuscript, and used the information in the modelling SI S4. However, we guess that the Referee hints to the (N)BLE noise level $\hat{\sigma}$, not the variance $\hat{\sigma}^2$. We checked the variances of both data as well. The variances on interval 1 and 3 (cf. Figure S14) are almost equal ($3.865 \cdot 10^{-5}$ and $3.721 \cdot 10^{-5}$, respectively). Nevertheless, we also hint to the fact that the two empirical time series are taken from very different sources (analogue paper scan vs. digital measurement) with completely different time resolutions. Differences between the datasets might be (partially) also caused by that fact.

(8) *Are there any flaws in the data analysis, interpretation and conclusions? Do these prohibit publication or require revision? The authors refer to bifurcations for systems with noise, even though it is not clear in which sense they mean this. For example, for their test example Eq(2), Crauel [R2] showed that additive noise destroys the pitchfork bifurcation: one has a single globally stable attractor for all parameters. [R2] Crauel, Hans, and Franco Flandoli. "Additive noise destroys a pitchfork bifurcation." *Journal of Dynamics and Differential Equations* 10 (1998): 259-274.*

We thank the referee for the interesting comment and enjoyed reading the recommended article. The authors of the reference stick to a very formal distinction of a "static" and a "dynamical" bifurcation. They write:

No two (not even random) regions of the state space stay physically distinguishable for all times. The "dynamic" bifurcation is destroyed. However, the "static" bifurcation may be argued to survive. The distribution of the "random attractor" a_α , (which moves in a stationary way) has a one-peak density for $\alpha < 0$ and a two-peak density for $\alpha > 0$.

Apart from the classical static bifurcation (i.e., the emergence of two stable states and destabilization of the previously stable one) they define the dynamical bifurcation to be characterized by two close initial conditions of different sign which converge to a fixed distance from each other. It makes sense that this does not hold for noisy systems.

However, the referee is right that we did not state in which sense we mean it. We include the reference and hint to the static bifurcation defined therein. Stating this, Crauel et al.'s dynamical bifurcation definition which is compromised by noise does not touch the basic reasoning of our study example in Figure S1. Moreover, in Figure 2(a) we consider one single realisation with one single initial condition in the static regime of the double-peaked state probability density (PDF) without tuning the control parameter α . In that case, there

is no bifurcation included in the simulations, but only the random jumps between the two branches of the double-peaked PDF. We hint also there to the double-peaked PDF and the recommended reference.

(9) *The introduction spends time expounding on the importance of distinguishing between a change in decay rate in the "deterministic part" of the (assumed-to-exist) Langevin model and a change of the noise amplitude. However, this distinction is artificial as every change in the noise amplitude can be transformed into a change of the deterministic part (as long as the noise amplitude does not go to 0, see [R3]). [R3] <https://math.stackexchange.com/questions/4673542/transformation-of-ito-diffusion-such-that-the-diffusion-coefficient-becomes-cons>*

We thank the reviewer for the interesting comment. We remark that we have strong theoretical background for the assumption of a Langevin description of the macroscopic frequency dynamics. We hint to several references [12–14] and included an SI section S4 in which we further outline the relationships of macroscopic and mesoscopic descriptive levels for power grids.

Anyways, indeed, the Lamperti transform [15] is designed to normalize the variance of a stochastic differential equation (SDE) to one. Nevertheless, this normalization transformation is mainly used to

1. facilitate and stabilize numerical simulations of arbitrary SDEs,
2. successfully employ, e.g., Kalman filters for statistical inference,

and does not touch the formal nature of the Langevin equation. The original Langevin equation describes Brownian motion, i.e., the apparently random movement of a particle in a fluid due to random collisions with the molecules around. In the original scenario, the macroscopic velocity change of the particle is given by the deterministic Stokes friction and the stochastic random collisions. This distinction is an important feature of our statistical inference approach because it considers two time scales relative to the resolved macroscopic bus voltage frequency time series. This time scale separation of the model in a slow deterministic, macroscopic evolution and fast stochastic microscopic contributions is not touched by the Lamperti transformation. In particular, applying the Lamperti transformation

$$\Lambda(x(t), t) = \int \frac{1}{g(x(t), t)} dx$$

to the model parameterization

$$dx = h(x(t), t)dt + g(x(t), t)dW = (\theta_0 + \theta_1 x + \theta_2 x^2 + \theta_3 x^3)dt + \sigma dW \equiv h(x)dt + g_{\text{const.}}dW$$

leads to the transformed dynamics of the variable Z :

$$\begin{aligned} dZ &= \left(\frac{\partial \Lambda}{\partial t} + \frac{\partial \Lambda}{\partial x} h(x) + \frac{1}{2} \frac{\partial^2 \Lambda}{\partial x^2} g_{\text{const.}}^2 \right) dt + \frac{\partial \Lambda}{\partial x} g_{\text{const.}} dW \\ &= \left(0 + \frac{h(x)}{\sigma} + 0 \right) dt + 1 \cdot dW \\ &= \frac{1}{\sigma} (\theta_0 + \theta_1 x + \theta_2 x^2 + \theta_3 x^3) dt + dW. \end{aligned}$$

The above equation could be used for the Bayesian inference instead, but the coefficients θ_i and the noise level σ are present as before. Furthermore, summarizing the variable in terms of $\tilde{\theta}_i = \theta_i/\sigma$ only hides the information about the noise level in the new parameters $\tilde{\theta}_i$, but does not cancel it in terms of the Langevin equation's separation of deterministic and stochastic dynamics.

(10) *The argument that autocorrelation and variance of the time series do not permit to conclude separately about input noise amplitude σ^2 and decay rate $-\zeta$ is also not quite correct. Assuming that in each sliding window the time series is approximately governed by an Ornstein-Uhlenbeck process, the one-step autocorrelation AR1 determines ζ through $\zeta = (\log(\text{AR1}))/\Delta_t$ (Δ_t is the time step) and the variance V determines the σ^2 through $\sigma^2 = -2V(\log(\text{AR1}))/\Delta_t$. The estimates $\hat{\rho}$ and $\hat{\sigma}$ in Figure 3 look consistent with the above expressions. So, Figure 3 should plot the decay rate and noise amplitude implied by AR1 and V , and compare that to the quantities determined by the BL estimate. (They may still be less accurate, which would demonstrate superiority of the BL estimate.)*

We thank the referee for the valuable suggestion and followed his comment. Please note that we already brought the answer forward in comment (1).

(12) *Is the methodology sound? Does the work meet the expected standards in your field? The analysis methodology is sound as far as I could judge. The presentation is poor: the article is too long, spends too little space on the main method/result proposed in the abstract, and far too much on tangents (general tipping background, details of the power grid event).*

We agree about the points and made already several notes to corrections. We briefly summarize them again.

1. The significantly shortened the introduction by relocating the comparison to statistical EWIs and the OUE to the SI. Only the valuable main results are stated in place.
2. We sketch the method briefly in the main text to accomodate the reading flow.
3. We include several important details of the method in the corresponding Materials and Methods section.

(13) *Is there enough detail provided in the methods for the work to be reproduced? The code for BL estimation and reproducing the results is available according to references. However, as the description is very short, one would have to apply the code in a black-box fashion without understanding what is happening.*

We confirm the Referee's comment that the method and code is freely available and understand the concerns regarding black-box fashioned algorithms. However, our implementation is not a black-box for several reasons:

1. The whole code is written following best practices (<https://geo-python.github.io/site/notebooks/L1/gcp-1-variable-naming.html>, <https://www.appacademy.io/blog/python-coding-best-practices>) with readable class, function, and variable names and completely open to study. In that sense, this is the most detailed documentation of an algorithm on a numerical level which one could think of.
2. Furthermore, the whole package is documented in text form at <https://anticpy.readthedocs.io/en/latest/>. This includes short descriptions of the mathematical methods.
3. The github repository allows for customizing the package to individual needs.
4. Additionally, we include a more detailed description of the numerical procedure in the introduction and Materials and Methods section of the manuscript.

Response to Referee 2

Major remarks

(1) *The manuscript is very well written, but it is simply too long. The 1996 blackout case is discussed too extensively, with a lot of information being presented sometimes thrice! The example is exceptional – as is the work performed here – but it becomes too cumbersome for the reader. I would actually suggest one big addition, which could solve a lot of issues: Both a map of the affected areas, with the 3 and then 4 island formation, with a detailed timeline below, as a long left-to-right arrow, which each marking of the time stamps referred in the text. This would greatly increase the readability of the text – an image is a thousand words here. Overall, the case study needs to be streamlined for readability.*

We thank the Referee for the very constructive comment. We followed the suggestions regarding the time line and the illustration of the islanding process. Additionally, we combine the new figure with the presentation of the main results to streamline the overall results related to the real world event. Dublicate information are cancelled from the main text body.

(2) *As commented above, the manuscript lacks a certain congruence. The introduction focus heavily on tipping, the limitations of the two seminal CSD metrics, then we move to the Langevin model, and later the case-study of the 1996 power outage. The Discussion and Outlook focuses on the case study and steps forward to improve estimation using BL, yet we never arc back to discussing CSD metrics and only a few mentions are given to R-tipping. An outlook on the method is given, with suggestions to the Kramers' escape rate, which is a new topic in this regard.*

After reading the manuscript again, we agree with the Referee's comment and made mainly three adaptations:

1. Since the comparison was intended to briefly motivate our Bayesian Langevin approach (BLE) and the non-Markovian extension (NBLE) (which we additionally consider in the revised version), but occupied too much space, we moved it into the supplementary information (SI) S1 and S10. This balances the length of the introduction and augments the focus on the more important theoretical description of the (N)BLE, the macroscopic power grid frequency model estimation, and the subsequent real world analyses.
2. The conclusion is adapted accordingly and summarizes first the main findings of our study. Afterwards we briefly outline possible future research attempts.
3. Note that we added several further analyses of real world examples ranging from paleoclimatology, climate science, and biological living systems into the main body or SI with respect to the Referees' comments. In particular, the Greenland Icesheet is a prominent further example of increasing noise levels in the main text. For more detailed comments on this we refer to the answers to Referee 1. Moreover, in SI S15, we show that the NBLE yields a better description of the frequency time series. The overall results are not altered at all, but the provided comparison provides valuable additional insides in our opinion. We adapted the results accordingly. Not least, we enrich the SI with naive modelling attempts of the NBLE fingerprints during the NAWI outage (cf. SI S4). The addition addresses comments of Referee 3 concerning the theoretical background. We show that the real world NBLE fingerprints' qualitative behaviour can be recovered in a naive model of three key outage events in a mesoscale model guided by the approved disturbance report [16]. It might be seen as base for further discussion and developments. Nevertheless, we consider the real outage analysis with a direct comparison to real events as the key results.

Major questions

(1) In Eq. 1, we find a vector formulation with underbar x . Strictly speaking, the diffusion \underline{g} is a matrix and not a vector, as it is multiplied by the noise vector $\underline{\Gamma}$. Have the authors chosen the current nomenclature for a particular reason? Furthermore, for consistency, should the left-hand side not read $x\text{-dot}(x(t),t)$ instead of $x\text{-dot}(x,t)$? The authors are careful in including the time dependence of $x(t)$ on each function/vector/matrix on the right-hand side.

We appreciate the precise comment and corrected the notation.

(2) Continuing on this issue. The authors present, in the most general form an N -dimensional Langevin equation, but the work and most subsequent examples are 1 dimensional, as is the case study. The justification for power-grid frequency seems solid, as it is generally almost perfectly identical regardless of where in a power grid it is measured. The work in this manuscript relies heavily on antiCPy, which, to my understanding, is also only a 1 dimensional time series analysis tool. The authors offer a convincing argument to the study of 1-D time series, where the presented N -D Langevin equation might be misleading. It is not straightforward, for voltage analysis across a power grid, to reduce the problem to a 1-D time series. Much less so for climate or ecological studies. Could the authors comment on this? I would push for a less ambitious case herein only a 1-D case is presented.

It is true that we deal with one dimensional time series and antiCPy is designed only for that purpose up to now. Therefore, we adapted the Langevin equation in the introduction to a one-dimensional version depending on ω . However, we have preliminary results in which a two-dimensional BLE version announces the firing spike of a two-dimensional time series from a Neuron Hindmarsh-Rose model. Therefore, we point out in the main body that the basic framework of our estimation procedure—in particular because of the open-source nature of the code—is rather flexible and can be adapted to higher dimensions or specific needs of other research fields. Furthermore, the Infoboxes are intended to provide mainly conceptional information and were intended to be kept as general as possible, but with context to the main topic, i.e., power grids. We hint therein also to $\omega \neq \underline{x}$ to clarify that the power grid is a highly complex system with high-dimensional states \underline{x} which might be treated on various coarse-grained levels. These may be thought to be a solution of an assumed-to-exist N -dimensional lower scale model of a power grid. The crucial step is to outline the partial aggregation of these information in a macroscopic observable which is the frequency ω in this case. We further elaborate on that in SI S4 where we discuss in detail the theoretical justification of the (N)BLE applied to power grid frequencies and relations to less coarse-grained descriptions.

Not least, in course of the revision we added several real world examples which confirm that a heuristic approach of the one-dimensional (N)BLE can provide interesting insights in complex systems over a wide range of research fields (cf. in particular the answer to comment (1) for Referee 1; cf. also further comments to Referee 1). In the context of bifurcation induced tipping, we want to hint to Haken's slaving principle [17], which basically means that fast-scale degrees of freedom are forced to follow slower ones. These slower ones should be long term stable order parameters. The frequency is not the order parameter of the system, but nevertheless a

slow key observable. When the system approaches a bifurcation, the dynamics are progressively slowed down (critical slowing down [6]) which results in increasing dominance of the slow variables due to the slaving principle. Against that background, a high dimensional system implodes and a low dimensional heuristic description becomes approximately correct for observables near a bifurcation if they represent the order parameter of the system or are at least reasonably related to it.

(3) Eq. 19 has two parameters θ_0 and θ_1 on the left-hand side, yet depends only on θ_1 on the right-hand side. Is there something amiss here?

We agree that the notation might be confusing without further explanation. For a straight line $y = ax + b$ in 2 dimensions, the chosen prior is invariant under a variable transformation, i.e., unbiased with regard to the choice of the representation. It can be shown (cf. <https://jakevdp.github.io/blog/2014/06/14/frequentism-and-bayesianism-4-bayesian-in-python/#The-Prior>) that the prior is therefore uniform in b and $a = \sin(\alpha)$ with the angle $\alpha = \tan^{-1}(a)$ between the x-axis and the line y . Uniformity of a in $\sin(\alpha)$ instead of α is a result of the influence of b . In congruence with these considerations, we added some sentences to clarify the notation.

(4) What is the time series Gemini?

We thank the referee for the valuable feedback. It should underline that the presented time series of Figure S1 are rather similar, i.e., "time series twins". However, the question highlights the potential confusion and we cancelled the unprecise wording.

(5) It is hard for me to follow why Eq. 7 diffusion term is referred to a "coupled" noise amplitude. Coupled noise would seem to me to pertain to the non-diagonal diffusion matrix terms of a 2- dimensional stochastic process, which couples noise across the dimensions of a 2-d process. Is there a reason for this choice of wording?

We agree with the referee that the wording might not be precise enough. The wording should only clarify that the noise amplitude σ_{coupled} corresponds to the noise amplitude of the process y which basically "couples" as a correlated noise term into the x dynamics. We changed σ_{coupled} to σ_y to avoid confusion.

(6) For the correlated B-tipping model, the authors refer to Eq. 4 being coupled via the constant $q = 0.5$, but there is no constant q in Eq. 4. Subsequently, the noise term from Eq. 7 is $\sigma = \sqrt{c}$, which is not related to the noise but to the drift term. I find this quite confusing. Could the authors try to alleviate this and include directly an equation for this process?

We appreciate the clear hint and agree that including the whole set of equations for the correlated B-tipping set facilitates the reading flow significantly. The coupling constant q was thought to replace the multiplicative coupling of y into x in Eq. 4. Indeed, the OU process was kept as simple as possible for demonstration purposes with only one independent parameter c for drift and diffusion. However, the method applies independent from that choice as shown by the correlated drift-diffusion varying dataset $x_{g,h}^{\text{corr}}$. Please note also the additional results of the NBLE. It completely cancels out the bias of the hidden OU process in the non-Markovian examples in Figures 2(g,j).

Remarks and smaller questions

(7) In the second paragraph of the introduction, we find "[...] in connection with uprising bifurcation induced transitions [...]". What is the meaning of "uprising" here?

We are happy about the comprehensive minor corrections and English style comments. Indeed, this is basically a wrong wording. It was meant to mean "emerging". However, we improve the reading flow by deleting the word without replacement.

(8) In the third paragraph of the introduction, we find "Furthermore, the currently very important trend to an environmentally sustainable grid architecture will probably augment the amount of noisy grid participants significantly in future". This is a claim that requires a citation. I suggest the authors look at Milano et al. (2018) doi:10.23919/PSCC.2018.8450880.

That is a correct claim and a great literature suggestion. We enjoyed reading the article and included the citation.

(9) *The remark “Nevertheless, industrial and scientific stakeholders only report activations or deactivations up to a certain extent, and railway traffic can be heavily disrupted, for example, due to damage to the railway infrastructure caused by a storm or unforeseen strikes” seems out of place. I cannot follow what is the relevance of this statement in the context of the narrative. Could the author elaborate on this? Maybe best removed? – I would also like to point out to the authors that, to the extent of my knowledge, this might not be the best example to use. Various EU countries operate their train systems and 16.6 Hz instead of 50 Hz. Admittedly the power source is still the same, but for the argument the authors are seeking, there are better examples to pick. (Some Sbahnen are even DC powered).*

Considering the text passage again, we follow the Referee’s advice and removed the example. Originally, the example should illustrate that individual components of the power grid might act on various time scales. In consequence, changes of their dynamics may influence also the grid frequency on different scales, i.e., $h(\omega(t), t)$ and $g(\omega(t), t)$. Since industry and power units are often involved in response and demand agreements, they partially act on pushing frequency deviations back to the reference frequency. At the same time industry might act on faster scales by partially stochastically energy demand. The example’s intention was to clarify that. However, since it led to confusion instead, we reduce the example to industrial consumers and power units. This might suffice for to illustrate the idea.

(10) *The use of the term “mountainside”, although clear, seems like more added baggage. A much more common term – which the authors also use – is “landscape”. This is a term used almost uniformly from psychology to ecology and physics. I suggest the authors stick to “landscape” only.*

We agree with the referee that “landscape” is used over multiple disciplines and is the base for every figurative explanation of critical transitions. However, to our knowledge “mountainside” and “landscape” are not synonyms. In the manuscript, the word “mountainside” is used to refer to the slope of the mountain below the mountain top following the figurative style of the Infoboxes. A quick search in the Oxford English Dictionary supports this meaning (<https://www.oed.com/search/dictionary/?scope=Entries&q=mountainside>):

mountainside, n.

the sloping surface of a mountain below the summit. Also figurative.

Thus, for now we stick to the wording. However, if we misinterpret the English meaning, we appreciate a hint to the correct term.

(11) *Pertaining to the sentence “Note that power outages as well as control mechanisms live on time scales of seconds up to several minutes, e.g. global grid primary control actions in Germany react to a maximum frequency deviation of ± 0.2 Hz in times up to 30 s and have to be stable under full load up to 15 min before the simultaneously starting secondary control in regional grids should replace the primary control completely.” Firstly, this seems to refer to article 31 which seems to point to regulations from 2003. Are these still the statutory limits in Europe/Continental Synchronous area?*

We appreciate that the reference is checked carefully by the referee. It was difficult to find a reliable citation and we were initially happy with that one. Anyways, we checked up again and found a revised version from 2022 which confirms the stated requirements for primary control power units in Germany. Of course, we updated the citation accordingly.

(12) *Secondly, it might be better to refer to the regulation of the Continental Synchronous area in general. Germany is but one of various country in this supergrid.*

That is a good point. Including these information improves the quality of the manuscript. To this end, we read the legal text from the European Union which deals with the regulation of the European Transmission Grid. The guidelines for the Continental Synchronous Area are congruent to the German ones (as it is part of the continental synchronous area). However, we included also information about the interconnected grids with the corresponding citation. All grids’ primary control mechanisms operate in the same order of magnitude regarding frequency deviations and activation times, i.e., ± 0.2 Hz to 0.5 Hz and 10 s to 30 s (cf. main text of the manuscript for more detailed information).

(13) *The (although commonly used) abbreviation “pdf” in the last paragraph in page 2 is not introduced. For consistency, introduce it.*

We agree that the abbreviation is not introduced at the first place in the main text. It was erroneously introduced in the Infobox 1 which is referenced first. The referee is right that this is against the convention of defining the abbreviation in first place of the main text and might not be seen if the reader prefers following the main text only. We corrected the issue by introducing it additionally in the main text and keeping the cumbersome definition in the Infobox, since the boxes are designed to stand alone in the sense of illustrative explanations of the basic mechanisms.

(14) *The sentence “Since a sustainable adaptation of our power grids is indisputable” at the end of page 2, reads weirdly. I believe the authors intend to imply that renewable energies will become ubiquitous. The way it is written implies that there has been a steady adaptation of the power grid, which is not necessarily the same. I believe rephrasing this would bring the point to the readers in a clearer fashion.*

We are glad to correct for the imprecise formulation. The referee is completely correct regarding our intention and we hope our alternative formulation is clearer.

(15) *Comparing the titles of Infobox 1 and Infobox 2, Infobox 1 should include a mention to “tipping”, as this discussed therein.*

We modified the headline accordingly.

(16) *In Infobox 1, Figure 1, we read “[...] be captured by the diffusion $g(x(t), t)\Gamma(t)$ [...]” but further on, e.g., in page 4, bottom, “[...] in principle contribute to the diffusion [sic] $g(x(t)t)$ [...]” (note the missing comma!). In my view, g is the diffusion, not $g\Gamma$. It would be best to be consistent here.*

We agree with the referee that $g(x(t), t)$ only should be denoted as the diffusion and modified the manuscript. The comma is also included. Note that we changed the formulation in the box slightly to avoid the wrong usage of “diffusion”, but keeping the whole term $\underline{g}(x(t), t) \cdot \underline{\Gamma}(t)$ as “intrinsic stochastic dynamics” to stay congruent with the orange color-coding of the Infobox.

(17) *In Infobox 1, “[...] If the restoring force (light green arrow line) given by the steepness of the potential mountainsides is high enough [...]”. I believe the correct English form here is “[...] is large enough [...]”, but do check as I am not a native speaker.*

We checked for English language examples which confirm that the referee is right. We corrected the wording.

(18) *In Infobox 2, “[...] but under stationary noise level if the potential landscape is negatively affected”. I believe is this a bit of an abuse of language. This landscape is not an actual landscape, thus it cannot be negatively affected (in a sense of a loss of vegetation, flora, of fauna). This is still a function, which cannot be “negatively affected”. Maybe there is a better wording to be found here?*

We are glad to suggest an alternative with a more precise wording during keeping the figurative language of the Infoboxes (cf. highlighted manuscript).

(19) *Eq. 7 includes, not erroneously, a noise term dW . I would advise against this. In the current formulation, the authors use x -dot, which implies a $dx(t)/dt$ (more conventionally a physics notation) on the left-hand side. The drift on the right-hand side is correct, yet the final term of the equation is a differential, which is incongruent with the notation. I think it is best to stick with $\Gamma(t)$.*

The referee is right. We changed the notation to $\Gamma(t)$ and included a short hint to the Wiener process assumption $\Gamma(t) = \frac{dW}{dt}$ in the method section. Note also that we basically forgot the denominator in the original manuscript’s notation.

(20) *Eqs. 2, 4, and 6 all have their respective control parameters as $-\nu$, $-r$, and $-\alpha$, yet, their values are also all negative. Not critical, but feels a bit pointless. Would it not be more elegant to just have these are positive values and remove the minus signs?*

That is a good point. Actually, we decided consciously to change from positive to negative notation, since it is convenient for plotting the control parameter trends within the corresponding figures. We thought including

the actual control parameter trends directly in the figure helps a lot to understand what is going on behind the scenes. For example, readers which are less familiar with the topic might expect the whole time series to be stationary. That is, e.g., due to the control parameter changes, not the case. Nevertheless, this information is basically given by the ground truths of the drift slope. Together with the referee’s comments on the figures we decided that it would be best to remove the control parameter plots. Thus, we also changed to the simple positive notation.

(21) Eqs. 2, 4, and 6 all introduce new notation for h and g with various different subscripts, like h_{pf} . This seems superfluous as there are never used in the text. The only terms mentioned are X_g and so on. I suggest alleviating the notation, as there is little gain here in separating h and g . They are all the same mathematical objects, just with different functional values in the various examples.

We are happy for that hint. Indeed, it reduces the notation’s overhead.

(22) Subsequently, the use of the term set in N -tipping set X_g might be misleading. These are not sets, just stochastic processes. Note that the last one is denoted correlated B -tipping model and not set.

Thank you for the valuable comment. The error in the manuscript is due to imprecise wording. We modified “set” to “dataset”, since we denote specific realisations $X_{...}$ of the general stochastic process equations.

(23) There is also an inconsistency in the presentation of Eq. 6 and 7, which give the SDEs directly, instead of the drift and diffusion term. I think it would be more elegant to stick to one notation only and report as in the previous examples.

We adapted the notation accordingly.

(24) In the same line as above, the last example, denoted correlated B -tipping model is not given the honour of an equation, just inline passages. It would be best to also give this model an equation, for presentational purposes.

We thought it might shorten the main text, but we agree with the referee that a systematic presentation has to be preferred. We included the fourth model with explicit equations.

(25) Fig. 4 and Fig. 5 are missing the y-labels for the plots. It is best to include those. Moreover, the sub-plot labels need work: Fig. 4 has no parentheses and the labels are in the subplots, Fig.5 is the opposite. For Fig. 5 I suggest removing the legends entirely and instead include the terms in the y-label, i.e., $\hat{\sigma}$ and $\hat{\zeta}$. The rest can be mentioned in the caption. For Fig. 4 all legends can be removed, $\hat{\sigma}$ and $\hat{\zeta}$ added accordingly in the y-labels, and the rest referred to in the caption.

We are happy to adapt the figures to improve the overall quality. Originally, we have omitted the actual y-labels, since the figures show *different* quantities, e.g., the drift slope ground truth, the drift slope estimate, and the control parameters in one axis. Using one y-label would have been inconsistent. However, we decided to remove the control parameter plots because the crucial information about non-stationarity of the process is already included in the varying ground truths. In that way we achieve consistent y-labels and can also account for the double minus notation of the control parameters which was only chosen because of the space-efficient plot. To solve the inconsistency issue regarding the subfigure labels we propose to keep the positions (since they are the most space efficient ones), but make them identical, i.e., we removed the parentheses and wrote all subfigure labels in bold type. The changes enabled also a more precise usage of the legends.

(26) In section 2.2.1, the authors write “[...] grid components were satellite-synchronized.” I suggest the authors write “GPS-synchronized”. This is more appropriate for it clearly states what timing method was used (if that was indeed the case!)

The approved disturbance report underlines that the Portland Area Dispatch SCADA system was not satellite synchronized. This is the reason given in the report why it cannot be aligned precisely in the time line of events. We agree with the referee that the wording probably refers to the GPS satellite system. However, there is no further explanation or more specific technical term than that included in the approved report. We were not able to find a more specific hint to the actual satellite system. It might also be that a combination of the GPS (USA) and GLONASS (Russia) system was used at that time (even if it might be unlikely due to the vicinity to the end of the Cold War). At least today, private parties have access to both (and more) GNSS systems. The crucial information for the manuscript is that the time stamps are well-aligned without technical issues

which makes the interpretation more robust. Since we are not experts in time synchronization methods, their history, and their technical implementation, we would prefer to keep the same wording as in the approved report.

(27) In section 2.2.1, page 11, the authors include “Additionally, power grids are commonly modeled by coupled Kuramoto oscillators which contain a linear damping term. This damping can be interpreted in reality as control actions in order to stabilize the deterministic dynamics of the system.” This seems misplaced. The narrative is about actual events. The way power grids are modelled is perfectly unrelated to the question at hand.

We agree that the focus of the article lies on the analysis of the actual NAWI power outage event which is a well-documented case to compare individual real world events with the coarse-grained drift-diffusion coarse-grained frequency dynamics. However, in order to address concerns of Referee 3 regarding the theoretical model base we constructed a simple mesoscale model of the pre-outage period, the Keeler-Allston line tripping, and the McNary units with subsequently insufficient damping based on the classical swing equations (cf. SI S4). For suddenly decreased damping as indicated by the approved report due to failing equipment, we observe an increase of drift slopes and noise levels in line with the very end of the real world time series analysis. We accordingly adapted the paragraph and added a reference to the SI, since it provides further insights into relations between different scale models and the NBLE.

(28) In Section 3, page 14, the reference to Kramers’ rate requires a citation, as this might not be known to all readers.

We agree that citations are necessary. We include the Kramers’s original work, a didactic example including a comparison between Markovian and non-Markovian models, and an article that introduces a correction of Kramers’s escape rate for multiplicative noise.

(29) In Section 5, between Eq. 10 and Eq. 11, “of the corresponding Fokker-Planck equation of the time series (a), with the potential”. What is the “(a)” here?

We included the missing reference to the figure.

(30) Below Eq. 17, the authors use “ $p(\cdot)$ ”. Better to be explicit. This notation is used only once here.

We agree that it might be best to avoid the implicit notation in the manuscript. Accordingly, we restructured the respective sentences.

(31) Section 7 should just be part of a Supplemental Material.

We agree with the referee that it is important to define a precise criterion to decide what should be included in the main article and what should be part of a supplementary information. In this case, the outlier correction paragraph is directly related to the presented results of the restoration time series interval. The grey lines show the results without the outlier correction. Furthermore, the outliers are shown in the insets of the figure. We believe that it is more convenient for the reader to include the short paragraph about the outlier correction in the main article to dispel any doubts about the proper treatment of these points. Following the referee’s argument more details are discussed in the supplementary information.

(32) Reference 33 includes various acknowledgements that I believe should be included in the acknowledgements, wherein I hope they are more visible, and thus pay homage to the people involved accordingly.

We are glad for the detailed comments on the reference section and adapt the acknowledgments in the designated section. Nevertheless, we want to point out why it might be reasonable to keep the additional “acknowledgments” also in the reference section: Apart from the fact that without the help of these people the very intricate search for the missing metadata would have been without success after almost two years, the specific references are not easily available online. If researchers want to read them, they might contact us or even the institutions and persons which were involved in the search. This is the most appropriate way to keep the sources accessible to everyone over the coming decades. Of course, we have asked for permission to include the mentioned persons in the acknowledgments and the reference section for this purpose.

Minor issues, typos, and typography

(33) All references to the use of Gaussian kernels for filtering data indicate the bandwidth chosen with a σ , which naturally pertain to the standard deviation of the Gaussian filter. In this particular paper, this adds a bit of confusion, since the noise term in the various models is also a σ , but that pertains to the magnitude of the noise, not the standard deviation of the pdfs of each model! I strongly advise using a different term for one or the other, to avoid confusion.

That is a very good point. We modified the bandwidth σ and σ_{kernel} (there was some inconsistency) to the short and clear notation β .

(34) The use of the dot in the equations feels a bit unnecessary, and sometimes confusing. Eq. 1, which is written in vectorial form, implies a matrix operation on a vector, g on Γ , without the use of a product, which reads perfectly fine for me. From hereon, we find a consistent use of the dot to multiply scalar and functions, which seems excessive for a standard mathematical operation. I suggest the authors remove the dot, apart from the separation of scientific notation for, e.g. in Eq. 3, where we read $2.4 \cdot 10^{-4}$. [From a typographical standpoint in latex, consider using “.” instead of “.”]. Check, e.g., Eq. 15 vs Eq. 16 and the use of the dot.

We thank for the practical latex advice and applied it to the scientific notation. Furthermore, we introduced a consistent use of the dot in all equations for better readability, i.e., dots are included for multiplication of alphabetic variables and function symbols, but excluded for (de)nominators with more than one summand and for explicit numerical factors like 2 or 2π .

(35) I dislike that Infobox 1 has figure 1, and therein, figure 1 has a caption. It seems best to simply refer to it as Infobox 1. There is no difference between “Infobox 1” and “Infobox 1, figure 1”, it is just a mouthful the last one. Idem from Infobox 2.

We agree with the referee. Originally, it was only introduced because of problems with resolving the Infobox references in latex. We worked on that and could define a customized box that allows for proper referencing.

(36) “Supplemental Material” and “Supplemental Information” are both use. At best, stick to one style.

We now use consistently “supplementary information”. This is the term used in the author guidelines of the journal.

(37) Second, I would exempt from writing “Supplemental Material, section S1”. “Supplemental Material S1” or “Supplemental Information S1” is clear.

We moved to the shorter expression.

(38) The term MCMC is never explained.

Definition is inserted.

(39) The use of “o’clock” should be removed. It reads too colloquial. I would simply suggest adding a sentence on the first reference to the time of the event to start that all measurements are in local, PDT time, and then only use the time, without “o’clock”.

We inserted a footnote for the first time stamp and deleted the “o’clock”.

(40) Please double check the term “1:00”. It should read “01:00”.

Corrected.

(41) Various times one find “on 11th August 1996” or “at the 11th August 1996”. This should always be “on the 11th August 1996”. Either “the” is missing or “at” should be “on”.

Corrected.

(42) The use of $\delta[\omega](t)$ seems cumbersome to me. This reads as an operator on ω , where it is just the detrended timeseries ω . I think a tilde above would suffice and be cleaner.

We believe that the “detrending” can be seen as kind of an operator on the raw time series object. However, we followed the referee’s suggestion for a more lightweight notation.

(43) *The font size of the figures is generally too large. This can be amended later once placed in the final template.*

We will let this point open as indicated by the referee.

(44) *The sub-sub-labels in the Infoboxes are excessive. “Infobox 1, figure 1 a.1”. This could easily just be “Infobox 1 b)”.*

We changed the labels accordingly.

(45) *In Sec. 1: “bifurcation-induced transitions (B-tipping)” is not italicised; “rate-dependent tipping (R-tipping)” is italicised; “noise-induced tipping” is partially italicised. Please be consistent. Remove “events” after the last, for consistency.*

We corrected for that.

(46) *In last sentence of page 1, we find “ – as they are needed in general for leading indicator analyses –”. This seems superfluous and can be removed without any loss of information.*

Removed.

(47) *Add the abbreviation “BL” in the abstract.*

We added BLE in the whole manuscript.

(48) *In page 4 we find both American English in “favored” and British English in “favourable”. Best choose one style only.*

We read through the manuscript and adapted the language to our best knowledge to British English.

(49) *There is a mixed used of “hydro plant” and “hydroelectric power units”. Stick to one style.*

We use now consistently “hydroelectric power units”.

Response to Referee 3

(1) *In this manuscript it is argued that statistical early warning signals (EWS) can be obtained from measurements of frequency fluctuations in the AC power net preceding a power disruption. It is argued that such a disruption can be modelled as a noise-induced tipping similar to standard transitions in stochastically driven bifurcating dynamical systems.*

The Referee is right that the stochastic noise contributions play a crucial role in the analysis. Around 15:40:09.00 the fast-scale process’s noise increases. Nevertheless, we want to point out that we do not argue for one specific route to destabilization, i.e, pure B-tipping or N-tipping. The overall analysis rather suggests a complex interplay of changing dynamics on the slow and the fast scale of the aggregated swing equation (ASE) (cf. following comments) which are related to different tipping risks. Apart from the increasing noise amplitude, the loss of the McNary power units is resembled in a decreasing restoring force (cf. Figure 1) which is in favour of B-tipping. Furthermore, the analysis reveals that cascading failures cannot be seen as smoothly destabilizing B-tipping events as argued in previous studies [9, 18]. Instead, individual events can have severe impact in a short period of time, e.g., the mentioned loss of the McNary units, or they are stretched over longer periods. Real world examples for the latter are indicated by the continuous positive drift slope trend after the McNary loss because of insufficient damping and the continuous negative trend over the restoration interval’s

green shaded period.

Note that in the following we changed the order of the Referee’s comments below for a more streamlined argumentation. Of course, all comments are included. However, we prefer to start with the Referee’s major suggestions.

(2) I recommend that the authors, firstly, specify which bifurcating or non-bifurcating model is relevant for describing the observed power grid. (I am not an expert).

We believe that the Referee’s advice largely improves the quality of the article. We include much more theoretical background of related frequency and power grid models on various descriptive levels and in close relation to the NAWI real world scenarios analysed in the main article to adequately address the Referee’s suggestion. In particular:

1. We sketch the (N)BLE briefly in the Introduction and outline the theoretical model motivation of its use for analysing power grid frequency time series ω . Briefly, the aggregated swing equation (ASE)

$$\dot{\omega} = -\bar{\gamma} \cdot \bar{\omega} + \bar{\sigma} \bar{\xi}(t) \quad (1)$$

with the center of inertia or bulk frequency $\bar{\omega}$, restoring rate $\bar{\gamma}$, and a noise process $\bar{\xi}(t)$ with amplitude $\bar{\sigma}$ can in a first approximation describe the power grid frequency in stable operation. However, the linear approximation leads to diverging frequencies for $\bar{\gamma} > 0$ without an alternative stable state which is not observed in reality. Furthermore, real frequency data hint to nonlinear terms present in the drift. An example are frequency signals from the Ireland grid [19], but also the analysed NAWI pre-outage time series (cf. SI S6). Compared to the (N)BLE drift parameterisation, the nonlinear drift derivative ζ and the (N)BLE noise level σ are thus closely related to the restoring rate $\bar{\gamma}$ and the noise level $\bar{\sigma}$ of eq. 1, respectively.

2. The equation can be derived from the mesoscopic, i.e., less coarse-grained, classical swing equation (CSE) that models power grids as networks of coupled nonlinear oscillators. The CSE is mainly used to study self-synchronisation and topological effects in power grid dynamics. In this description, nodes represent basically generators and consumers connected by graph lines which mirror electric transmission lines, such as the Keeler-Allston line. In contrast to the macroscopic ASE frequency description, the mesoscopic CSE can directly involve bifurcation- and noise-induced transitions. Since the ASE and CSE are closely related we discuss their connections in more detail in SI S4. To this end, we employ a simple mesocale CSE model which reproduces the observed BLE and NBLE signatures of the real world scenario including the the pre-outage region (with increased fast-scale system stress), the Keeler-Allston line tree-related high impedance fault (THIF), and the loss of the McNary power units with subsequently insufficient damping. Note that the model is oriented on the documented conditions and real cascading episodes outlined in the approved report and kept as simple as possible. In that way, we yield a better understanding of how the (N)BLE estimation of the ASE is related to disturbances from the lower descriptive scale of the CSE in the context of the historic NAWI cascading failure (cf. following answer). The results strengthen the interpretation in the main article.
3. Finally, we also remark that, if needed, the ansatz of the (N)BLE drift-diffusion parameterisation is rather flexible and straightforward to be adapted to specific systems in other fields or might be applied in a black box fashion to systems that lack a proper model description.

(3) Secondly, they should specify, which part of the fluctuations can be modelled as a drift and which part can be modelled as stochastic. (the h- and the g-functions).

That is valuable feedback and we guess that the comment partially implies that Infobox 1 is not introduced well. Therefore, we address this comment twofold. Firstly, we have revised the Infobox explanation and include more details, such as different possible levels of coarse-grained model descriptions in which the states \underline{x} might be represented. Furthermore, we describe more precisely its relation to the frequency ω . Furthermore we state in the revised version:

1. It shall predominantly explain the basic concepts on an intuitively accessible level. These conceptual descriptions cover the studied system (power grids), related components, and the modelling thoughts (Langevin equation).
2. The starting point of our estimation is the Langevin equation for macroscopic observables which can be derived from the system components’ microscopic dynamics, e.g., by the well-established Mori-Zwanzig

formalism [20, 21]. This results generally in a separation of resolved slow deterministic dynamics and fast dynamics which are described in a stochastic fashion instead. We include a remark on that in the revised version.

3. Furthermore, in the revised manuscript we emphasize that we model the frequency dynamics and not individual power grids' components. Nevertheless, the components are related to the frequency dynamics which is the mentioned macroscopic observable. The rare case of a well-documented cascading failure like the one under study is a valuable direct source to better understand such relations. Moreover, please note that the contributions to the drift h and diffusion g depend on the resolution of the time series that we model. We include a remark on this in the manuscript. In particular, we study the frequency dynamics on a second to subsecond scale on which the stabilizing dynamics, such as primary control, and grid disturbances as the Keeler-Allston line THIF live.

Secondly in this time domain, we want to answer the question what is contributing to which function from two perspectives:

1. From a formal point of view, the drift $h(\omega(t), t)$ models the trend components of the studied data, i.e., the force that attracts the frequency towards its stationary reference value. The residuals of that trend correspond to the remaining fast stochastic dynamics $g(\omega(t), t)\Gamma(t)$.
2. From a more technical point of view, the contributions relate to specific grid components. There is no precise microscopic model that can state for a given time series resolution which component will go into the deterministic and stochastic part and we highlight in the manuscript that, e.g., Infobox 1, is only an approximative illustration of the modelling ideas. However, please note that these approximations are not taken out of the blue. They are supported by the analysis of the real world NAWI power outage and previous studies [12, 22]. In our opinion, this makes the NAWI outage an exceptional example of how real world disturbances can affect the drift and diffusion of the power grid frequency and their time evolution. In particular, the discussed primary control of power units lives on a time scale of seconds and the loss of the McNary power units directly goes into the estimated drift slopes. Additionally, the restoration of the physical grid components and resupply of the customers influences the deterministic part. That is why we tend to assign the traditional power units to affect mostly the deterministic part for the given time series resolution. At the same time, several studies deal with the strong influence of the much faster and unpredictable power supply of renewable energy sources (cf. ref. [13] as an example). For that reason, we approximately attribute them to the stochastic part. In the end, most of the consumer decisions take place randomly on fast time scales. An example might be the extensive use of air conditioning because of the heat wave prior to the considered power outage or the mentioned high energy exports due to good conditions for the hydroelectric power units. They might be related to the much higher noise level of the pre-outage time series compared to the post-outage time series which covers mostly the morning of the next day in which the weather and trading conditions might already differ from the past day. Further evidence is provided by the mesoscale NAWI outage model based on the CSE which is closer related to individual components (but still a significant simplification) (cf. SI S4). Therein, we show that increased mesoscopic noise levels are captured by the (N)BLE noise levels. Furthermore, the Keeler-Allston line's THIF model adds further insights into the contributions to $h(\omega(t), t)$ and $g(\omega(t), t)$. The mesoscale noise amplitude of the CSE model remains constant, but the line's transmission properties change, i.e., the coupling between two nodes. In consequence, the NBLE noise level of the ASE increases. That means a modified signal propagation on lower scales can affect the coarse-grained ASE $g(\omega(t), t)$. We denote that as a relative noise increase which quantifies the dominance of fast-scale fluctuations from lower scales in the ASE. In contrast, the load imbalance of the McNary units is captured by $h(\omega(t), t)$ which is also supported by the NAWI model. Not least, in the very end of the pre-outage interval, we find increasing drift slopes and noise levels. The report states that signal oscillations were not sufficiently damped by the equipment directly before the outage. Following equation 1, the missing damping on the less-coarse grained CSE level goes into the restoring rate $\bar{\gamma}$, i.e., $h(\omega(t), t)$ of the ASE. However, on the descriptive level of the ASE, it also affects the noise level $g(\omega(t), t)$, since slow and fast signal contributions are less damped. These additional results provide a more complete picture of the ongoing processes during the cascading failure and help to better interpret the (N)BLE metrics related to different model scales as provided by the ASE and CSE. We modified the respective paragraphs and added several sentences and remarks to clarify that we understand the discussion as an illustrative and approximate reasoning which is supported by our empirical findings.

(4) Finally, they should specify what is the Bayesian part of the analysis. (It seems to me that the "credibility bands" are mere statistical "confidence bands", which is also fine, but for a mathematically oriented paper to be

relevant to the reader, it must be precise and correct).

We thank the Referee for the constructive comment. We addressed this issue by including an overall much more detailed description of the method in the revised Methods section and a brief sketch in the introduction. Briefly, the employed Markov Chain Monte Carlo sampled joint parameter posterior distribution allows for the derivation of percentiles from the drift slope and marginal noise level distributions, such that we end up with Bayesian credibility bands. We added also some information in the Method section to clarify that.

(5) The paper is obscure, with a mixture of qualitative (inaccurate) descriptions of electric power grids, poor descriptions of the mathematics of tipping scenarios and specific description of the North American Interconnection power blackout in 1996.

We agree that there were several points to streamline the article’s presentation. We followed carefully the suggestions of the referees to improve the overall quality of the manuscript in this regard. We hope that this addresses the Referee’s criticism conveniently. For example, we emphasize in the manuscript that the general description of power grids is simplified and mainly used for illustration purposes. The goal is to provide the necessary basic understanding (fixed frequency over the whole grid for balanced power input-output, basic ideas of Langevin model) to follow the analysis of the real world example which is basically the main topic of the article. The description of tipping scenarios follows mainly the common sense of practically oriented research articles in the field (cf. [6, 7]) which is why we neglected going into further mathematical detail. However, we include some references which deal with the description of tipping phenomena in a more formal fashion (e.g. ref. [3]).

(6) Infoboxes 1,2: There is very obscure connection between the stochastic differential equation (“Langevin model”) and the diagram below in (a). The cartoons with mountains and houses are really not helpful for describing the transitions in a potential. Pdf’s in b.2 seems to be taken out of the blue and the time-independent potential $V(x)$ in the text is an integral of an explicitly time-dependent drift function $h(x(t),t)$.

As outlined in previous comments we largely improved the description of the theoretical background and related models for the (N)BLE analysis of the NAWI cascading failure. A detailed description on the fine-grained level of the diagram with several individual components, which exhibit complex dynamics on their own, and connected layers of different grids might be out of reach at the current state of scientific knowledge and would probably test the limits of currently available hardware. However, such a detailed description is just not the idea of the Langevin ansatz. The strength of the Langevin description is to reduce the overall modelling effort by keeping the most relevant part of the dynamics. Therefore, the equation in Infobox 1 should be seen as as summarising “headline” of the otherwise too complex microscopic dynamics. We explain this better in the revised version. Nevertheless, our specific analysis of a real world event provides deeper understanding of how the single components may go into the deterministic and stochastic contributions of the Langevin model. For example, the power units and restoration actions of the physical grid act on the deterministic scale, whereas the short circuit of the Keeler-Allston line influences mainly the stochastic part of the better fitting NBLE. These real world findings for the NAWI outage are further confirmed by first modelling attempts on differently coarse-grained scales (cf. SI S4). This fills at least partially the gap between individual components and the macroscopic Langevin model of the grid frequency. Concerning the description of the transitions in a potential, we refer to several examples which use similar illustrative explanations. In the manuscript we adapt them to the specific power outage and the involved mixed tipping scenarios. We are conscious about the figurative style of our versions which are adapted to the overall illustrative diagram style. We agree with the Referee that they do not resemble reduced mathematical figures. For this reason, we decided to place them apart into Infoboxes. The simple reduced figures of this type are well-known and we cite related articles. In that sense, a reader can also stick to the reduced versions of other articles. The Referee is right that the pdfs do not resemble a specific scenario. Since the Infoboxes aim to give an intuition for the dynamics and mathematics rather than precise derivations, the pdfs are reasonably designed to fit the N-tipping description in the potential landscape of Infobox 1(d). We added a sentence to avoid confusion about that fact. At last, we thank the Referee for the hint to the missing notation of the time dependence of the potential which is added in the revised version. Basically, this time dependence is mirrored by the rolling windows of the estimation method.

(7) Figure 3 shows (quite arbitrary) sets of simulations of pitchfork and fold bifurcation scenarios with constant and changing noise intensities. The EWS are calculated without mentioning of running windows, with “credibility bands”, without specifying anything about the Bayesian-model part of the BL-estimation (BL meaning Bayesian-Langevin). It is very difficult to get any useful information from this figure.

The Referee is right about the missing hint to the rolling windows and the Bayesian method details in the introductory part of the article. We added those in the introductory part and the figure caption. Please note that Figure 3 is now placed in SI S1. We restructured this part to complement the (N)BLE comparison to AR1 and STD by the related parametric Ornstein-Uhlenbeck estimator (OUE) suggested by Referee 1. Following the comments of the referees, we also include a more detailed description of the methods in the Method section. We refer to the detailed description in the appropriate positions in the introduction and the figure captions. The main advantages of the data-driven parametric BLE in contrast to relying exclusively on the AR1, STD statistics or the OUE are as follows:

1. The parametric Langevin measure gives quantitative estimates of the drift slope and the noise level. Based on that, the distance of the drift slope to zero is a quantitative measure of resilience that is comparable to results for data of equal time resolution from a congruent system.
2. The parametric Langevin measure is able to resolve the different dynamics of the four examples, whereas the statistical measures AR1 and standard deviation are not able to do so.
3. Furthermore, the statistical leading indicators are not applicable in the example data of Figure 1(b) due to an ambiguous trend in the standard deviation.
4. The OUE is much less robust and exhibits significantly higher deviations from the ground truth than the BLE counterparts. Not least, they diverge for AR1 $\rho_{\tau_1} < 0$ (cf. SI S10)

(8) *The descriptions in Section 2.1 of the conceptual models is poor. Eqs. (5) and (8) does not describe noise intensities as shown in Figure 4 f, i.*

We thank the referee for the careful vision of the mathematics. Indeed, we forgot to adapt the t -dependence in the second part of the intervals. The issue is solved by correcting the dependence to $(t - 1000)$.

(9) *Statements as in lines 250-252: “Note also that for monitoring uprising tipping risk factors the exact estimates are much less important than their relative time evolution which encodes the desired information.” Is completely obscure to me, and this is just an example.*

We are sorry for the misunderstanding. We modified and complemented the respective formulations and added precise definitions of the chosen wording in the introduction:

1. We denote leading indicators based on significant positive trends (such as AR1 ρ_{τ_1} and STD $\tilde{\sigma}$), but without employing a parametric model as *qualitative* or *non-parametric* [4–7] ones.
2. Alternative measures which rely on parameterisations of the data generating processes to derive reliable resilience measures are denoted as *parametric* [3,23] leading indicators. For example, (N)BLE, OUE.
3. Finally, we denote estimates that deviate from a known ground truth generally as *biased* ones throughout the article.

What we intended to say was the following: The commonly discussed leading indicators AR1 and STD are essentially qualitative measures for an increasing tipping risk, i.e., there is no threshold and only a weak connection to a parametric model (like the previously mentioned OUE [3] (cf. also SI S1)). Instead analyses with these indicators have to rely on trend changes without definition of a destabilization threshold. These trend changes are termed “qualitative” in the manuscript. In contrast, the Langevin estimation is shown to yield quantitatively unbiased results in the examples with precisely defined destabilization threshold of zero in the case of Markovian dynamics (cf. ground truth vs. estimates in Figure 2). It also reconstructs a reasonable trend in non-Markovian cases, i.e., the measures become comparable to the qualitative reasoning of statistical leading indicators AR1 and STD. This holds for drift slope and noise level estimates apart from the noise amplitude in the B-tipping scenario which we already discussed separately in the original version of the manuscript. In addition, we present the advanced NBLE procedure which yields unbiased estimates also for the non-Markovian datasets, i.e., it clearly outperforms the basic statistical measures. We added the expansion of the method in the corresponding Method section and the quantitatively correct results for the non-Markovian examples in figure 2 of the revised version.

(10) *Other examples from the conclusions: “Nonetheless, the qualitative evolution of the stability and the noise level is also possible if the assumption is violated which is an important implication for many real world*

applications”. (line 492) I simply cannot make sense of such a sentence.

This goes back to the answer to the comment above. We corrected the imprecise wording accordingly.

References (Revision I)

- [1] Martin Heßler. antiCPy. <https://github.com/MartinHessler/antiCPy>, 2021.
- [2] Martin Heßler. antiCPy’s documentation. <https://anticpy.readthedocs.io>, 2021.
- [3] Andreas Morr and Niklas Boers. Detection of approaching critical transitions in natural systems driven by red noise. [arXiv](https://arxiv.org/abs/2010.12345), oct 2023.
- [4] Annelies J. Veraart, Elisabeth J. Faassen, Vasilis Dakos, Egbert H. van Nes, Miquel Lurling, and Marten Scheffer. Recovery rates reflect distance to a tipping point in a living system. *Nature*, 481(7381):357–359, dec 2011.
- [5] Vasilis Dakos, Stephen R. Carpenter, William A. Brock, Aaron M. Ellison, Vishweshha Guttal, Anthony R. Ives, Sonia Kéfi, Valerie Livina, David A. Seekell, Egbert H. van Nes, and Marten Scheffer. Methods for detecting early warnings of critical transitions in time series illustrated using simulated ecological data. *PLoS ONE*, 7(7):e41010, jul 2012.
- [6] Marten Scheffer, Jordi Bascompte, William A. Brock, Victor Brovkin, Stephen R. Carpenter, Vasilis Dakos, Hermann Held, Egbert H. van Nes, Max Rietkerk, and George Sugihara. Early-warning signals for critical transitions. *Nature*, 461(7260):53–59, sep 2009.
- [7] Marten Scheffer, Stephen R. Carpenter, Timothy M. Lenton, Jordi Bascompte, William Brock, Vasilis Dakos, Johan van de Koppel, Ingrid A. van de Leemput, Simon A. Levin, Egbert H. van Nes, Mercedes Pascual, and John Vandermeer. Anticipating critical transitions. *Science*, 338(6105):344–348, 2012.
- [8] Niklas Boers and Martin Rypdal. Critical slowing down suggests that the western greenland ice sheet is close to a tipping point. *Proceedings of the National Academy of Sciences*, 118(21), May 2021.
- [9] Eduardo Cotilla-Sanchez, Paul D. H. Hines, and Christopher M. Danforth. Predicting critical transitions from time series synchrophasor data. *IEEE Transactions on Smart Grid*, 3(4):1832–1840, dec 2012.
- [10] Daqing Hou. High-impedance fault detection — field tests and dependability analysis. <https://api.semanticscholar.org/CorpusID:142456396> (Retrieved: 24 October 2023), 2015.
- [11] John Goodfellow and Paul Appelt. How trees cause outages. <https://www.eci-consulting.com/wp-content/uploads/2017/10/How-Trees-Cause-Outages-ECI-Research-Summary.pdf> (Retrieved: 24.10.2023), 2023.
- [12] Benjamin Schäfer, Christian Beck, Kazuyuki Aihara, Dirk Witthaut, and Marc Timme. Non-gaussian power grid frequency fluctuations characterized by lévy-stable laws and superstatistics. *Nature Energy*, 3(2):119–126, January 2018.
- [13] Andreas Ulbig, Theodor S. Borsche, and Göran Andersson. Impact of low rotational inertia on power system stability and operation. *IFAC Proceedings Volumes*, 47(3):7290–7297, 2014.
- [14] Prabha Kundur. *Power system stability and control*. McGraw-Hill, New York [u.a.], [nachdr.] edition, 2007.
- [15] S.G. de Boer. The lamperti transform. applications to stochastic local volatility models. mathesis, TU Delft, 2020.
- [16] Don Watkins (BPA), Bernie Alfonso (SDGE), Dean Athow (WAPA), Don Badley (NWPP), Jim Beck (NCPA), Sharon Blair (BPA), Mark Bonsall (SRP), Steve Brockschink (COE), Jim Burns (BPA), Gene Byars (NERC), Paul Carrier (DOE), Douglas Cave (BCHA), Bill Comish (WSCC), Jon Daume (BPA), Robert Dintelman (WSCC), Jim Eden (PGE), Christine Elliott (NWPP), Michael Flores (TEP), John Forman (TANC), Kevin Graves (WAPA), Dharam Gupta (PAC), Mark Hanson (IPC), Bob Harshbarger (PSPL), Jack Kelley (SCL), Eric Law (PG&E), Lou Leffler (NERC), Eddie Lim (SMUD),

Mike Lopez (SCE), Wayne Mays (PGE), Al McCuen (CEC), Don McInnis (NERC), Bill Mittelstadt (BPA), Bernie O’Connell (PAC), Brad Osburn (WAPA), Leslie Padilla (SDGE), Vern Parry (COE), Dan Sauter (IPC), Ron Schelberg (IPC), Joe Seabrook (PSPL), Bob Smith (APS), Bob Stuart (PG&E), Brian Theaker (LDWP), Yu Wang (PAC), and Scott Waples (WWP). Western Systems Coordinating Council. Disturbance Report. For the Power System Outage that Occurred on the Western Interconnection. August 10, 1996. 15:48 PAST. Approved by the WSCC Operation Committee on October 18, 1996, oct 1996. The text source without Appendices was provided by Bonneville Power Administration via a Freedom of Information Act request thanks to James King (FOIA Public Liaison, BPA) and Brian Roth (FOIA Case Coordinator, BPA). The Appendices 2,3,5,9 of the timeline were provided thanks to Mary Schaff from the Washington State Library, currently operating under the Secretary of State, Steve Hobbs, via mail correspondence with the "Ask a Librarian". (Washington State Library, Point Plaza East, 6880 Capitol Blvd. SE, Tumwater, PO Box 42460, Olympia WA 98504-2460, Phone: (360) 704-5200, Email: askalibrarian@sos.wa.gov). A low quality scan of the restoration frequency time series from the approved report’s Exhibit 10 was provided by WECC, a readable version from the preliminary report by Jeanie Fisher from the Seattle Municipal Archives (Seattle Municipal Archives, 600 Fourth Avenue, Third Floor, Seattle, WA, 98104, PO Box 94728, Seattle, WA, 98124-4728, Phone: (206) 684-8353, Email: archives@seattle.gov). The contact with Jeanie Fisher of the Seattle Municipal Archives was established by Mary Schaff.

- [17] H. Haken. Synergetics: Introduction and Advanced Topics. Physics and astronomy online library. Springer, 2004.
- [18] P Hines, E Cotilla-Sanchez, and S Blumsack. Topological models and critical slowing down: Two approaches to power system blackout risk analysis. In 2011 44th Hawaii International Conference on System Sciences. IEEE, 2011.
- [19] Ulrich Oberhofer, Leonardo Rydin Gorjão, G. Cigdem Yalcin, Oliver Kamps, Veit Hagenmeyer, and Benjamin Schäfer. Non-linear, bivariate stochastic modelling of power-grid frequency applied to islands. arXiv, June 2023.
- [20] Yen Ting Lin, Yifeng Tian, Marian Anghel, and Daniel Livescu. Data-driven learning for the mori-zwanzig formalism: a generalization of the koopman learning framework. arXiv, jan 2021.
- [21] Eric Darve, Jose Solomon, and Amirali Kia. Computing generalized langevin equations and generalized fokker–planck equations. Proceedings of the National Academy of Sciences, 106(27):10884–10889, July 2009.
- [22] Benjamin Schäfer, Leonardo Rydin Gorjão, G. Cigdem Yalcin, Ellen Förstner, Richard Jumar, Heiko Maass, Uwe Kühnapfel, and Veit Hagenmeyer. Microscopic fluctuations in power-grid frequency recordings at the subsecond scale. Complexity, 2023:1–13, March 2023.
- [23] S. R. Carpenter and W. A. Brock. Early warnings of unknown nonlinear shifts: a nonparametric approach. Ecology, 92(12):2196–2201, dec 2011.

Answer Letter

We thank the Reviewer again for his valuable advice. We carefully followed the suggestions. Please find our answers attached.

Reviewer #1 (Remarks to the Author):

The most recent version of the manuscript is in my view suitable for publication if the authors address a few final points, listed below. The editor may decide whether the points have been addressed in a subsequent revision (but I am happy to have another look if needed).

My main reason for supporting publication for the revision (changing from previous recommendations) is that the presentation has now much better focus on the main contribution with fewer distractions than previous revisions. In my view the main contribution is the demonstration of the (N)BLE estimators for first-order resilience, often used as "Early-Warning Indicators" (EWIs) for tipping. It is instructive to see how including higher-order terms in the estimator improves the robustness of the first-order terms (not only in synthetic examples but also in credible real-world examples).

The contribution is mostly a methodological advance. Nat. Comm. is not the ideal venue for proposing and demonstrating methods. However, the methodology is an obvious improvement over the EWIs used frequently in high-profile general-interest science journals to discuss imminence of tipping, such that a case for publication in Nat. Comm. exists.

I am not convinced that the real-world example (the NAWI power outage in 1996) is a strong case for using these EWIs: the time series before the outage has highly coherent periodic oscillations (at about 0.2-0.3Hz) with an amplitude 10 times larger than post-outage. The authors admit in the SI that their methods are not designed to capture such periodicity, resulting in large deviations from the known ground truth in the resulting autocorrelation estimates (Fig S12 left). In addition to these oscillations, large spikes corresponding to a tree-related fault of the power line serve as clear problem-specific EWIs, as the paper's discussion reveals. However, the paper demonstrates that the (N)BLEs behave consistent with theoretical expectations in such cases.

The following remaining concerns and objections can be addressed without major changes or additional analysis.

Response to Referee 1

(1) [*1: Remove overclaim] My main objection is still that the paper claims to "quantify tipping risks" in the title, even though their analysis does not provide an estimate for the tipping risk. The usage of "tipping risk" is an unsupported overclaim. As Nat. Comm. is a general interest journal the word risk should be used close to what the general public would understand, which is in this case of noise-induced tipping a rate: what is the probability of tipping per time unit? The authors refer to [11], which claims to "quantify resilience". The word resilience is sometimes used in the weak sense as in this manuscript and [11], but "risk", especially in relation to tipping, should involve the probability of tipping. The proposed (N)BLE method automatically implies an estimate for the tipping rate, because the authors approximate the potential "landscape" shown in their infobox S1 up to fourth order such that they can check if their estimated landscapes have local maxima ("tipping barriers"). Together with their estimated noise levels the authors' statistical method predicts the tipping rate and its uncertainty. However, the authors choose not to report this estimate for the power failure or for the synthetic examples. For the examples they could have then objectively checked if the estimate is realistic. If the authors expect this rate estimate to be extremely uncertain (because it requires extrapolation), such that they advise against using this estimate, then they also imply that knowledge of the restoring rate and noise level around the equilibrium leaves the tipping rate (risk) equally uncertain. In summary, please remove any claim that the (N)BLE is able to quantify tipping risk or resilience.

We have removed any claim that the (N)BLE is capable of quantifying tipping risk or resilience, both in the main article and the supplementary information, by using more precise technical language.

(2) [*2: Sensitivity to window size] Provide a comprehensive sensitivity analysis of dependence on window size for the NAWI power outage and the synthetic examples in the SI. Especially the NAWI pre-outage $\hat{\Psi}$ looks as if it is affected by window size dependent artifacts. The rapid increase of $\hat{\Psi}$ appears to be caused by a single spike, the subsequent plateauing looks as if caused by the 50s window size artifact. So, the argument that there is a "sustained sudden increase in noise level" depends on appropriately addressing of this sensitivity.

We thank the referee for this valuable feedback and provide a twofold response. First, we varied the window sizes by $\pm 25\%$ and $\pm 50\%$ compared to the chosen baseline window size for both the real-world and synthetic data of the NAWI power outage. An exception is made for the McNary loss model, where the window size is varied by -50% , -25% , and $+12.5\%$ to ensure rolling window estimations that are independent of both the McNary loss and the primary control action. For more details on the procedure and the results, please see SI S16. Briefly, the analysis demonstrates that BLE and NBLE fingerprints are robust against window size variation, aside from slower trend response and dampened impact of outliers for larger windows, which is a general and well-known feature of rolling window methods.

Second, the referee is correct that the increase at the end of the first red interval is a window size artefact caused by the single peak. In the text, we refer to "the beginning of the first red interval", which marks the onset of significantly changing drift-diffusion dynamics that last much longer than one rolling window length. To avoid confusion, we have highlighted this increase with a dimension line with two-headed arrows in Figure 1(d).

(3) [*3: Reproducibility] My original impression about availability of code was overly positive: the cited source https://github.com/MartinHessler/Disentangling_Tipping_Types only provides the time series shown in the paper and SI, not the code reproducing the figures. The source should contain scripts that one can call to reproduce the figures from the original time series, clearly listing the method parameters chosen. If necessary, a snapshot of the library version used should be included.

We have added the folder "Quantifying Local Stability and Noise Levels" to the repository mentioned above. In this folder, we provide Python scripts that clearly list all parameters and library versions used, enabling comprehensive reproduction and plotting of the results presented in the main article as well as all key arguments referring to the SI. In particular, we include data, computation codes, and plot scripts for Figures 1 and 2 as well as SI Figures S1, S3, S6, S12, S13, S14, and S15. Moreover, SI Figures S2 and S5 can be reproduced based on the provided data. Additionally, the SI is complemented by a short section (cf. SI S19) that lists the most important method parameters once again.

(4) [*4: Oscillatory nature of NAWI pre-outage time series] Clearly highlight that the pre-outage data has coherent oscillations (e.g., by an inset similar to post-outage in Fig.1) and alert the reader to this feature prominently the text. Also, for the analysis of the mesoscopic network model S.5-S.6 clearly demonstrate that these models reproduce the coherent oscillations with similar frequency and amplitude as the original data to make them credible. This essential feature is hard to discern from Figure S.3.

That is a good point which we have addressed by introducing insets in Figures 1 and S3. Additionally, we added several sentences in the SI text and the caption of Figures 1 and S3 to clarify the scope and limitations of the model simulations.

As described in SI S3, our aim was to construct the simplest possible CSE model that reproduces the (N)BLE fingerprints and aligns with the real timeline of events. It serves as a further theoretical foundation. In this sense, the model does not replicate the complex real-world topology of the NAWI, with thousands of nodes and lines, each with varying features, such as real power production and consumption by power plants, industries, and towns, or line capacities). Therefore, differences in the absolute magnitudes of the simulated data are expected.

However, we focused on approximately reproducing the features in the data that are most important for numerical analysis. The simulated data exhibits periodicities (0.2 Hz to 0.8 Hz) in the same order of magnitude

as the real-world data (0.2 Hz to 0.4 Hz). The magnitude of the amplitudes is less important, as it typically only modifies the absolute range of the estimates while leaving the analysed (N)BLE trends/fingerprints untouched. Nonetheless, we ensured that the averaged amplitudes (roughly ~ 0.5 a.u.) typically do not differ by more than one order of magnitude.

Smaller issues:

(5) * *Define THIF also in main text*

We thank the referee for this valuable correction and have introduced the acronym in the main text of the revised version. Additionally, we repeat the definition in the caption of Figure 1(a) to avoid confusion.

(6) * *I do not understand the binning procedure behind the blue curve in Fig S5: the data is highly oscillatory with amplitude about 0.02.*

We added a few sentences clarifying the procedure introduced in Refs. [1–3]. The increments \dot{x} of the time series data are computed and assigned to the corresponding data points x . These data are binned, averaged, and result in a non-zero drift function for the frequency data.

(7) * *extend the x axis in the left column of FigS12 (autocorrelation of original time series and (N)BLE) to include at least the first maximum of the autocorrelation (to about $\tau=60$?). This clearly illustrates the mismatch between the dynamics underlying the real data and the (N)BLE models.*

We followed the suggestion to make this point clearly perceptible.

Reviewer #1 (Remarks on code availability):

(8) *My original impression about availability of code was overly positive: the cited source https://github.com/MartinHessler/Disentangling_Tipping_Types only provides the time series shown in the paper and SI, not the code reproducing the figures. The source should contain scripts that one can call to reproduce the figures from the original time series, clearly listing the method parameters chosen. If necessary, a snapshot of the library version used should be included.*

Please see our answer to Comment (3).

References

- [1] R. Friedrich and J. Peinke. Description of a turbulent cascade by a fokker-planck equation. Phys. Rev. Lett., 78:863–866, Feb 1997.
- [2] S. Siebert, R. Friedrich, and J. Peinke. Analysis of data sets of stochastic systems. Physics Letters A, 243(5):275–280, 1998.
- [3] R. Friedrich, S. Siebert, J. Peinke, St. Lück, M. Siefert, M. Lindemann, J. Raethjen, G. Deuschl, and G. Pfister. Extracting model equations from experimental data. Physics Letters A, 271(3):217–222, jun 2000.

I have read through the manuscript *Quantifying Tipping Risks in Power Grids and beyond* by Martin Heßler and Oliver Kamps and found it of appreciative quality. I have a few remarks and questions, pertaining to both the presentation and mathematical formulation, which I hope the authors can take into consideration. I believe this work is well suited for a Nature Communication article, yet, as is, I find it to extensive and somewhat cumbersome to read. Firstly, I must commend the authors for what is, in my view, and excellent exercise into looking into power system's data from the 1996 blackout in the US and Canada! It is not often that data-driven analysis from *real-world* records are seen in power system's studies – much less from data dating a quarter of a century!

With this said, I would place this work as **major revision** – with a strong support for publication once some of the issues presented here are tended to.

Now, to my current concerns with the manuscript. Firstly, I think the title of the article deserves a particular mention of the blackout studied. About half the text in manuscript pertains to the analysis of this event. I suggest something in line with: “Quantifying Tipping Risks in Power Grids and beyond: An examination of the 1996 blackout” (maybe with the location in the title). Herein, I would also suggest removing “and beyond”. The paper does not cover any application outside power systems apart from numerical models. There is space to keep the “and beyond” – but that requires a more thorough discussion and outlook pertaining to application *outside* power systems. Note that in the discussion there is not a single mention to other fields of study *beyond* power systems!

Major remarks

The manuscript is very well written, but it is simply too long. The 1996 blackout case is discussed too extensively, with a lot of information being presented sometimes thrice! The example is exceptional – as is the work performed here – but it becomes too cumbersome for the reader. I would actually suggest one big addition, which could solve a lot of issues: Both a map of the affected areas, with the 3 and then 4 island formation, with a detailed timeline below, as a long left-to-right arrow, which each marking of the time stamps referred in the text. This would greatly increase the readability of the text – an image is a thousand words here. Overall, the case-study needs to be streamlined for readability.

As commented above, the manuscript lacks a certain congruence. The introduction focus heavily on tipping, the limitations of the two seminal CSD metrics, then we move to the Langevin model, and later the case-study of the 1996 power outage. The Discussion and Outlook focuses on the case study and steps forward to improve estimation using BL, yet we never arc back to discussing CSD metrics and only a few mentions are given to R-tipping. An outlook on the method is given, with suggestions to the Kramers' escape rate, which is a new topic in this regard.

Major questions

In Eq. 1 we find a vector formulation with underbar x . Strictly speaking, the diffusion underbar g is a matrix and not a vector, as it is multiplied by the noise vector underbar Γ . Have the authors chosen the current nomenclature for a particular reason? Furthermore, for consistency, should the left-hand side not read $x\text{-dot}(x(\mathbf{t}),t)$ instead of $x\text{-dot}(x,t)$? The authors are careful in including the time dependence of $x(t)$ on each function/vector/matrix on the right-hand side.

Continuing on this issue. The authors present, in the most general form, an N-dimensional Langevin equation, but the work and most subsequent examples are 1 dimensional, as is the case study. The justification for power-grid frequency seems solid, as it is generally almost perfectly identical regardless of where in a power grid it is measured. The work in this manuscript relies heavily on antiCPy, which, to my understanding, is also only a 1 dimensional time series analysis tool. The authors offer a convincing argument to the study of 1-D time series, where the presented N-D Langevin equation might be misleading. It is not straightforward, for voltage analysis across a power grid, to reduce the problem to a 1-D time series. Much less so for climate or ecological studies. Could the authors comment on this? I would push for a less ambitious case herein only a 1-D case is presented.

Eq. 19 has two parameters θ_0 and θ_1 on the left-hand side, yet depends only on θ_1 on the right-hand side. Is there something amiss here?

What is the time series Gemini?

It is hard for me to follow why Eq. 7 diffusion term is referred to a “coupled” noise amplitude. Coupled noise would seem to me to pertain to the non-diagonal diffusion matrix terms of a 2-dimensional stochastic process, which couples noise across the dimensions of a 2-d process. Is there a reason for this choice of wording?

For the correlated B-tipping model, the authors refer to Eq. 4 being coupled via the constant $q=0.5$, but there is *no* constant q in Eq. 4. Subsequently, the noise term from Eq. 7 is $\sigma=\sqrt{c}$, which is *not* related to the noise but to the drift term. I find this quite confusing. Could the authors try to alleviate this and include directly an equation for this process?

Remarks and smaller questions

In the second paragraph of the introduction, we find “[...] in connection with uprising bifurcation-induced transitions [...]”. What is the meaning of “uprising” here?

In the third paragraph of the introduction, we find “Furthermore, the currently very important trend to an environmentally sustainable grid architecture will probably augment the amount of noisy grid participants significantly in future”. This is a claim that requires a citation. I suggest the authors look at Milano *et al.* (2018) doi:10.23919/PSCC.2018.8450880.

The remark “Nevertheless, industrial and scientific stakeholders only report activations or deactivations up to a certain extent, and railway traffic can be heavily disrupted, for example, due to damage to the railway infrastructure caused by a storm or unforeseen strikes” seems out of place. I cannot follow what is the relevance of this statement in the context of the narrative. Could the author elaborate on this? Maybe best removed? -- I would also like to point out to the authors that, to the extent of my knowledge, this might not be the best example to use. Various EU countries operate their train systems and 16.6 Hz instead of 50 Hz. Admittedly the power source is still the same, but for the argument the authors are seeking, there are better examples to pick. (Some S-bahnen are even DC powered).

The use of the term “mountainside”, although clear, seems like more added baggage. A much more common term – which the authors also use – is “landscape”. This is a term used almost uniformly from psychology to ecology and physics. I suggest the authors stick to “landscape” only.

Pertaining to the sentence “Note that power outages as well as control mechanisms live on time scales of seconds up to several minutes, e.g. global grid primary control actions in Germany react to a maximum frequency deviation of ± 0.2 Hz in times up to 30 s and have to be stable under full load up to 15 min before the simultaneously starting secondary control in regional grids should replace the primary control completely.” Firstly, this seems to refer to article 31 which seems to point to regulations from 2003. Are these still the statutory limits in Europe/Continental Synchronous area? Secondly, it might be better to refer to the regulation of the Continental Synchronous area in general. Germany is but one of various country in this supergrid.

The (although commonly used) abbreviation “pdf” in the last paragraph in page 2 is not introduced. For consistency, introduce it.

The sentence “Since a sustainable adaptation of our power grids is indisputable” at the end of page 2, reads weirdly. I believe the authors intend to imply that renewable energies will become ubiquitous. The way it is written implies that there has been a steady adaptation of the power grid, which is not necessarily the same. I believe rephrasing this would bring the point to the readers in a clearer fashion.

Comparing the titles of Infobox 1 and Infobox 2, Infobox 1 should include a mention to “tipping”, as this discussed therein.

In Infobox 1, Figure 1, we read “[...] be captured by the diffusion $g(x(t),t)\cdot\Gamma(t)$ [...]” but further on, e.g., in page 4, bottom, “[...] in principle contribute to the diffusion [sic] $g(x(t)t)$ [...]” (note the missing comma!). In my view, g is the diffusion, not $g\Gamma$. It would be best to be consistent here.

In Infobox 1, “[...] If the restoring force (light green arrow line) given by the steepness of the potential mountainsides is high enough [...]”. I believe the correct English form here is “[...] is *large* enough [...]”, but do check as I am not a native speaker.

In Infobox 2, “[...] but under stationary noise level if the potential landscape is negatively affected”. I believe is this a bit of an abuse of language. This landscape is not an actual landscape, thus it cannot be negatively affected (in a sense of a loss of vegetation, flora, of fauna). This is still a function, which cannot be “negatively affected”. Maybe there is a better wording to be found here?

Eq. 7 includes, not erroneously, a noise term a dW . I would advise against this. In the current formulation, the authors use x -dot, which implies a $dx(t)/dt$ (more conventionally a physics notation) on the left-hand side. The drift on the right-hand side is correct, yet the final term of the equation is a differential, which is incongruent with the notation. I think it is best to stick with $\Gamma(t)$.

Eqs. 2, 4, and 6 all have their respective control parameters as $-v$, $-r$, and $-\alpha$, yet, their values are also all negative. Not critical, but feels a bit pointless. Would it not be more elegant to just have these are positive values and remove the minus signs?

Eqs. 2, 4, and 6 all introduce new notation for h and g with various different subscripts, like h_{pf} . This seems superfluous as there are never used in the text. The only terms mentioned are X_g and so on. I suggest alleviating the notation, as there is little gain here in separating h and g . They are all the same mathematical objects, just with different functional values in the various examples. Subsequently, the use of the term *set* in *N-tipping set* X_g might be misleading. These are not sets, just stochastic processes. Note that the last one is denoted *correlated B-tipping model* and not *set*.

There is also an inconsistency in the presentation of Eq. 6 and 7, which give the SDEs directly, instead of the drift and diffusion term. I think it would be more elegant to stick to one notation only and report is as in the previous examples.

In the same line as above, the last example, denoted *correlated B-tipping model* is not given the honour of an equation, just inline passages. It would be best to also give this model an equation, for presentational purposes.

Fig. 4 and Fig. 5 are missing they y-labels for the plots. It is best to include those. Moreover, the sub-plot labels need work: Fig. 4 has no parentheses and the labels are in the subplots, Fig.5 is the opposite. For Fig. 5 I suggest removing the legends entirely and instead include the terms in the y-label, i.e. $\hat{\sigma}$ and $\hat{\xi}$. The rest can be mentioned in the caption. For Fig. 4 all legends can be removed, $\hat{\sigma}$ and $\hat{\xi}$ added accordingly in the y-labels, and the rest referred to in the caption.

In section 2.2.1, the authors write “[...] grid components were satellite-synchronized.” I suggest the authors write “GPS-synchronized”. This is more appropriate for it clearly states what timing method was used (if that was indeed the case!)

In section 2.2.1, page 11, the authors include “Additionally, power grids are commonly modeled by coupled Kuramoto oscillators which contain a linear damping term. This damping can be interpreted in reality as control actions in order to stabilize the deterministic dynamics of the system.” This seems misplaced. The narrative is about actual events. The way power grids are modelled is perfectly unrelated to the question at hand.

In Section 3, page 14, the reference to Kramers’ rate requires a citation, as this might not be known to all readers.

In Section 5, between Eq. 10 and Eq. 11, “of the corresponding Fokker-Planck equation of the time series (a), with the potential”. What is the “(a)” here?

Below Eq. 17, they authors use “ $p(.)$ ”. Better to be explicit. This notation is used only once here.

Section 7 should just be part of a Supplemental Material.

Reference 33 includes various acknowledgements that I believe should be include in the acknowledgements, wherein I hope they are more visible, and thus pay homage to the people involved accordingly.

Minor issues, typos, and typography

All references to the use of Gaussian kernels for filtering data indicate the bandwidth chosen with a σ , which naturally pertain to the standard deviation of the Gaussian filter. In this particular paper, this adds a bit of confusion, since the noise term in the various models is also a σ , but that pertains to the magnitude of the noise, not the standard deviation of the pdfs of each model! I strongly advise using a different term for one or the other, to avoid confusion.

The use of the dot \cdot in the equations feels a bit unnecessary, and sometimes confusing. Eq. 1, which is written in vectorial form, implies a matrix operation on a vector, g on Γ , without the use of a product, which reads perfectly fine for me. From hereon, we find a consistent use of the dot to multiply scalar and functions, which seems excessive for a standard mathematical operation. I suggest the authors remove the dot, apart from the separation of scientific notation for, e.g. in Eq. 3,

where we read $2.4 \cdot 10^{-4}$. [From a typographical standpoint in latex, consider using “ $\{\cdot\}$ ” instead of “ \cdot ”]. Check, e.g., Eq. 15 vs Eq. 16 and the use of the dot.

I dislike that Infobox 1 has figure 1, and therein, figure 1 has a caption. It seems best to simply refer to it as Infobox 1. There is no difference between “Infobox 1” and “Infobox 1, figure 1”, it is just a mouthful the last one. Idem from Infobox 2.

“Supplemental Material” and “Supplemental Information” are both used. At best, stick to one style. Seconding, I would exempt from writing “Supplemental Material, section S1”. “Supplemental Material S1”/“Supplemental Information S1” is clear.

The term MCMC is never explained.

The use of “o’clock” should be removed. It reads too colloquial. I would simply suggest adding a sentence on the first reference to the time of the event to start that all measurements are in local, PDT time, and then only use the time, without “o’clock”.

Please double check the term “1:00”. It should read “01:00”.

Various times one finds “on 11th August 1996” or “at the 11th August 1996”. This should always be “on the 11th August 1996”. Either “the” is missing or “at” should be “on”.

The use of $\delta[\omega](t)$ seems cumbersome to me. This reads as an operator on ω , where it is just the detrended timeseries ω . I think a tilde \sim above would suffice and be cleaner.

The font size of the figures is generally too large. This can be amended later once placed in the final template.

The sub-sub-labels in the Infoboxes are excessive. “Infobox 1, figure 1 a.1”. This could easily just be “Infobox 1 b”.

In Sec. 1: “bifurcation-induced transitions (B-tipping)” is not italicised; “*rate-dependent tipping* (R-tipping)” is italicised; “*noise-induced tipping*” is partially italicised. Please be consistent. Remove “events” after the last, for consistency.

In last sentence of page 1, we find “ – as they are needed in general for leading indicator analyses –”. This seems superfluous and can be removed without any loss of information.

Add the abbreviation “BL” in the abstract.

In page 4 we find both *american English* in “favored” and *british english* in “favourable”. Best choose one style only.

There is a mixed use of “hydro plant” and “hydroelectric power units”. Stick to one style.

If any questions arise, feel free to contact me.
Leonardo Rydin Gorjão,
28/01/2024